# Grammars of Formal Uncertainty:
# When to Trust LLMs in Automated Reasoning Tasks

**Debargha Ganguly**[1], **Vikash Singh**[1], **Sreehari Sankar**[1], **Biyao Zhang**[1], **Xuecen Zhang**[1],
**Srinivasan Iyengar**[2], **Xiaotian Han**[1], **Amit Sharma**[3], **Shivkumar Kalyanaraman**[2], **Vipin Chaudhary**[1]

[1]Case Western Reserve University [2]Microsoft Corporation [3]Microsoft Research
{debargha,vikash,sreehari,bxz297,xxz1037,xhan,vipin}@case.edu
{sriyengar,amshar,shkalya}@microsoft.com
https://github.com/DebarghaG/grammars-formal-uncertainty (Quantify uncertainty)
https://github.com/DebarghaG/proofofthought (Auto formalization library)

## Abstract

Large language models (LLMs) show remarkable promise for democratizing automated reasoning by generating formal specifications. However, a fundamental tension exists: LLMs are probabilistic, while formal verification demands deterministic guarantees. This paper addresses this epistemological gap by comprehensively investigating failure modes and uncertainty quantification (UQ) in LLM-generated formal artifacts. Our systematic evaluation of five frontier LLMs reveals Satisfiability Modulo Theories (SMT) based autoformalization's domain-specific impact on accuracy (from +34.8% on logical tasks to -44.5% on factual ones), with known UQ techniques like the entropy of token probabilities failing to identify these errors. We introduce a probabilistic context-free grammar (PCFG) framework to model LLM outputs, yielding a refined uncertainty taxonomy. We find uncertainty signals are task-dependent (e.g., grammar entropy for logic, AUROC>0.93). Finally, a lightweight fusion of these signals enables selective verification, drastically reducing errors (14-100%) with minimal abstention, transforming LLM-driven formalization into a reliable engineering discipline.

## 1 Introduction

Formal methods offer robust mathematical guarantees for system reliability [Huth and Ryan, 2004], but their widespread adoption is impeded by high expertise and labor demands, traditionally limiting their application to safety-critical domains where failures have catastrophic consequences [Clarke et al., 2018, Woodcock et al., 2009]. Concurrently, Large Language Models (LLMs) have emerged with a remarkable ability to generate formal artifacts such as code, proofs, and specifications [Brown et al., 2020, Chen et al., 2021, Jiang et al., 2023a], potentially democratizing formal methods [Hou et al., 2023] and finding new roles in formally correct reasoning and LLM verification [Ganguly et al., 2024, Pan et al., 2023]. However, these two approaches embody fundamentally different epistemological paradigms. Formal methods are rooted in deterministic logical calculi, where conclusions derive necessarily from premises via unambiguous inference rules. LLMs, in contrast, operate probabilistically, representing knowledge as distributions over tokens where multiple, even contradictory, outputs can possess non-zero probability [Wei et al., 2022a]. This inherent tension presents a core challenge: *how can we harness the generative power of LLMs for formal reasoning while upholding the rigorous guarantees that define formal verification's value?*

The central thesis of this paper is that the inherent probabilistic uncertainty in LLM outputs for formal reasoning tasks, particularly when generating formal artifacts like SMT-LIB programs, is not a mere nuisance but a valuable source of information for guiding verification. Existing methods often ignore this by selecting only the highest-probability output [Chen et al., 2022], a simplification that we argue undermines the deterministic correctness guarantees required for formal verification. In contrast, we

demonstrate how to systematically capture and analyze this output uncertainty by modeling LLM-generated SMT-LIB program distributions with Probabilistic Context-Free Grammars (PCFGs). Instead of focusing on a single output, we analyze ensembles of LLM-generated SMT-LIB programs, treating these as samples from the model's internal probability distribution [Kadavath et al., 2022], which we then approximate by applying PCFGs to the ensembles. This approximation not only identifies the most likely solutions but also reveals strategic diversity, common structural motifs, and areas of high model uncertainty. Deriving a comprehensive suite of metrics from this structured, quantifiable understanding of uncertainty can then directly guide the verification workflow—for instance, by assessing artifact reliability, focusing human review on more ambiguous or structurally complex candidates, and improving error detection strategies.

The core contributions of this paper are:

- We systematically evaluated frontier LLMs on four formal reasoning datasets, finding SMT-based autoformalization significantly boosted accuracy on tasks like ProofWriter (+34.8%) but harmed others like FOLIO (-44.5%), thus quantifying LLM-driven formal verification's failure modes. We then demonstrate that known uncertainty quantification techniques do not capture enough information to identify errors in FV artifacts.
- Introduce a framework using probabilistic context-free grammars to model LLM-generated SMT-LIB programs, enabling mathematically sound uncertainty quantification and bridging neural models with formal verification.
- Developed and evaluated 25 uncertainty metrics, revealing a refined taxonomy (epistemic-knowledge, epistemic-procedural, recursive-complexity, capacity-limited) that offers a more nuanced understanding of uncertainty in neurosymbolic systems than the traditional epistemic/aleatoric dichotomy.
- Demonstrated that formal reasoning uncertainty is task-dependent and introduced a lightweight, model-agnostic fusion of these varied uncertainty signals. This approach outperforms individual metrics, improves calibration, enables selective verification to cut error rates by 14-100% with minimal abstention, and suggests modality-aware architectures for enhanced reliability.

## 2   Methodology

Generating formal artifacts using ad-hoc Domain-Specific Languages (DSLs) introduces significant engineering friction. This friction arises from the need to redesign generators, models, and parsers for syntax changes, and it also complicates debugging erroneous outputs (e.g. syntactically incorrect FV artifacts). To mitigate this overhead, we adopt the stable, widely supported SMT-LIB standard as a common intermediate representation targeting SMT solvers. In this section, we consequently present a theoretical framework linking language models and verification to analyze LLM-generated SMT-LIB program distributions, enabling principled uncertainty quantification.

**Problem Setup:** We formalize the probability space over SMT-LIB programs. Let $\Sigma$ be a finite alphabet. The set of all finite strings $\Sigma^*$ forms a measurable space $(\Sigma^*, \mathcal{F})$, where $\mathcal{F}$ is the $\sigma$-algebra generated by cylinder sets (strings sharing a common prefix $w$). The SMT-LIB language $L_{SMT} \subseteq \Sigma^*$, approximated by its standard context-free grammar $G_{SMT}$, is measurable in $\mathcal{F}$. For a task $T$ and an LLM with parameters $\theta$, the LLM induces a probability measure $\mu_{T,\theta}$ on $(\Sigma^*, \mathcal{F})$. The measure over valid SMT-LIB programs is then the conditional measure $\mu_{T,\theta,SMT}(A) = \mu_{T,\theta}(A \cap L_{SMT}) / \mu_{T,\theta}(L_{SMT})$ for $A \in \mathcal{F}$. This definition requires $\mu_{T,\theta}(L_{SMT}) > 0$, a reasonable assumption that is empirically validated, as LLMs are generally trained to generate syntactically valid code and formal specifications.

**Modeling Background:** To model distributions over structured programs like $\mu_{T,\theta,SMT}$, we employ Probabilistic Context-Free Grammars (PCFGs). PCFGs extend standard Context-Free Grammars (CFGs) by associating probabilities with their production rules. Formally, a PCFG is a 5-tuple $G = (V, \Sigma, R, S, p)$, where $V$ is a finite set of non-terminals; $\Sigma$ is a finite set of terminals disjoint from $V$; $R \subseteq V \times (V \cup \Sigma)^*$ is a finite set of production rules; and $S \in V$ is the start symbol, such that $(V, \Sigma, R, S)$ collectively form a CFG. The fifth component, $p: R \to [0,1]$, is a probability function assigning a probability $p(r)$ to each rule $r \in R$. For each non-terminal $A \in V$, these probabilities must satisfy $\sum_{r \in R_A} p(r) = 1$, where $R_A$ denotes the set of rules with $A$ as their left-hand side. The probability of a derivation $\pi$ that applies rules $r_1, ..., r_k$ in sequence is $p(\pi) = \prod_{i=1}^{k} p(r_i)$. Consequently, for any terminal string $w \in L(G)$, where $L(G)$ is the language generated by the underlying CFG, its probability under $G$ is $\mu_G(w) = \sum_{\pi \in \Pi(w)} p(\pi) = \sum_{\pi \in \Pi(w)} \prod_{r \in \pi} p(r)$, where $\Pi(w)$ represents the set of all leftmost derivations of $w$ from $S$. It is important to note that a PCFG $G$ defines a consistent probability measure (i.e., $\sum_{w \in L(G)} \mu_G(w) = 1$) if and only if the spectral radius of

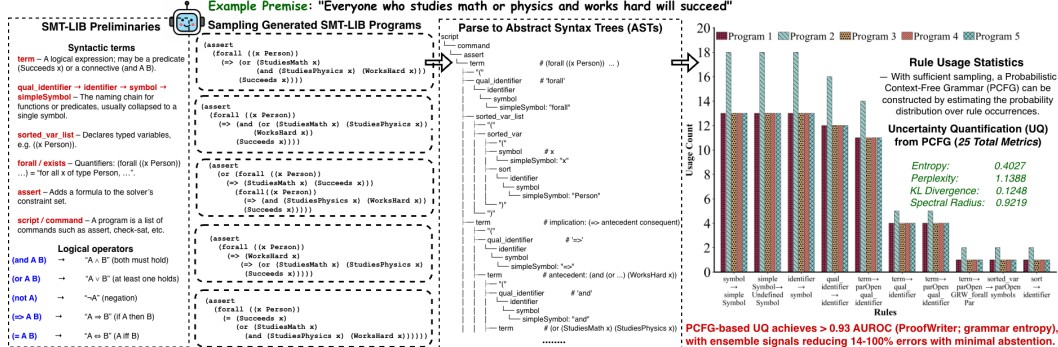

Figure 1: Empirical analysis of LLM-generated SMT-LIB formalizations for the statement "Everyone who studies math or physics and works hard will succeed." The figure illustrates (left) syntactic and logical forms in SMT-LIB, (center) multiple LLM-generated formal programs and their parsed Abstract Syntax Trees (ASTs), and (right) rule-usage statistics learned by a Probabilistic Context-Free Grammar (PCFG). With sufficient sampling, rule-occurrence frequencies define a probability distribution from which structural uncertainty metrics—entropy, perplexity, KL divergence, and spectral radius—are computed. These PCFG-derived uncertainty measures correlate strongly with correctness and formalization reliability, achieving AUROC ≈ 0.93 on ProofWriter (grammar entropy), with ensemble signals reducing 14-100% errors through selective verification. The analysis uses only observed LLM outputs (i.e., no synthetic or simulated data) to quantify uncertainty in formal reasoning.

its moment matrix $M_G$ is less than or equal to 1. This condition ensures that probabilities are well-defined and sum to one across the entire language generated by $G$.

*Note:* Constrained decoding with deterministic CFGs zeroes out grammar-invalid tokens at inference to enforce syntax (e.g., JSON/code). Unlike our PCFG-based UQ, these methods (i) ensure syntax validity rather than quantify uncertainty among already well-formed but potentially semantically wrong SMT-LIB outputs; and (ii) require white-box token probabilities, or modifications to the LM head, unsuited to black-box frontier API. We analyze valid outputs without changing model internals.

**Approximation:** To connect the theoretical LLM distribution $\mu_{T,\theta,SMT}$ with a tractable probabilistic model, we estimate parameters for a PCFG. We use the SMT-LIB v2 grammar $G_{SMT} = (V_{SMT}, \Sigma_{SMT}, R_{SMT}, S_{SMT})$ as its structural basis. We now generate $N$ SMT-LIB program samples $\mathcal{P}_N = \{P_1,...,P_N\}$ from the target LLM (parameterized by $\theta$) and parse them using $G_{SMT}$. This yields a set of parse trees $\Pi(\mathcal{P}_N)$ from the successfully parsed programs. From each parse tree $\pi \in \Pi(\mathcal{P}_N)$, we identify and record every applied production rule $r = (A \to \beta) \in R_{SMT}$. This record typically includes the rule itself, its source program identifier, structural context (such as depth), and optionally, the corresponding source text mapping for qualitative analysis. This data collection also allows for extracting richer contextual features than those used by standard Maximum Likelihood Estimation for estimating the rule probability function $p: R_{SMT} \to [0,1]$ from these rule application frequencies.

Maximum Likelihood Estimation (MLE) is used to estimate rule probabilities $p(r)$ using counts from $\Pi(\mathcal{P}_N)$. (Note that MLE itself is not an uncertainty quantification metric; rather, it is the method used to estimate the PCFG rule probabilities from which our uncertainty metrics are subsequently calculated.) Given total application counts $C(r)$ for a rule $r = (A \to \beta)$ and $C(A) = \sum_{r' \in R_A} C(r')$ for its left-hand side (LHS) non-terminal $A$ (where $R_A = \{r' \in R_{SMT} \mid \text{left}(r') = A\}$), the MLE is its relative frequency $\hat{p}_{MLE}(r) = C(r)/C(A)$, defined if $C(A) > 0$. If $C(A) = 0$, rules in $R_A$ are assigned a uniform probability $1/|R_A|$. For independent and identically distributed (i.i.d.) samples $\mathcal{P}_N$ from $\mu_{T,\theta,SMT}$, these estimated probabilities $\hat{p}_N(r)$ converge almost surely (a.s.) to $p^*(r)$ as $N \to \infty$. The limits $p^*(r)$ are the parameters of the $G_{SMT}$-based PCFG that is closest in Kullback-Leibler (KL) divergence to the true distribution $\mu_{T,\theta,SMT}$ (i.e., $p^* = \text{argmin}_p D_{KL}(\mu_{T,\theta,SMT} \| \mu_G(p))$). For finite $N$, additive (Lidstone) smoothing with $\beta_s > 0$ (e.g., $\beta_s = 1$ for Laplace smoothing) addresses problematic zero counts where $C(r) = 0$ but $C(A) > 0$, yielding $\hat{p}_N^{(\beta_s)}(r) = (C(r) + \beta_s)/(C(A) + \beta_s |R_A|)$.

**Theorem 1** (Coverage Guarantee). *Let $\mu$ be a distribution on a discrete sample space $\Sigma^*$, with Shannon entropy $H(\mu) = -\sum_{x \in \Sigma^*} \mu(x) \log_2 \mu(x)$. Suppose we draw $N$ i.i.d. samples from $\mu$. Then, for any measurable subset $A \subseteq \Sigma^*$ with $\mu(A) = \epsilon$, the probability that none of the $N$ samples land in $A$ is at most* $\exp\left(-\frac{N\epsilon}{2^{H(\mu)}}\right)$, *provided $N$ is sufficiently large. Equivalently, the probability of failing to sample at least*

*one point in* every *region of mass $\epsilon$ is at most* $\exp(-N\epsilon/2^{H(\mu)})$. *Moreover, the largest $\epsilon$ for which this "miss probability" is itself at most $\epsilon$ satisfies* $\epsilon = \frac{2^{H(\mu)}}{N}W\left(\frac{N}{2^{H(\mu)}}\right)$, *where $W(\cdot)$ is the Lambert $W$-function. As $N$ grows large, $\epsilon \approx \frac{2^{H(\mu)}}{N}\ln\left(\frac{N}{2^{H(\mu)}}\right)$, which vanishes at rate on the order of $\ln(N)/N$. This result builds upon foundational principles of information theory and typical set arguments [Cover, 1999].* **Proof is provided in the appendix.**

This theorem guarantees $N$ samples can represent the LLM's implicit uncertainty via the output distribution, thereby providing the first formal bridge linking the sample count ($N$), the distribution's Shannon entropy, and the probabilistic bound on covering the space of possible formal programs.

Our PCFG construction and uncertainty quantification are instance-level. For each question, we generate $N = 100$ SMT programs and induce a question-specific PCFG from that ensemble alone. Metrics from this PCFG (as described in next paragraphs) are then used to predict correctness for that same question. Our method therefore naturally adapts to each reasoning problem's unique characteristics without training.

## 2.1   Probabilistic Context-Free Grammar (PCFG) Derived Metrics

We derive several PCFG metrics to quantify different facets of uncertainty, using established notation (e.g., $R_A$ for the set of rules expanding non-terminal $A$). These metrics can be organized into five conceptual categories that capture complementary aspects of uncertainty in the presence of a formal well-defined grammar for the language: **(1) Static grammar structure metrics** describe the grammar's architectural scale; **(2) Information-theoretic measures** quantify probabilistic uncertainty through entropy and divergence; **(3) Spectral properties** characterize recursive structure and derivation complexity; **(4) Self-consistency measures** assess agreement across multiple LLM generations; and **(5) Ensemble methods** combine signals for robust prediction. We detail each category below.

**(1) Static Metrics for Grammar Structure and Complexity** Basic structural properties of the grammar $G_{SMT}$ provide a foundational understanding of its scale and potential complexity. These include the number of non-terminals ($|V_{SMT}|$) and rules ($|R_{SMT}|$), the average number of rules per non-terminal ($\frac{|R_{SMT}|}{|V_{SMT}|}$), and the average right-hand side (RHS) length ($\frac{1}{|R_{SMT}|}\sum_{A\to\beta\in R_{SMT}}|\beta|$, where $|\beta|$ denotes the length of $\beta$). Further metrics cover the maximum branching factor ($\max_{A\in V_{SMT}}|R_A|$), and the detection of various forms of recursion (e.g., left-recursion $A\to A\gamma$ or right-recursion $A\to\gamma A$). These metrics collectively characterize the grammar's static architecture.

Beyond static metrics, analyzing the set of all rule probabilities $\mathcal{P} = \{p(A\to\beta)\,|\,A\to\beta\in R_{SMT}\}$ we compute descriptive statistics such as mean, median, minimum, maximum, standard deviation ($\sigma(\mathcal{P})$), skewness ($\gamma_1(\mathcal{P})$), and kurtosis ($\gamma_2(\mathcal{P})$). These statistics characterize the shape and spread of the learned probabilities.

**(2) Information-Theoretic Measures** Information theory provides principled ways to measure the uncertainty associated with probabilistic choices within the grammar. The *Shannon Entropy per Non-terminal*, for each $A\in V_{SMT}$, quantifies the average uncertainty (in bits) in selecting a production rule from $R_A$: $H(A) = -\sum_{A\to\beta\in R_A}p(A\to\beta)\log_2 p(A\to\beta)$ A higher $H(A)$ indicates greater uncertainty or variability in the expansions of $A$. The *Rényi Entropy per Non-terminal*, $H_\alpha(A)$, generalizes Shannon entropy and is parameterized by an order $\alpha\geq 0$. For $\alpha\neq 1$: $H_\alpha(A) = \frac{1}{1-\alpha}\log_2\sum_{A\to\beta\in R_A}p(A\to\beta)^\alpha$ Key special cases include Shannon entropy ($H_1(A) = H(A)$ as $\alpha\to 1$), max-entropy ($H_0(A) = \log_2|R_A|$ for $\alpha = 0$, reflecting the number of choices), collision entropy ($H_2(A) = -\log_2\sum_{A\to\beta\in R_A}p(A\to\beta)^2$ for $\alpha = 2$, sensitive to rule choice repetition), and min-entropy ($H_\infty(A) = -\log_2\max_{A\to\beta\in R_A}p(A\to\beta)$ for $\alpha\to\infty$, determined by the most probable rule). Calculating Rényi entropy for different $\alpha$ values (e.g., 0.5, 2) provides a richer characterization of the uncertainty profile than Shannon entropy alone.

The *Overall Grammar Entropy* is typically defined as the weighted average of the Shannon entropies of its non-terminals, $H(A)$, where weights $\pi(A)$ correspond to the stationary distribution or expected frequency of non-terminal $A$ in derivations starting from $S_{SMT}$: $H(G) = \sum_{A\in V_{SMT}}\pi(A)H(A)$. The frequencies $\pi(A)$ can be estimated iteratively. $H(G)$ represents the average uncertainty per derivation step across the entire grammar. *Perplexity*, $PP(G)$, measures how well the PCFG predicts derivations and is the exponentiated grammar entropy: $PP(G) = 2^{H(G)}$. It can be interpreted as the effective average number of choices the grammar presents at each derivation step, weighted by likelihood; lower perplexity indicates a more predictable grammar. The KL divergence, $D_{KL}(p_A||U_A) = \log_2|R_A| - H(A)$, quantifies the inefficiency (in bits) of assuming a uniform rule distribution ($U(A\to\beta) = 1/|R_A|$) for a non-terminal $A$ compared to using

the true PCFG probabilities ($p(A \rightarrow \beta)$). The overall grammar KL divergence from uniform, $D_{KL}(G||U) = \sum_{A \in V_{SMT}} \pi(A) D_{KL}(p_A||U_A)$, quantifies the PCFG's deviation from maximum uncertainty.

Finally, we hypothesize a novel composite metric, $NSUI(G)$, for probabilistic uncertainty and structural complexity. This metric, which ranges from 0 to 1, combines normalized grammar entropy with a factor reflecting the grammar's recursive structure (via its spectral radius $\rho(B)$). It is calculated as $NSUI(G) = E_{ratio} \times S_{factor}$. The entropy ratio $E_{ratio} = H(G)/H_{\max}(G) \in [0,1]$ uses the maximum grammar entropy $H_{\max}(G) = \sum_{A \in V_{SMT}} \pi(A) \log_2 |R_A|$ (assuming uniform rule choices). The spectral factor is $S_{factor} = \rho(B)/(1+\rho(B)) \in [0,1)$. The motivation was to link higher NSUI values to indicate greater probabilistic uncertainty via structural/recursive complexity.

**(3) Spectral Properties** The spectral radius of the grammar's mean matrix (often referred to as the Jacobian matrix in this context), $B \in \mathbb{R}^{|V_{SMT}| \times |V_{SMT}|}$, offers insights into its recursive structure and complexity. An element $B_{ji}$ is the expected number of times non-terminal $A_j$ appears on the right-hand side (RHS) of a production chosen for $A_i$: $B_{ji} = \sum_{A_i \rightarrow \beta \in R_{A_i}} p(A_i \rightarrow \beta) \times \text{count}(A_j, \beta)$, where $\text{count}(A_j, \beta)$ is $A_j$'s occurrences in $\beta$. The spectral radius $\rho(B) = \max\{|\lambda| \mid \det(B - \lambda I) = 0\}$ is $B$'s maximum absolute eigenvalue. Typically, $\rho(B) < 1$ indicates a 'proper' grammar with finite expected derivation lengths, while $\rho(B) \geq 1$ suggests potentially unbounded derivations or higher complexity, contributing to structural uncertainty. This spectral radius is also a key component of the NSUI metric (introduced later in this section).

**(4) Self-Consistency (SC) metrics** quantify agreement across multiple generations from the LLM. Text SC reflects solution consistency (e.g., via majority vote) across multiple LLM textual outputs, while SMT SC measures it (e.g., via SMT solver agreement) across diverse LLM-generated SMT-LIB programs for the same prompt, adapting principles from [Wang et al., 2022].

**(5) Ensemble Predictors** We also implement four distinct ensemble predictors for enhanced uncertainty quantification. These are: (1) Ensemble Simple, an unweighted average of a key subset of metrics; (2) Ensemble Average, a comprehensive unweighted average of all metric scores; (3) Ensemble Weighted, where individual metric contributions are varied based on validation performance or theoretical importance; and (4) Ensemble ML, a meta-machine learning model (e.g., logistic regression) trained on the vector of metric scores to predict errors. This approach aims to improve overall predictive accuracy, calibration, and robustness by combining varied uncertainty signals.

# 3 Results

We have evaluated five frontier LLMs, namely o3-mini, DeepSeekR1 (with CoT enabled [Wei et al., 2022b]), DeepSeek-v3-04-21, Gemini Flash 2.0 & Lite (non-reasoning), on four datasets which are widely adopted reasoning tasks; StrategyQA [Geva et al., 2021], ProntoQA [Saparov and He, 2023], ProofWriter[Tafjord et al., 2021] and FOLIO [Han et al., 2024]. Our experimental setup involves two distinct sampling configurations: For the initial benchmarking comparison of Text versus SMT reasoning approaches (Tab. 1), we generate $N = 5$ samples per question. For our core uncertainty quantification analysis (Tables 4 and 5), we generate a much larger corpus of $N = 100$ SMT-LIB program samples per question to induce statistically meaningful PCFGs for each reasoning instance (as detailed in Appendix A.2). From these samples, answers were derived via: 1) Text: direct LLM output (intrinsic reasoning over text); 2) SMT: LLM-generated SMT-LIB solved by Z3 (autoformalization). Notably, SMT-LIB generation required significantly less effort (more syntactically valid programs in fewer attempts) and used dramatically fewer tokens per prompt compared to [Ganguly et al., 2024], while also offering multi-solver interoperability.

## 3.1 Benchmarking SMT based Formal Reasoning

Table 1: Benchmarking accuracy of frontier LLMs using direct text output (Text) versus SMT-LIB generation solved by Z3 (SMT). No approach universally outperforms the other across all models and datasets. Finer grained results are available in the supplementary material.

| ACCURACY | StrategyQA | | | ProntoQA | | | ProofWriter | | | FOLIO | | |
|---|---|---|---|---|---|---|---|---|---|---|---|---|
| | Text | SMT | $\delta$ | Text | SMT | $\delta$ | Text | SMT | $\delta$ | Text | SMT | $\delta$ |
| **o3-mini** | 0.7828 | 0.7980 | -0.0152 | 1.0000 | 0.9980 | 0.0020 | 0.8893 | 0.9418 | -0.0524 | 0.9450 | 0.5000 | 0.4450 |
| **DeepSeekv3** | 0.8292 | 0.6720 | 0.1572 | 1.0000 | 0.4501 | 0.5499 | 0.8057 | 0.5800 | 0.2257 | 0.9333 | 0.5961 | 0.3372 |
| **DeepSeek R1** | 0.8580 | 0.7760 | 0.0820 | 0.9939 | 0.7440 | 0.2499 | 0.9423 | 0.4935 | 0.4488 | 0.9252 | 0.5200 | 0.4052 |
| **Flash 2.0** | 0.7188 | 0.5360 | 0.1828 | 0.9820 | 0.9000 | 0.0820 | 0.4900 | 0.6660 | -0.1760 | 0.9010 | 0.5625 | 0.3385 |
| **Flash 2.0 Lite** | 0.6760 | 0.4500 | 0.2260 | 0.9980 | 0.9980 | 0.0000 | 0.4060 | 0.7540 | -0.3480 | 0.9017 | 0.7321 | 0.1696 |

On ProofWriter, a task closely aligned with symbolic logic, SMT-based methods yielded substantial improvements for three models, particularly benefiting those that struggle with direct formal reasoning. Conversely, on ProntoQA and FOLIO, direct textual reasoning consistently outperformed SMT across most models, suggesting that for these QA tasks, the overhead introduced during autoformalization outweighs potential benefits. StrategyQA showed mixed results, with o3-mini slightly benefiting from SMT while other models performed better with direct reasoning.

The SMT approach systematically alters error profiles compared to direct reasoning, often trading precision for recall. For struggling models, autoformalization often increases recall at the cost of precision, yielding more formal answers but also more false positives, a tendency consistent with LLMs' documented proclivity toward proving satisfiability in [Ganguly et al., 2024]. On ProofWriter, where SMT generally helped, performance gains stemmed from simultaneous improvements in both precision and recall, indicating the approach successfully addressed fundamental reasoning errors. Conversely, on datasets where direct reasoning excelled, SMT's underperformance typically manifested as reduced recall, suggesting failures in the formalization process resulted in missed correct answers.

Our results reveal predominantly epistemic uncertainty in both reasoning approaches. Direct reasoning fails through knowledge gaps and procedural errors, while SMT introduces formalization errors when translating to formal specifications. This explains task-dependent performance: SMT benefits tasks with explicit premises (ProofWriter) by isolating deductive reasoning, while knowledge-intensive tasks (StrategyQA) expose formalization bottlenecks. These findings highlight the critical need for Uncertainty Quantification on LLM-generated formal artifacts to prevent upstream formalization errors from propagating through otherwise sound solvers.

## 3.2 Benchmarking Uncertainty Quantification Techniques

**Experiment Setup** To evaluate uncertainty quantification (UQ) methods for identifying prediction errors, we examine several facets: *Error Discrimination* utilizes the Area Under the Receiver Operating Characteristic Curve (AUROC) to assess if uncertainty scores distinguish correct from incorrect predictions; a higher AUROC signifies better uncertainty-error alignment. *Selective Prediction Utility* employs the Area Under the Risk-Coverage Curve (AURC) to measure practical risk mitigation via abstention (including analysis of optimal abstention percentages, associated error rates, and relative error reduction); lower AURC indicates effective error identification, improving performance on retained samples. Finally, *Calibration Assessment* evaluates the probabilistic reliability of confidence scores using metrics like Expected Calibration Error (ECE), Reliability Diagrams, and the Brier Score; lower ECE and Brier scores denote better calibration, where predicted confidence accurately reflects empirical correctness rates.

Table 2: **Token-level baseline uncertainty quantification techniques** and their results at detecting autoformalization errors w.r.t. ground truth for DeepSeek-v3-0324 across reasoning datasets. While conventional metrics (AUROC, ECE, Brier) show moderate performance, they inadequately capture the distinct epistemic uncertainties in formalization versus reasoning processes. Notably, no UQ method consistently excels across tasks . The uncertainty-aware abstention metrics reflect how the model can selectively answer questions by applying an optimal uncertainty threshold (Opt.Thresh) that minimizes error rate (Err@T) and maximizes error reduction (RelErrRed) compared to answering all questions.This suggests the need for specialized UQ approaches that explicitly model the distribution of formal artifacts. DeepSeekv3 is the only model we examined that provides token logprobs.

|  |  | AUROC | ECE | Brier | AURC | Opt.T | Err@T | RelErrRed |
|---|---|---|---|---|---|---|---|---|
| StrategyQA | Entropy | 0.5872 | 0.1415 | 0.2433 | 0.2399 | 0.0500 | 0.3221 | 0.0180 |
|  | Perplexity | 0.6179 | 0.0802 | 0.2218 | 0.2218 | 0.5000 | 0.2520 | 0.2317 |
|  | Kurtosis | **0.6227** | 0.3038 | 0.3075 | 0.2236 | 0.5000 | 0.2440 | 0.2561 |
| ProntoQA | Entropy | 0.5622 | 0.1395 | 0.2685 | 0.4585 | 0.0500 | 0.5408 | 0.0166 |
|  | Perplexity | **0.6118** | 0.1009 | 0.2522 | 0.4218 | 0.0500 | 0.5365 | 0.0244 |
|  | Kurtosis | 0.6078 | 0.2390 | 0.2990 | 0.4265 | 0.2000 | 0.5102 | 0.0722 |
| ProofWriter | Entropy | 0.5165 | 0.1666 | 0.2761 | 0.3938 | 0.0500 | 0.4105 | 0.0226 |
|  | Perplexity | **0.5893** | 0.2430 | 0.3021 | 0.3214 | 0.4000 | 0.3633 | 0.1349 |
|  | Kurtosis | 0.5656 | 0.1657 | 0.2705 | 0.3322 | 0.4500 | 0.3564 | 0.1515 |
| FOLIO | Entropy | **0.7001** | 0.2101 | 0.2465 | 0.4737 | 0.5000 | 0.4767 | 0.2679 |
|  | Perplexity | 0.5609 | 0.2385 | 0.2926 | 0.5514 | 0.0500 | 0.6380 | 0.0202 |
|  | Kurtosis | 0.5548 | 0.1761 | 0.2428 | 0.5560 | 0.0500 | 0.6319 | 0.0296 |

We benchmarked our syntactic PCFG approach against semantic entropy and clustering [Kuhn et al., 2023, Farquhar et al., 2024] using entailment checking with independent prompts to DeepSeek-v3-0324, comparing each generated SMT program with the original question as context to determine semantic equivalence classes. Table 3 presents results across all four datasets. We evaluated: (1) Discrete Entropy over semantic clusters, (2) Continuous Entropy using Monte Carlo sampling (MC), (3) Continuous Entropy using Rao-Blackwellized estimator (RAO), and (4) the raw number of semantic clusters as an uncertainty signal. The semantic methods failed to provide superior uncertainty signals, yielding near-random performance (AUROC $\approx$ 0.5-0.6) and no practical error reduction (poor AURC). Computationally, semantic clustering is inefficient, requiring $O(M^2)$ undecidable comparisons, whereas our PCFG method is efficient. We also confirmed that standard NLI models (DeBERTa) fail completely, as they cannot parse formal SMT artifacts.

Table 3: **Baseline semantic entropy based uncertainty quantification techniques** for detecting auto-formalization errors with respect to ground truth using DeepSeek-v3-0324. While moderate performance is observed in limited configurations, semantic uncertainty does not distinctly capture epistemic uncertainties in autoformalization. The artificially high AUROC on FOLIO reflects disproportionate error rates rather than effective uncertainty quantification, as evidenced by high AURC values indicating poor risk-coverage tradeoffs.

| Dataset | Metric | AUROC | ECE | Brier | AURC | Opt.T | Err@T | RelErrRed |
|---------|--------|-------|-----|-------|------|-------|-------|-----------|
| StrategyQA | Discrete Entropy | 0.5341 | 0.2345 | 0.3517 | 0.2724 | 0.0500 | 0.3263 | 0.0051 |
| | Continuous Entropy MC | 0.5502 | 0.2050 | 0.2669 | 0.2555 | 0.0000 | 0.3280 | 0.0000 |
| | Continuous Entropy RAO | **0.5801** | 0.2744 | 0.2987 | 0.2432 | 0.0000 | 0.3280 | 0.0000 |
| | Num. semantic clusters | 0.5331 | 0.2160 | 0.3410 | 0.2681 | 0.0500 | 0.3263 | 0.0051 |
| ProntoQA | Discrete Entropy | **0.6273** | 0.0788 | 0.2955 | 0.4453 | 0.0000 | 0.5507 | 0.0000 |
| | Continuous Entropy MC | 0.5015 | 0.2900 | 0.3459 | 0.4910 | 0.1000 | 0.5461 | 0.0084 |
| | Continuous Entropy RAO | 0.5779 | 0.4340 | 0.4515 | 0.4609 | 0.0500 | 0.5480 | 0.0049 |
| | Num. semantic clusters | 0.5953 | 0.1258 | 0.3195 | 0.4585 | 0.1000 | 0.5507 | 0.0001 |
| ProofWriter | Discrete Entropy | 0.5717 | 0.2256 | 0.3316 | 0.3254 | 0.0000 | 0.4200 | 0.0000 |
| | Continuous Entropy MC | 0.5127 | 0.2340 | 0.3091 | 0.3611 | 0.3500 | 0.4154 | 0.0110 |
| | Continuous Entropy RAO | **0.5966** | 0.3053 | 0.3469 | 0.3178 | 0.0000 | 0.4200 | 0.0000 |
| | Num. semantic clusters | 0.5645 | 0.1775 | 0.3251 | 0.3322 | 0.0000 | 0.4200 | 0.0000 |
| FOLIO | Discrete Entropy | 0.7542 | 0.2508 | 0.2541 | 0.4506 | 0.0500 | 0.6584 | 0.0007 |
| | Continuous Entropy MC | 0.6811 | 0.4571 | 0.4186 | 0.4790 | 0.0000 | 0.6588 | 0.0000 |
| | Continuous Entropy RAO | **0.7564** | 0.5383 | 0.5194 | 0.4364 | 0.0000 | 0.6588 | 0.0000 |
| | Num. semantic clusters | 0.7523 | 0.3132 | 0.2886 | 0.4496 | 0.0000 | 0.6588 | 0.0000 |

**Evaluation tasks:** We expanded and evaluated our uncertainty metrics in two distinct prediction tasks: (1) whether LLM-generated SMT programs, when executed by Z3, would yield the correct ground truth answer (as prior baselines), and (2) whether the SMT output would be consistent with the model's own natural language reasoning.

Experiments for model and dataset combinations were chosen where a performance gap was observed, with N=100 samples, thereby making UQ analysis meaningful, but where the SMT performance was not close to random guessing. Our argument here is that because we are relying on information within the FV artifacts, and those artifacts are not well-calibrated for the task (i.e., operating at the level of random guessing), we cannot extract information about failure from them. We also employed temperature sampling with values uniformly distributed between 0.1 and 2.0 to ensure diverse exploration of the LLM's output distribution

**Task-Dependent Signal Dominance in SMT vs Ground Truth Prediction:**

**Knowledge-Intensive Reasoning:** For StrategyQA, cross-modal agreement metrics consistently dominated. O3-mini showed strong performance with grammar entropy (AUROC=0.7448, AURC=0.1113) and text consistency (AUROC=0.7369, AURC=0.1081). For DeepSeek-R1, text consistency substantially outperformed all pure PCFG metrics (AUROC=0.7835, AURC=0.0983). This indicates epistemic uncertainty in world knowledge as the primary correctness bottleneck as these cross-modal metrics effectively gauge if the SMT formalization aligns with the LLM's initial (potentially flawed) semantic interpretation.

**Premise-Explicit Reasoning:** For ProofWriter, PCFG-derived metrics demonstrated exceptional discriminative power for ground truth prediction. O3-mini achieved near-perfect performance with grammar entropy (AUROC=0.9301, AURC=0.0008) and perplexity (AUROC=0.9194, AURC=0.0008). This confirms procedural epistemic uncertainty dominates in formal reasoning tasks, where an LLM's primary challenge shifts from knowledge recall to the correct application of formal rules. Thus, PCFG metrics assessing structural variance in the SMT-LIB output can identify such deductive missteps with

high precision—o3-mini's AURC of 0.0008 using grammar entropy, for instance, enables filtering nearly all errors by abstaining on a minute fraction of outputs.

Table 4: Uncertainty quantification metrics for **predicting ground truth correctness** via PCFGs of LLM-generated SMT programs. Results show AUROC, ECE, Brier, and AURC across models and reasoning tasks, with ensemble methods consistently outperforming individual metrics. Color intensity indicates performance strength (darker green = better)

| | StrategyQA | | | | | | | | ProofWriter | | | | | | | |
| | o3-mini | | | | DeepSeek-v3-0324 | | | | o3-mini | | | | Gemini 2.0 Flash Lite | | | |
| Metric | AUROC | ECE | Brier | AURC | AUROC | ECE | Brier | AURC | AUROC | ECE | Brier | AURC | AUROC | ECE | Brier | AURC |
|---|---|---|---|---|---|---|---|---|---|---|---|---|---|---|---|---|
| Grammar Entropy | 0.7448 | 0.3058 | 0.2340 | 0.1113 | 0.7087 | 0.1575 | 0.2302 | 0.2097 | 0.9301 | 0.4419 | 0.2500 | 0.0008 | 0.5380 | 0.3185 | 0.2869 | 0.1405 |
| Perplexity | 0.5589 | 0.3107 | 0.2862 | 0.1811 | 0.6122 | 0.1601 | 0.2641 | 0.2497 | 0.9194 | 0.5358 | 0.3515 | 0.0074 | 0.5934 | 0.3888 | 0.3182 | 0.1267 |
| KL Divergence | 0.6428 | 0.2485 | 0.2385 | 0.1471 | 0.5723 | 0.1393 | 0.2322 | 0.2878 | 0.5108 | 0.5167 | 0.3260 | 0.0074 | 0.5164 | 0.3080 | 0.2797 | 0.1573 |
| NSUI | 0.6334 | 0.2436 | 0.2539 | 0.1250 | 0.5997 | 0.0781 | 0.2191 | 0.2672 | 0.5645 | 0.5710 | 0.3843 | 0.0084 | 0.5243 | 0.3186 | 0.2642 | 0.1514 |
| Renyi Ent (2) | 0.5175 | 0.3303 | 0.2997 | 0.1977 | 0.6195 | 0.1622 | 0.2679 | 0.2429 | 0.8871 | 0.5405 | 0.3598 | 0.0013 | 0.5996 | 0.4102 | 0.3368 | 0.1285 |
| Renyi Ent (0.5) | 0.5973 | 0.3398 | 0.3042 | 0.1634 | 0.6126 | 0.1623 | 0.2626 | 0.2517 | 0.9301 | 0.4724 | 0.2879 | 0.0008 | 0.5933 | 0.4401 | 0.3581 | 0.1258 |
| Max Ent | 0.6649 | 0.3553 | 0.2935 | 0.1297 | 0.6851 | 0.0473 | 0.2099 | 0.2271 | 0.9086 | 0.7198 | 0.5550 | 0.0013 | 0.5417 | 0.3503 | 0.3045 | 0.1420 |
| Ent Ratio | 0.5385 | 0.3283 | 0.3028 | 0.1834 | 0.5311 | 0.1336 | 0.2538 | 0.3306 | 0.5860 | 0.5714 | 0.3764 | 0.0055 | 0.5177 | 0.3943 | 0.3426 | 0.1548 |
| Spectral Factor | 0.6334 | 0.2173 | 0.2364 | 0.1319 | 0.6800 | 0.0992 | 0.2236 | 0.2365 | 0.7473 | 0.3458 | 0.2247 | 0.0032 | 0.5011 | 0.5048 | 0.4157 | 0.1578 |
| Spectral Radius | 0.6334 | 0.2892 | 0.2747 | 0.1319 | 0.6800 | 0.0686 | 0.2148 | 0.2365 | 0.7473 | 0.3545 | 0.2305 | 0.0032 | 0.5011 | 0.3930 | 0.3172 | 0.1578 |
| # Nonterminals | 0.5111 | 0.3540 | 0.3188 | 0.2006 | 0.6115 | 0.1329 | 0.2469 | 0.2547 | 0.8011 | 0.4267 | 0.2397 | 0.0019 | 0.5167 | 0.4838 | 0.4215 | 0.1672 |
| # Rules | 0.5548 | 0.2117 | 0.2385 | 0.1855 | 0.6197 | 0.1109 | 0.2252 | 0.2583 | 0.5108 | 0.4186 | 0.2201 | 0.0084 | 0.5549 | 0.2422 | 0.2315 | 0.1370 |
| Avg Rules / NT | 0.5737 | 0.2400 | 0.2415 | 0.1752 | 0.6021 | 0.0902 | 0.2260 | 0.2616 | 0.9301 | 0.5393 | 0.3499 | 0.0008 | 0.5840 | 0.2790 | 0.2656 | 0.1301 |
| Avg RHS Len | 0.5350 | 0.6141 | 0.5651 | 0.1979 | 0.5122 | 0.1753 | 0.2712 | 0.3279 | 0.8011 | 0.6535 | 0.5086 | 0.0026 | 0.5631 | 0.3413 | 0.2906 | 0.1480 |
| Max Branch Factor | 0.5181 | 0.1500 | 0.1997 | 0.1990 | 0.6180 | 0.1450 | 0.2270 | 0.2688 | 0.5914 | 0.2979 | 0.1377 | 0.0055 | 0.5745 | 0.2189 | 0.2293 | 0.1355 |
| Rule Dist Mean | 0.5740 | 0.3161 | 0.2836 | 0.1752 | 0.6021 | 0.1811 | 0.2534 | 0.2616 | 0.9301 | 0.4945 | 0.3034 | 0.0008 | 0.5838 | 0.4368 | 0.3713 | 0.1301 |
| Rule Dist StdDev | 0.5291 | 0.3995 | 0.3517 | 0.1811 | 0.5281 | 0.1251 | 0.2573 | 0.3116 | 0.5108 | 0.5555 | 0.3723 | 0.0074 | 0.5144 | 0.4474 | 0.3795 | 0.1559 |
| Rule Dist Skew | 0.5833 | 0.3178 | 0.2850 | 0.1689 | 0.6036 | 0.1431 | 0.2489 | 0.2610 | 0.9301 | 0.4923 | 0.2987 | 0.0008 | 0.5844 | 0.4511 | 0.3779 | 0.1313 |
| Rule Dist Kurtosis | 0.5659 | 0.3948 | 0.3420 | 0.1785 | 0.5787 | 0.1720 | 0.2600 | 0.2754 | 0.5860 | 0.5107 | 0.3115 | 0.0055 | 0.5044 | 0.1437 | 0.1913 | 0.1726 |
| Self Consistency Text | 0.7369 | 0.1604 | 0.1603 | 0.1081 | 0.6017 | 0.2874 | 0.3048 | 0.2882 | 0.8990 | 0.0423 | 0.0280 | 0.0020 | 0.5525 | 0.2283 | 0.2419 | 0.1376 |
| Self Consistency SMT | 0.7416 | 0.1523 | 0.1609 | 0.1051 | 0.6203 | 0.2318 | 0.2745 | 0.2513 | 0.7121 | 0.8501 | 0.7764 | 0.0025 | 0.7364 | 0.3535 | 0.2751 | 0.1031 |
| Ensemble Average | 0.7622 | 0.3724 | 0.2916 | 0.1103 | 0.6795 | 0.1214 | 0.2077 | 0.2182 | 0.9949 | 0.3356 | 0.1414 | 0.0005 | 0.6140 | 0.3922 | 0.3192 | 0.1240 |
| Ensemble Weighted | 0.7657 | 0.1738 | 0.1617 | 0.1099 | 0.7211 | 0.1257 | 0.2135 | 0.1989 | 0.9785 | 0.4612 | 0.2566 | 0.0003 | 0.7235 | 0.3327 | 0.2539 | 0.1035 |
| Ensemble ML | 0.7850 | 0.2090 | 0.1756 | 0.1013 | 0.7709 | 0.0877 | 0.1968 | 0.1847 | 0.9892 | 0.0572 | 0.0280 | 0.0003 | 0.7631 | 0.2897 | 0.2229 | 0.0823 |
| Ensemble Simple | 0.6702 | 0.2055 | 0.2104 | 0.1410 | 0.6401 | 0.1763 | 0.2514 | 0.2483 | 0.9355 | 0.4419 | 0.2582 | 0.0008 | 0.6476 | 0.3867 | 0.3039 | 0.1071 |

Table 5: Uncertainty metrics based on PCFGs for **predicting consistency between LLM's SMT formalization and its natural language reasoning.** Task-specific uncertainty patterns emerge: rule distribution kurtosis dominates for StrategyQA (AUROC=0.8695), while different metrics excel for each model-task combination, highlighting the multifaceted nature of formalization uncertainty.

| | StrategyQA | | | | | | | | | | | | ProofWriter | | | |
| | DeepSeek R1 | | | | DeepSeekv3-0324 | | | | Gemini 2.0 Flash Lite | | | | o3-mini | | | |
| Metric | AUROC | ECE | Brier | AURC | AUROC | ECE | Brier | AURC | AUROC | ECE | Brier | AURC | AUROC | ECE | Brier | AURC |
|---|---|---|---|---|---|---|---|---|---|---|---|---|---|---|---|---|
| **Grammar Entropy** | 0.7609 | 0.4617 | 0.2968 | 0.0216 | 0.7354 | 0.3058 | 0.2551 | 0.0970 | 0.6622 | 0.1277 | 0.2095 | 0.5689 | 0.8602 | 0.4419 | 0.2542 | 0.0013 |
| Perplexity | 0.6211 | 0.3789 | 0.2640 | 0.0361 | 0.6721 | 0.3282 | 0.2718 | 0.1205 | 0.7212 | 0.3255 | 0.2849 | 0.5273 | 0.9032 | 0.4429 | 0.2616 | 0.0013 |
| KL Divergence | 0.5776 | 0.4408 | 0.2855 | 0.0443 | 0.5335 | 0.1195 | 0.1780 | 0.1633 | 0.5486 | 0.2387 | 0.2630 | 0.6341 | 0.6667 | 0.5167 | 0.3238 | 0.0039 |
| NSUI | 0.5963 | 0.2529 | 0.1433 | 0.0462 | 0.5973 | 0.1768 | 0.1919 | 0.1411 | 0.6741 | 0.1195 | 0.1624 | 0.5221 | 0.9355 | 0.4077 | 0.2172 | 0.0008 |
| Renyi Ent (2) | 0.6242 | 0.3980 | 0.2746 | 0.0378 | 0.6799 | 0.3403 | 0.2808 | 0.1160 | 0.7624 | 0.3192 | 0.2624 | 0.5037 | 0.9032 | 0.4382 | 0.2580 | 0.0013 |
| Renyi Ent (0.5) | 0.6149 | 0.3942 | 0.2755 | 0.0376 | 0.6667 | 0.3333 | 0.2751 | 0.1223 | 0.6994 | 0.2766 | 0.2619 | 0.5438 | 0.8925 | 0.5063 | 0.3246 | 0.0013 |
| Max Ent | 0.7174 | 0.3994 | 0.2341 | 0.0262 | 0.6353 | 0.1309 | 0.1738 | 0.1247 | 0.5982 | 0.3401 | 0.3083 | 0.5900 | 0.9355 | 0.2589 | 0.1029 | 0.0008 |
| Ent Ratio | 0.5217 | 0.5047 | 0.3529 | 0.0543 | 0.6154 | 0.4091 | 0.3307 | 0.1446 | 0.5511 | 0.1641 | 0.2511 | 0.6346 | 0.9677 | 0.4074 | 0.2082 | 0.0003 |
| Spectral Factor | 0.5481 | 0.1533 | 0.0927 | 0.0640 | 0.7034 | 0.1254 | 0.1580 | 0.1206 | 0.6538 | 0.0943 | 0.1677 | 0.5316 | 0.5269 | 0.6329 | 0.5061 | 0.0084 |
| Spectral Radius | 0.5481 | 0.2067 | 0.1166 | 0.0640 | 0.7034 | 0.1896 | 0.1752 | 0.1206 | 0.6538 | 0.1648 | 0.1703 | 0.5316 | 0.5269 | 0.6243 | 0.4953 | 0.0084 |
| # Nonterminals | 0.5264 | 0.5679 | 0.4237 | 0.0557 | 0.5549 | 0.2271 | 0.2455 | 0.1480 | 0.5166 | 0.4180 | 0.3958 | 0.6509 | 0.6505 | 0.5520 | 0.3650 | 0.0047 |
| # Rules | 0.6087 | 0.3083 | 0.1839 | 0.0396 | 0.5675 | 0.2025 | 0.2077 | 0.1485 | 0.5749 | 0.4256 | 0.3818 | 0.6252 | 0.5108 | 0.4186 | 0.2201 | 0.0064 |
| Avg Rules / NT | 0.7034 | 0.4068 | 0.2532 | 0.0273 | 0.6034 | 0.1393 | 0.1854 | 0.1399 | 0.6565 | 0.2913 | 0.2761 | 0.5923 | 0.8011 | 0.4394 | 0.2554 | 0.0026 |
| Avg RHS Len | 0.5637 | 0.2491 | 0.1566 | 0.0544 | 0.5208 | 0.1254 | 0.1818 | 0.1972 | 0.6014 | 0.4855 | 0.4420 | 0.6098 | 0.5054 | 0.6535 | 0.5116 | 0.0074 |
| Max Branch Factor | 0.6801 | 0.3325 | 0.2042 | 0.0324 | 0.5937 | 0.1563 | 0.1920 | 0.1457 | 0.5818 | 0.5167 | 0.4747 | 0.6313 | 0.7419 | 0.6809 | 0.5100 | 0.0032 |
| Rule Dist Mean | 0.7034 | 0.5352 | 0.3733 | 0.0273 | 0.6034 | 0.3210 | 0.2742 | 0.1399 | 0.6565 | 0.2197 | 0.2280 | 0.5923 | 0.8011 | 0.4842 | 0.2971 | 0.0026 |
| Rule Dist StdDev | 0.6056 | 0.6755 | 0.5361 | 0.0413 | 0.6281 | 0.2226 | 0.2221 | 0.1445 | 0.7183 | 0.3136 | 0.2807 | 0.5455 | 0.8710 | 0.4232 | 0.2392 | 0.0013 |
| Rule Dist Skew | 0.6848 | 0.4800 | 0.3260 | 0.0309 | 0.5986 | 0.3057 | 0.2619 | 0.1406 | 0.6796 | 0.1783 | 0.2199 | 0.5720 | 0.8172 | 0.4864 | 0.2964 | 0.0019 |
| Rule Dist Kurtosis | 0.5311 | 0.2521 | 0.1588 | 0.0534 | 0.6600 | 0.0731 | 0.1576 | 0.1256 | 0.8695 | 0.3187 | 0.2412 | 0.1256 | 0.9462 | 0.5107 | 0.3056 | 0.0008 |
| Self Consistency Text | 0.8245 | 0.1002 | 0.0751 | 0.0155 | 0.5778 | 0.1874 | 0.2008 | 0.1754 | 0.5350 | 0.2788 | 0.2616 | 0.6064 | 0.7050 | 0.9352 | 0.9023 | 0.0030 |
| Self Consistency SMT | 0.7570 | 0.2062 | 0.1373 | 0.0268 | 0.7116 | 0.1662 | 0.1909 | 0.1054 | 0.7505 | 0.5380 | 0.4357 | 0.4822 | 0.7100 | 0.8600 | 0.7863 | 0.0030 |
| Ensemble Average | 0.8183 | 0.4695 | 0.3058 | 0.0180 | 0.7064 | 0.1876 | 0.1831 | 0.0983 | 0.7927 | 0.1531 | 0.1702 | 0.4848 | 0.9300 | 0.3560 | 0.1741 | 0.0007 |
| **Ensemble Weighted** | 0.8494 | 0.3256 | 0.1711 | 0.0155 | 0.7709 | 0.2340 | 0.1971 | 0.0798 | 0.8070 | 0.1673 | 0.1632 | 0.4718 | 1.0000 | 0.4231 | 0.2375 | 0.0003 |
| **Ensemble ML** | 0.8245 | 0.3084 | 0.2003 | 0.0170 | 0.8517 | 0.1861 | 0.1703 | 0.0573 | 0.7946 | 0.1308 | 0.1592 | 0.4784 | 1.0000 | 0.0496 | 0.0199 | 0.0003 |
| Ensemble Simple | 0.6957 | 0.4363 | 0.2748 | 0.0316 | 0.6727 | 0.3021 | 0.2567 | 0.1119 | 0.7584 | 0.0796 | 0.1568 | 0.4929 | 0.6667 | 0.4419 | 0.2661 | 0.0039 |

**Predicting SMT-Text Consistency:**

**Arithmetic Reasoning:** On ProntoQA with Gemini Flash 2.0, SMT consistency achieved remarkable performance in predicting text-SMT alignment (AUROC=0.9291, AURC=0.0084), while spectral radius (AUROC=0.6425, AURC=0.0379) emerged as the only effective structural metric. This isolates recursive complexity as a distinct source of uncertainty in arithmetic formalization, as excessively convoluted SMT structures, indicated by a high spectral radius, risk diverging from the model's more direct textual reasoning on numerical problems.

**Model-Specific Patterns:** For Gemini Flash 2.0 Lite on StrategyQA, kurtosis of the rule distribution was the strongest predictor of SMT-Text consistency (AUROC=0.8695). Analysis revealed a distinctive "switching" behavior between minimal and verbose SMT patterns, producing a bimodal distribution with heavy tails—a novel diagnostic for capacity limitations in formalization, whereby such stylistic oscillations between overly terse or verbose SMT, captured by kurtosis, make the resulting formal artifact more prone to misalign with the intended textual meaning.

**Ablation Study:** PCFG spectral radius from LLM-generated SMT-LIB programs consistently decreases with sampling temperature (In App B), as broader exploration diversifies rule selections, reducing fixation on recursive productions that heavily influence moment matrix eigenvalues. Probability mass spreads more uniformly across production alternatives, diminishing single recursive pattern dominance and thus lowering the mean matrix's maximum absolute eigenvalue. Notably, grammatical properties lack sharp phase transitions across temperature ranges; derived PCFGs show smooth, monotonic changes in spectral and information-theoretic characteristics, implying a continuous, rather than abrupt, generative response to temperature. Non-terminal expansion distributions shift from concentrated to broader with increasing temperature, though this plateaus, indicating finite exploration capacity, possibly constrained by inherent model biases or the grammar's finite structure. The striking consistency of these spectral-temperature curves across diverse LLMs points to a fundamental, universal mechanism by which these models navigate the coherence-diversity trade-off when generating structured formal languages. Finally, our fine-grained localized entropy within PCFG production rules surpasses global or non-grammatical standard techniques in error prediction, confirming that granular structural uncertainty in specific grammatical constructs directly flags component-level semantic error likelihood, offering more precise diagnostics.

## 4 Discussion

Our analysis reveals a fundamental insight: the syntactic atypicality (e.g., in PCFG rule entropy or usage kurtosis) of LLM-generated formal artifacts serves as a powerful signal for semantic errors, reminiscent of OOD detection [Ganguly et al., 2025]. When LLMs correctly understand logical relationships, they consistently produce high-probability rule sequences, whereas semantic misunderstandings manifest as statistical anomalies—creating distinctive "syntactic fingerprints" of reasoning failure that enable our exceptional error detection (AUROC=0.9301 on ProofWriter). This typicality-based approach transcends architectures, its PCFG metric rankings consistently capturing intrinsic difficulties like formalizing ambiguous language across diverse models.

However, the relationship between typicality and correctness isn't straightforward; metrics with superior discriminative ability often exhibit poor calibration (ECE=0.4419), indicating anomaly magnitude doesn't linearly predict error probability—necessitating calibration-aware fusion, perhaps by integrating consistency signals. Even more revealing is asymmetric self-consistency (e.g., Gemini/ProntoQA: SMT AUROC=0.9291 vs. text AUROC=0.5108), suggesting LLMs may use distinct, imperfectly aligned formal versus textual reasoning pathways, not just translate a unified process. Such insights shift neurosymbolic design from translation-focus to pathway-alignment and grounding, e.g., via joint training, as SMT syntactic typicality alone is insufficient if its pathway misaligns with textual reasoning.

## 5 Related Works

**Formal Reasoning with LLMs** LLMs show proficiency in formal reasoning [Welleck et al., 2022a, Chen et al., 2022], but face challenges including hallucination, uncertainty expression [Lin et al., 2022a], self-verification [Hou et al., 2023], and reasoning opacity [Wei et al., 2022b]. Hybrid approaches combine LLMs with formal tools but often overlook model uncertainty. For autoformalization, early sequence-to-sequence models [Wang et al., 2018, 2020] evolved into LLM-based approaches [Wu et al., 2022, Agrawal et al., 2022, Gadgil et al., 2022, Murphy et al., 2024], with structured methods [Jiang et al., 2023b, Zhao et al., 2024] combining LLMs with ATPs, and various applications [Liu et al., 2023, Pan et al., 2023, Olausson et al., 2023, Ye et al., 2023, Zhou et al., 2024, Huang et al., 2024a, Xin et al., 2024a, Jiang et al., 2024, Quan et al., 2024, Xin et al., 2024b]. Proofstep generation advanced from classification [Whalen, 2016, Huang et al., 2019, Bansal et al., 2019] to language modeling [Polu and Sutskever, 2020, First et al., 2023, Wang et al., 2024, Welleck et al., 2022b, Jiang et al., 2022], with recent work exploring zero-shot capabilities [Zhang et al., 2023, Yousefzadeh and Cao, 2023, Scheidt, 2023, Frieder et al., 2023a,b,c, Zhang et al., 2024a] and formal proof generation [Zheng et al., 2024, Xin et al., 2024a, Huang et al., 2024a, Thakur et al., 2024]. Proof search strategies include supervised learning [Loos et al., 2017, Chvalovskỳ et al., 2019], reinforcement

learning [Kusumoto et al., 2018, Crouse et al., 2021, Piepenbrock et al., 2021], MCTS [Wu et al., 2021, Lample et al., 2022, Wang et al., 2023a], and language-agent methods [Thakur et al., 2024, An et al., 2024].

**Uncertainty in LLM Reasoning** Research explores various uncertainty estimation approaches in language models: information-theoretic methods using entropy [Kadavath et al., 2022, Kuhn et al., 2023, Duan et al., 2024], perplexity [Mora-Cross and Calderon-Ramirez, 2024, Margatina et al., 2023], and mutual information [Malinin, 2019, Wimmer et al., 2023, Depeweg, 2019, Ash, 1965]; ensemble strategies like MC Dropout [Srivastava et al., 2014, Gal and Ghahramani, 2016a, Lakshminarayanan et al., 2017], Deep Ensembles [Fadeeva et al., 2023, Lakshminarayanan et al., 2017], and BatchEnsemble [Gal and Ghahramani, 2016b, Lakshminarayanan et al., 2017, Wen et al., 2020] for hallucination detection [Arteaga et al., 2024]; consistency techniques evaluating output agreement [Wang et al., 2023b, Cole et al., 2023, Huang et al., 2024b, Zhang et al., 2024b, Lakshminarayanan et al., 2017, Gawlikowski et al., 2023, Manakul et al., 2023, Chen and Mueller, 2024]; similarity-based methods [Lin et al., 2024]; Bayesian approaches including BNNs [Shridhar et al., 2019, Blundell et al., 2015], variational inference [Graves, 2011, Jordan et al., 1999, Kullback and Leibler, 1951], Gaussian processes [Iwata and Ghahramani, 2017, Liu et al., 2020], and MCMC [Xiao and Wang, 2018]; and language-based methods extracting uncertainty from verbalizations [Cosmides and Tooby, 1996, Lin et al., 2022b, Tian et al., 2023, Xiong et al., 2024, Kojima et al., 2022, Groot and Valdenegro-Toro, 2024]. Our work models implicit uncertainty in distributions over multiple formal outputs rather than relying on individual response signals.

**Formal Grammars for Language Model Outputs** Context-Free Grammars (CFGs) have been employed to constrain language model generation at inference time, ensuring syntactically valid outputs for structured formats like JSON [Sengottuvelu, 2023], SQL and programming languages[Melcer et al., 2024]. These constrained decoding methods modify token probability distributions by masking invalid tokens according to grammar rules. Probabilistic Context-Free Grammars (PCFGs) extend CFGs by associating probabilities with production rules, finding applications in natural language parsing [Manning and Schutze, 1999], bioinformatics [Durbin et al., 1998], and program analysis [Alur et al., 2014]. Recent work [Li et al., 2024] applies PCFGs to guide enumerative program synthesis by learning probability distributions over production rules from successful synthesis traces. While their application domain differs (synthesis guidance vs. uncertainty quantification), it demonstrates the power of PCFGs for reasoning about structured program spaces. HySynth [Barke et al., 2024] similarly enables probabilistic analysis for LLM-generated DSL programs. Our work represents the first application of PCFGs to uncertainty quantification for LLM-driven autoformalization.

**Verification and Reasoning Uncertainty** DTV [Zhou et al., 2024], SAT-LM [Ye et al., 2023], and related approaches [Quan et al., 2024] connect LLMs with formal verification, while latent space methods [Lee et al., 2020, Wu and Wu, 2021] complement uncertainty estimation research [Kadavath et al., 2022, Lin et al., 2022b]. Our work extends PCFG inference [De la Higuera, 2010] to uncertainty quantification in formal verification artifacts, representing the first systematic framework for diagnosing LLM autoformalization reliability through grammatical uncertainty analysis.

# 6 Conclusion

Our research presents a PCFG-based framework for SMT-LIB UQ, establishing that syntactic atypicalities in LLM-generated formal artifacts, previously underexploited, serve as potent, quantifiable signals of underlying semantic errors. Our evaluations revealed nuanced LLM behaviors—task-dependent uncertainty responses, localized PCFG entropy's diagnostic power, and asymmetric self-consistency suggesting distinct, imperfectly aligned formal versus textual reasoning pathways. Building on these discoveries, we introduced a novel uncertainty taxonomy and a lightweight, model-agnostic signal fusion technique that improves metric synergy and calibration. Applied together, this PCFG framework and fusion strategy achieve substantial error rate reductions via selective verification, offering an empirically validated methodology to significantly enhance LLM reliability in formal verification workflows.

## Acknowledgment

This work was supported in part by the NSF research grant #2112606, #2117439, #2320952, #2137603.

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

# A Appendix: Proofs

## A.1 Coverage Guarantees

**Theorem 1 : Coverage Guarantees**

*Proof.* **Step 1: Construct a "high-probability" set.** By definition of Shannon entropy in bits, there can be at most $2^{H(\mu)}$ points $x \in \Sigma^*$ each having probability $\mu(x) \geq 2^{-H(\mu)}$. Gather all such points into a set

$$S_{\text{high}} = \{x \in \Sigma^* : \mu(x) \geq 2^{-H(\mu)}\}.$$

A standard "typical-set" argument shows that $\mu(S_{\text{high}}) \geq 1/2$ (or at least a fixed positive constant).

**Step 2: Missing $S_{\text{high}}$.** The probability that one sample $X \sim \mu$ does *not* land in $S_{\text{high}}$ is at most $1 - \mu(S_{\text{high}}) \leq 1/2$. Hence the probability that *none* of $N$ i.i.d. draws land in $S_{\text{high}}$ is at most $(1/2)^N = 2^{-N}$. Thus any set of measure at least $1/2$ is missed with exponentially small probability.

**Step 3: Missing a smaller-mass region $A$.** If $\mu(A) = \epsilon \leq 1/2$, the probability that one draw misses $A$ is $1 - \epsilon$, so the probability *all* $N$ draws miss $A$ is $(1-\epsilon)^N \approx \exp(-N\epsilon)$ if $N\epsilon$ is not too large.

However, we need a *uniform* guarantee over *all* possible $A$ of measure $\epsilon$. By representing $\Sigma^*$ as composed of at most $2^{H(\mu)}$ "atoms" each of probability at least $2^{-H(\mu)}$, the number of distinct subsets is at most $\exp(2^{H(\mu)}\ln 2)$. A union bound then modifies the exponent by about a factor of $1/2^{H(\mu)}$, so for large $N$ one has

$$\Pr[\exists A : \mu(A) = \epsilon \text{ and all } N \text{ samples miss } A] \leq \exp\left(-\tfrac{N\epsilon}{2^{H(\mu)}}\right).$$

This is the desired exponential coverage bound.

**Step 4: Inverting the bound with the Lambert $W$-function.** We have

$$\Pr\left[\text{miss some set of mass } \epsilon\right] \leq \exp\left(-\tfrac{N\epsilon}{2^{H(\mu)}}\right).$$

We want to find $\epsilon$ such that this probability is itself at most $\epsilon$:

$$\exp\left(-\tfrac{N\epsilon}{2^{H(\mu)}}\right) = \epsilon.$$

Rearrange as

$$\epsilon = \exp\left(-\tfrac{N\epsilon}{2^{H(\mu)}}\right) \quad \Longleftrightarrow \quad \epsilon \exp\left(\tfrac{N\epsilon}{2^{H(\mu)}}\right) = 1.$$

Let $x = \tfrac{N\epsilon}{2^{H(\mu)}}$. Then $\epsilon = \tfrac{2^{H(\mu)}}{N}x$, and the above becomes

$$\frac{2^{H(\mu)}}{N} x \exp(x) = 1 \quad \Longleftrightarrow \quad xe^x = \frac{N}{2^{H(\mu)}}.$$

By definition of the Lambert $W$-function, $x = W\left(\tfrac{N}{2^{H(\mu)}}\right)$. Substituting back, we get

$$\epsilon = \frac{2^{H(\mu)}}{N} W\left(\tfrac{N}{2^{H(\mu)}}\right).$$

Thus $\epsilon$ decreases to 0 as $N \to \infty$, roughly like

$\tfrac{2^{H(\mu)}}{N}\ln\left(\tfrac{N}{2^{H(\mu)}}\right)$.

Hence, the probability of missing *any* set of mass at least $\epsilon$ is $\leq \epsilon$, with $\epsilon$ scaling on the order of $\tfrac{\ln(N)}{N}$. This completes the proof. $\square$

## A.2 Temperature Sampling & Ablations

Sampling from an LLM at higher temperatures effectively "flattens" its probability distribution over next-token choices, increasing the entropy of the samples and thus encouraging exploration of lower-probability (more diverse) regions of the program space. Conversely, sampling at lower temperatures sharpens the distribution, concentrating probability mass on the model's highest-confidence predictions

and yielding lower-entropy (more conservative) samples. In other words, low-temperature sampling focuses on the most likely, canonical SMT-LIB programs (small effective support), while high-temperature sampling ventures into rarer, more varied corners of the output space (large effective support). If instead of a smoothly varying temperature schedule you simply draw many samples at fixed temperatures—say 0.5, 1.0, 1.5, and 2.0—you will still span low- to high-entropy regimes, but less systematically. You risk oversampling similar outputs at each temperature (especially near the extremes) and undersampling the intermediate entropy levels that lie between 0.5->1.0 and 1.5->2.0. A continuous schedule allocates exactly one sample per intermediate temperature, guaranteeing uniform coverage of entropy levels; fixed-temperature repetition may require substantially more draws to approximate that coverage, potentially leaving gaps in the distribution of generated programs.

**Definition 1** (Gaussian Temperature Schedule). *To smoothly explore the distribution over SMT-LIB programs, we can define a temperature schedule for $N$ samples as:*

$$\tau_i = \tau_{\min} + (\tau_{\max} - \tau_{\min}) \cdot \exp\left(-\frac{(i - N/2)^2}{2\sigma^2}\right) \tag{1}$$

*where $\tau_{\min} = 0.1$, $\tau_{\max} = 1.5$, and $\sigma = \frac{N}{5}$ controls the spread of the Gaussian.*

We can also skew this gaussian towards lower temperatures.

**Definition 2** (Exponential Temperature Schedule). *We can define a schedule that emphasizes sampling at lower temperatures using:*

$$\tau_i = \tau_{\min} + (\tau_{\max} - \tau_{\min}) \cdot \exp(-\lambda \cdot i) \tag{2}$$

*where $\lambda > 0$ controls the decay rate, and $i = 0, 1, ..., N-1$.*

**Comparison:** From a purely coverage-guarantee standpoint (i.e. our goal of hitting every "significant" region of the SMT-LIB output distribution at least once), the Systematic uniform schedule remains the most theoretically justified. It uniformly samples every temperature exactly once, from low to high. Provably minimizes the worst-case "miss probability" by evenly covering the full entropy range. Gaussians concentrates samples near the middle temperature; fewer at extremes. It does provide smooth transitions; and avoids extreme high-entropy noise. However undersamples both very low-entropy (conservative) and very high-entropy (creative) regions resulting in weaker uniform coverage. The exponential decay schedule heavily biases toward low temperatures (low entropy), and therefore quickly focuses on high-confidence outputs. However, there is almost no exploration of rare programs; poor coverage of tail regions.

## B  Temperature-Varied SMT Generation and PCFG Analysis

To empirically investigate the influence of LLM sampling temperature on the characteristics of generated formal artifacts, we performed SMT-LIB v2 program generation across a defined temperature spectrum (e.g., $T_{min} = 0.0$ to $T_{max} = 2.0$). Distinct Probabilistic Context-Free Grammars (PCFGs) were induced from the SMT program ensembles parsed at each temperature point $T_i$, modeling the LLM's syntactic and structural tendencies under each generative condition.

Our analysis of these per-temperature PCFGs revealed distinct and significant trends as sampling temperature was varied. Notably, the PCFG spectral radius generally trended upwards with increasing temperature. This intriguing behavior suggests that higher temperatures, while fostering diversity, may also enable the LLM to access and generate SMT structures with more pronounced or varied recursive complexity, perhaps by activating a broader range of complex production rules rather than simplifying structural choices. Consistent with expectations of increased diversity, grammar entropy and its associated perplexity also demonstrated an upward trend, quantifying the heightened uncertainty and the expanded set of effective choices exercised by the LLM at higher temperatures.

A particularly interesting observation was that the KL divergence from a uniform distribution also tended to increase with temperature. This implies that as the LLM explores a wider variety of production rules (evidenced by increased entropy), its choices within this expanded repertoire become, in a relative sense, more specific or structured, deviating further from a purely random uniform selection over the increasingly diverse set of utilized rules. Correspondingly, the entropy ratio generally decreased, which could occur if the maximum possible entropy (based on the growing set of observed rules and non-terminals at higher temperatures) increases at a faster rate than the actual grammar entropy. The composite metric NSUI showed

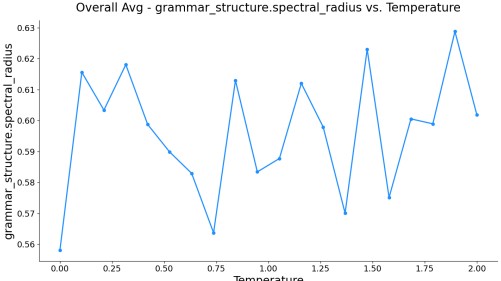

Figure 2: Spectral Radius VS Temperature

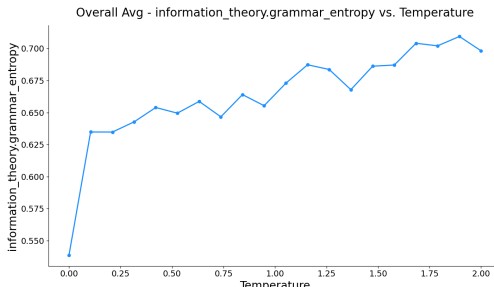

Figure 3: Grammar Entropy VS Temperature

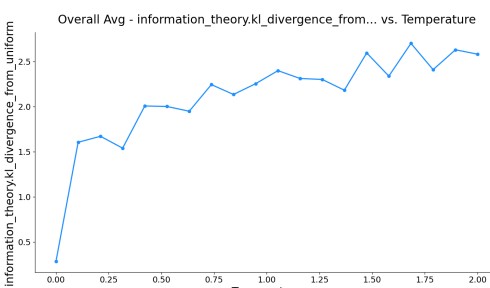

Figure 4: KLD vs Temp

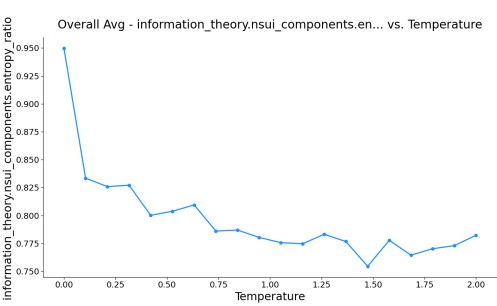

Figure 5: Entropy Ratio vs Temp

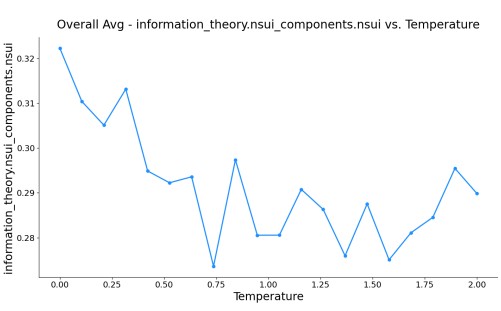

Figure 6: NSUI vs Temp

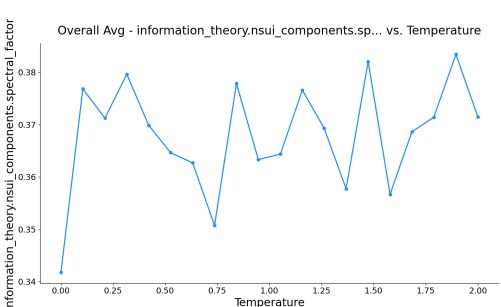

Figure 7: Spectral Factor vs Temp

fluctuating behavior without a clear monotonic direction, reflecting the complex interplay of its underlying components. The spectral factor, linked to the spectral radius, also exhibited a slight upward trend.

Metrics related to the observed grammar structure, such as the average number of rules utilized per non-terminal, the maximum observed branching factor, and the average right-hand side (RHS) length of applied rules, all generally increased with temperature. This supports the notion that higher temperatures lead the LLM to explore and employ a more extensive and potentially more elaborate subset of the SMT-LIB grammar. Regarding the shape of the rule probability distributions, kurtosis consistently decreased, indicating that these distributions become flatter (less peaked) as temperature promotes more uniform rule selection among the actively used rules. Conversely, the skew of these distributions tended to increase, suggesting a shift in the asymmetry of rule preferences as temperature changes.

These ablation studies are meaningful as they reveal a nuanced picture of the LLM's generative process for formal languages. The trends suggest that increasing temperature doesn't merely lead to random, uniform outputs, but rather allows the LLM to explore a richer, potentially more complex, and structurally diverse portion of the language space defined by $G_{SMT}$. This expansion, however, may also come with its own emergent structural specificities, as indicated by the KL divergence. These findings are crucial for

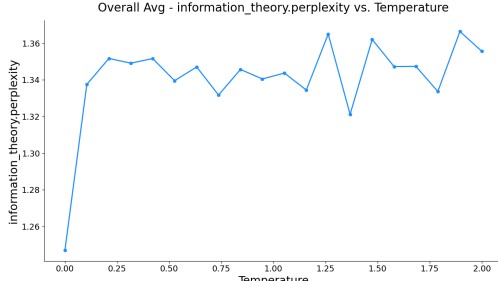

Figure 8: Perplexity vs Temp

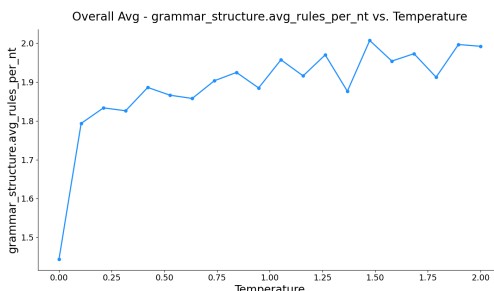

Figure 9: Average Rules per Non-terminal vs Temp

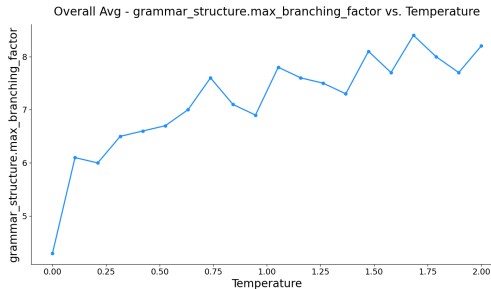

Figure 10: Max Branching Factor vs Temp

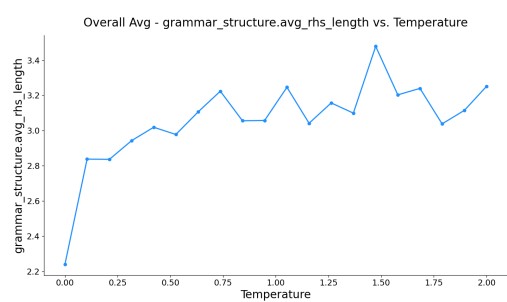

Figure 11: Average RHS Length vs Temp

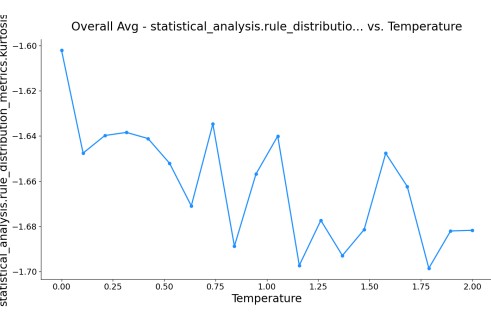

Figure 12: Kurtosis vs Temp

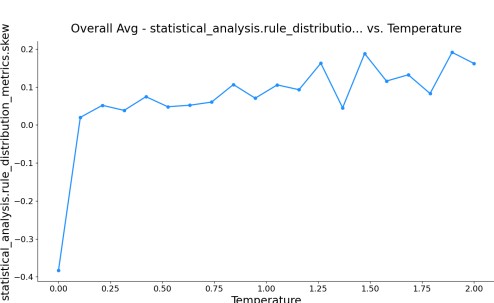

Figure 13: Skew vs Temp

understanding the coherence-diversity trade-off, for validating the sensitivity of PCFG-derived metrics, and for interpreting uncertainty scores, as the baseline characteristics of generated artifacts are systematically altered by temperature in complex ways. The observed responses underscore the value of empirical studies in characterizing LLM behavior for formal code generation.

# C    Detailed Results

## C.1    Benchmarking Autoformalization

The performance benchmarks detailed in 6, 7, 8, 9, 10 were generated by evaluating five Large Language Models (o3-mini, DeepSeekR1 with Chain-of-Thought, DeepSeek-v3-04-21, Gemini Flash 2.0, and Gemini Flash 2.0 Lite) on four reasoning datasets. For each question, five samples were generated, and answers were derived either directly from the LLM's textual output or by solving LLM-generated SMT-LIB programs using the Z3 solver. The "medium effort" designation for o3-mini indicates a specific prompting or iteration level for that model. In Table 6, the SMT approach for models like Deepseek v3 not only altered precision and recall but also resulted in substantial numbers of both False Positives (144) and True Positives (199), suggesting that while it attempted more proofs, a large fraction of these new attempts were erroneous. This contrasts with its text performance (42 FP, 174 TP). For the ProntoQA training set, with only true answers (Table 7), the SMT Precision of 1.0000 across all models is a direct consequence of the experimental design (no false statements to misclassify as true if TN is inherently 0); the variance in False Negatives (e.g., 270 for Deepseek v3 SMT) thus purely reflects the inability to successfully formalize and prove statements known to be true, a direct measure of formalization completeness for affirmatives.

Table 6: LLM Performance on StrategyQA (Text vs. SMT): SMT often boosts recall (e.g., Deepseek v3 from 0.81 to 0.91) but can reduce precision and overall accuracy for several models, highlighting model-dependent autoformalization success on knowledge-intensive tasks.

| | StrategyQA | | | | | | | | | | | | | | | |
| | Text | | | | | | | | SMT | | | | | | | |
| | Accuracy | Precision | Recall | F1 | TP | TN | FP | FN | Accuracy | Precision | Recall | F1 | TP | TN | FP | FN |
|---|---|---|---|---|---|---|---|---|---|---|---|---|---|---|---|---|
| o3-mini (medium effort) | 0.7828 | 0.8609 | 0.6047 | 0.7104 | 130 | 260 | 21 | 80 | 0.7980 | 0.8688 | 0.6347 | 0.7335 | 139 | 260 | 21 | 80 |
| Deepseek v3 0324 | 0.8292 | 0.8055 | 0.8055 | 0.8055 | 174 | 234 | 42 | 42 | 0.6720 | 0.5801 | 0.9086 | 0.7081 | 199 | 137 | 144 | 20 |
| DeepSeek R1 | 0.8580 | 0.8364 | 0.8402 | 0.8383 | 184 | 245 | 36 | 35 | 0.7760 | 0.7184 | 0.8037 | 0.7586 | 176 | 212 | 69 | 43 |
| Gemini Flash 2.0 | 0.7188 | 0.6880 | 0.6570 | 0.6720 | 144 | 214 | 65 | 75 | 0.5360 | 0.4840 | 0.9269 | 0.6363 | 203 | 65 | 216 | 16 |
| Gemini Flash 2.0 Lite | 0.6760 | 0.6770 | 0.4970 | 0.5736 | 109 | 229 | 52 | 110 | 0.4500 | 0.4419 | 0.9726 | 0.6077 | 213 | 12 | 269 | 6 |

Table 7: LLM Performance on ProntoQA Train (True Statements Only; Text vs. SMT): SMT exposes significant failures in formalizing and proving known true statements for models like Deepseek v3 (Accuracy 0.45 vs. Text 1.00), indicating critical autoformalization recall deficiencies rather than precision issues (SMT Precision remains 1.00 for all).

| | ProntoQA Train - ONLY TRUE Answers | | | | | | | | | | | | | | | |
| | Text | | | | | | | | SMT | | | | | | | |
| | Accuracy | Precision | Recall | F1 | TP | TN | FP | FN | Accuracy | Precision | Recall | F1 | TP | TN | FP | FN |
|---|---|---|---|---|---|---|---|---|---|---|---|---|---|---|---|---|
| o3-mini (medium effort) | 1.0000 | 1.0000 | 1.0000 | 1.0000 | 499 | 0 | 0 | 0 | 0.9980 | 1.0000 | 0.9980 | 0.9889 | 499 | 0 | 0 | 1 |
| Deepseek v3 0324 | 1.0000 | 1.0000 | 1.0000 | 1.0000 | 450 | 0 | 0 | 0 | 0.4501 | 1.0000 | 0.4501 | 0.6200 | 221 | 0 | 0 | 270 |
| DeepSeek R1 | 0.9939 | 1.0000 | 0.9939 | 0.9969 | 489 | 0 | 0 | 3 | 0.7440 | 1.0000 | 0.7440 | 0.8532 | 372 | 0 | 0 | 128 |
| Gemini Flash 2.0 | 0.9820 | 1.0000 | 0.9820 | 0.9900 | 491 | 0 | 0 | 9 | 0.9000 | 1.0000 | 0.9000 | 0.9470 | 450 | 0 | 0 | 50 |
| Gemini Flash 2.0 Lite | 0.9980 | 1.0000 | 0.9980 | 0.9980 | 499 | 0 | 0 | 1 | 0.9980 | 1.0000 | 0.9980 | 0.9989 | 499 | 0 | 0 | 1 |

The ProofWriter results Table 8 are notable. We advise the reader to ignore DeepSeek R1's SMT performance, since it is based on a "Partial Run," because of poor model ability to autoformalize, due to the overuse of thinking tokens, thereby causing an intractable timeline for converging to any solution and API call explosion. Here, o3-mini (medium effort) showcases a successful SMT application, where its accuracy improved to 0.9418 with a reduction in both False Positives (from 34 to 19) and False Negatives (from 21 to 10) compared to its text output. On the FOLIO dataset (Table 9), a common pattern observed in the SMT condition, beyond just low precision, was the significant reduction in True Negatives compared to the Text condition for several models (e.g., Deepseek v3 dropped from 34 TN via Text to 5 TN via SMT; Gemini Flash 2.0 from 37 TN to 0 TN). This suggests a systemic challenge in generating SMT formulas that correctly evaluate to unsatisfiable for statements that are indeed false within the FOLIO logical structure. Finally, the ProntoQA test set which includes both true and false statements (Table 10) revealed extreme model-specific behaviors under SMT. DeepSeek R1's SMT output, for instance, correctly identified all 242 false statements (0 FP, 242 TN) but failed to correctly identify any of the 258 true statements (0 TP, 258 FN), indicating a systematic bias in its SMT generation towards unsatisfiability or an inability to complete proofs for satisfiable formulas in a mixed-distribution context, a stark contrast to its perfect text performance and its SMT performance on true-only statements.

Table 8: LLM Performance on ProofWriter (Text vs. SMT): SMT substantially improves models struggling with formal logic (e.g., Gemini Flash 2.0 Lite accuracy from 0.41 to 0.75), yet can degrade performance for models already strong in textual formal reasoning (e.g., DeepSeek R1 accuracy from 0.94 to 0.49), showcasing task-specific SMT utility.

| | ProofWriter | | | | | | | | | | | | | | | |
| | Text | | | | | | | | SMT | | | | | | | |
| | Accuracy | Precision | Recall | F1 | TP | TN | FP | FN | Accuracy | Precision | Recall | F1 | TP | TN | FP | FN |
|---|---|---|---|---|---|---|---|---|---|---|---|---|---|---|---|---|
| o3-mini (medium effort) | 0.8893 | 0.8697 | 0.9153 | 0.8919 | 227 | 215 | 34 | 21 | 0.9418 | 0.9261 | 0.9597 | 0.9426 | 238 | 231 | 19 | 10 |
| Deepseek v3 0324 | 0.8057 | 0.8016 | 0.8225 | 0.8110 | 190 | 175 | 47 | 41 | 0.5800 | 0.6587 | 0.3320 | 0.4414 | 83 | 207 | 43 | 167 |
| DeepSeek R1 (Partial Run) | 0.9423 | 0.9597 | 0.9220 | 0.9400 | 143 | 151 | 6 | 12 | 0.4935 | 0.4750 | 0.1870 | 0.2685 | 29 | 125 | 32 | 126 |
| Gemini Flash 2.0 | 0.4900 | 0.4960 | 0.5710 | 0.5300 | 140 | 106 | 142 | 105 | 0.6660 | 0.6844 | 0.6160 | 0.5313 | 154 | 106 | 71 | 96 |
| Gemini Flash 2.0 Lite | 0.4060 | 0.3609 | 0.2440 | 0.2911 | 61 | 142 | 108 | 189 | 0.7540 | 0.7275 | 0.8120 | 0.7674 | 203 | 174 | 76 | 47 |

Table 9: LLM Performance on FOLIO (Text vs. SMT): Textual reasoning largely outperforms SMT. For many models, SMT results in high recall but poor precision (e.g., Gemini Flash 2.0 SMT F1 0.72 vs Text 0.92) and a failure to identify false statements (e.g., 0 SMT True Negatives for Gemini Flash 2.0), indicating issues with formalizing negation or complex FOL conditions.

| | Folio | | | | | | | | | | | | | | | |
| | Text | | | | | | | | SMT | | | | | | | |
| | Accuracy | Precision | Recall | F1 | TP | TN | FP | FN | Accuracy | Precision | Recall | F1 | TP | TN | FP | FN |
|---|---|---|---|---|---|---|---|---|---|---|---|---|---|---|---|---|
| o3-mini (medium effort) | 0.9450 | 0.9682 | 0.9384 | 0.9531 | 61 | 43 | 2 | 4 | 0.5000 | 0.6890 | 0.2985 | 0.4166 | 20 | 36 | 9 | 47 |
| Deepseek v3 0324 | 0.9333 | 0.9259 | 0.9615 | 0.9433 | 50 | 34 | 4 | 2 | 0.5961 | 0.6063 | 0.9193 | 0.7307 | 57 | 5 | 37 | 5 |
| DeepSeek R1 | 0.9252 | 0.9670 | 0.9090 | 0.9374 | 60 | 39 | 2 | 6 | 0.5200 | 0.6363 | 0.5303 | 0.5785 | 35 | 21 | 20 | 31 |
| Gemini Flash 2.0 | 0.9010 | 0.9275 | 0.9142 | 0.9200 | 64 | 37 | 5 | 6 | 0.5625 | 0.6000 | 0.9000 | 0.7200 | 63 | 0 | 42 | 7 |
| Gemini Flash 2.0-lite | 0.9017 | 0.904 | 0.9428 | 0.923 | 66 | 35 | 7 | 4 | 0.7321 | 0.7 | 1 | 0.8235 | 70 | 12 | 30 | 0 |

Table 10: LLM Performance on ProntoQA Test (True/False Mix; Text vs. SMT): SMT shows divergent outcomes: catastrophic failure for some (e.g., DeepSeek R1 SMT F1 0.00 vs. Text 1.00), yet significant improvement for others (Gemini Flash 2.0 Lite SMT Accuracy 0.78 vs. Text 0.56), highlighting inconsistent SMT reliability on mixed arithmetic statements.

| | ProntoQA TEST - BOTH TRUE AND FALSE | | | | | | | | | | | | | | | |
| | Text | | | | | | | | SMT | | | | | | | |
| | Accuracy | Precision | Recall | F1 | TP | TN | FP | FN | Accuracy | Precision | Recall | F1 | TP | TN | FP | FN |
|---|---|---|---|---|---|---|---|---|---|---|---|---|---|---|---|---|
| o3-mini (medium effort) | 1.0000 | 1.0000 | 1.0000 | 1.0000 | 258 | 240 | 0 | 0 | 1.0000 | 1.0000 | 1.0000 | 1.0000 | 258 | 242 | 0 | 0 |
| Deepseek v3 0324 | 0.7200 | 0.7138 | 0.7635 | 0.7378 | 197 | 163 | 79 | 61 | 0.5140 | 0.5484 | 0.3295 | 0.4116 | 85 | 172 | 70 | 173 |
| DeepSeek R1 | 1.0000 | 1.0000 | 1.0000 | 1.0000 | 253 | 242 | 0 | 0 | 0.4840 | 0.0000 | 0.0000 | 0.0000 | 0 | 242 | 0 | 258 |
| Gemini Flash 2.0 | 0.7180 | 0.8232 | 0.5770 | 0.6780 | 149 | 210 | 32 | 109 | 0.4560 | 0.4753 | 0.5232 | 0.4981 | 135 | 93 | 149 | 123 |
| Gemini Flash 2.0 Lite | 0.5630 | 0.5811 | 0.5333 | 0.5562 | 136 | 144 | 98 | 119 | 0.7820 | 0.7210 | 0.9418 | 0.8168 | 243 | 148 | 94 | 15 |

## C.2 Detailed Performance of Uncertainty Metrics for Ground Truth Prediction

This section provides a granular view of the performance of various Probabilistic Context-Free Grammar (PCFG) derived metrics, self-consistency measures, and ensemble methods in predicting the correctness of SMT-LIB outputs (with respect to ground truth) for specific LLM and dataset combinations. The metrics evaluated include AUROC (Area Under the Receiver Operating Characteristic Curve) for discrimination, ECE (Expected Calibration Error) and Brier score for calibration, and AURC (Area Under the Risk-Coverage Curve) along with optimal threshold (Opt.T), error rate at threshold (Err@T), and relative error reduction (RelErrRed) for selective prediction utility.

The UQ results for o3-mini on StrategyQA demonstrate moderate success in distinguishing correct SMT outputs from incorrect ones. While Grammar Entropy shows good individual discriminative power (AUROC 0.7448, AURC 0.1113), achieving a 13.88% relative error reduction by abstaining on just 5% of samples, many other standalone PCFG metrics exhibit weaker performance. The self-consistency metrics (Text and SMT) also perform well (AUROC 0.74), indicating that agreement between the LLM's own reasoning modalities is a key signal. Notably, the Ensemble ML method achieves the highest AUROC (0.7850) and a significant relative error reduction (29.29% by abstaining on 10% of samples), underscoring the benefit of integrating diverse uncertainty signals through a learned model. The comparatively higher ECE for many metrics suggests that while discriminative, their raw scores may not always be well-calibrated probabilities.

Table 11: Uncertainty Quantification for o3-mini on StrategyQA: Ensemble ML (AUROC 0.7850) and Self-Consistency metrics (Text/SMT AUROC 0.74) outperform most individual PCFG metrics (Grammar Entropy AUROC 0.7448 being a strong contender). This suggests that for o3-mini on this knowledge-intensive task, learned combinations or behavioral consistency signals are more potent than isolated SMT structural properties for error detection.

| Metric | AUROC | ECE | Brier | AURC | Opt.T | Err@T | RelErrRed |
|---|---|---|---|---|---|---|---|
| | | | StrategyQA - o3-mini | | | | |
| Grammar Entropy | 0.7448 | 0.3058 | 0.2340 | 0.1113 | 0.0500 | 0.1895 | 0.1388 |
| Perplexity | 0.5589 | 0.3107 | 0.2862 | 0.1811 | 0.2000 | 0.1750 | 0.2045 |
| KL Divergence | 0.6428 | 0.2485 | 0.2385 | 0.1471 | 0.3000 | 0.1429 | 0.3506 |
| NSUI | 0.6334 | 0.2436 | 0.2539 | 0.1250 | 0.1500 | 0.1882 | 0.1444 |
| Renyi Ent (2) | 0.5175 | 0.3303 | 0.2997 | 0.1977 | 0.3000 | 0.1857 | 0.1558 |
| Renyi Ent (0.5) | 0.5973 | 0.3398 | 0.3042 | 0.1634 | 0.2500 | 0.1600 | 0.2727 |
| Max Ent | 0.6649 | 0.3553 | 0.2935 | 0.1297 | 0.0500 | 0.1895 | 0.1388 |
| Ent Ratio | 0.5385 | 0.3283 | 0.3028 | 0.1834 | 0.3000 | 0.1857 | 0.1558 |
| Spectral Factor | 0.6334 | 0.2173 | 0.2364 | 0.1319 | 0.1500 | 0.1882 | 0.1444 |
| Spectral Radius | 0.6334 | 0.2892 | 0.2747 | 0.1319 | 0.1500 | 0.1882 | 0.1444 |
| # Nonterminals | 0.5111 | 0.3540 | 0.3188 | 0.2006 | 0.2000 | 0.2125 | 0.0341 |
| # Rules | 0.5548 | 0.2117 | 0.2385 | 0.1855 | 0.2000 | 0.1750 | 0.2045 |
| Avg Rules / NT | 0.5737 | 0.2400 | 0.2415 | 0.1752 | 0.2000 | 0.1625 | 0.2614 |
| Avg RHS Len | 0.5350 | 0.6141 | 0.5651 | 0.1979 | 0.0500 | 0.2105 | 0.0431 |
| Max Branch Factor | 0.5181 | 0.1500 | 0.1997 | 0.1990 | 0.1500 | 0.1882 | 0.1444 |
| Rule Dist Mean | 0.5740 | 0.3161 | 0.2836 | 0.1752 | 0.2000 | 0.1625 | 0.2614 |
| Rule Dist StdDev | 0.5291 | 0.3995 | 0.3517 | 0.1811 | 0.0500 | 0.2105 | 0.0431 |
| Rule Dist Skew | 0.5833 | 0.3178 | 0.2850 | 0.1689 | 0.1500 | 0.1765 | 0.1979 |
| Rule Dist Kurtosis | 0.5659 | 0.3948 | 0.3420 | 0.1785 | 0.0500 | 0.2000 | 0.0909 |
| Self Consistency Text | 0.7369 | 0.1604 | 0.1603 | 0.1081 | 0.1500 | 0.1529 | 0.3048 |
| Self Consistency SMT | 0.7416 | 0.1523 | 0.1609 | 0.1051 | 0.1500 | 0.1529 | 0.3048 |
| Ensemble Average | 0.7622 | 0.3724 | 0.2916 | 0.1103 | 0.1000 | 0.1556 | 0.2929 |
| Ensemble Weighted | 0.7657 | 0.1738 | 0.1617 | 0.1099 | 0.0500 | 0.1895 | 0.1388 |
| Ensemble ML | 0.7850 | 0.2090 | 0.1756 | 0.1013 | 0.1000 | 0.1556 | 0.2929 |
| Ensemble Simple | 0.6702 | 0.2055 | 0.2104 | 0.1410 | 0.2000 | 0.1500 | 0.3182 |

For DeepSeek-v3 on StrategyQA, UQ metrics show a somewhat different pattern compared to o3-mini. Ensemble ML again provides the best overall discrimination (AUROC 0.7709), achieving a relative error reduction of 8.96% by abstaining on 5% of the samples. Interestingly, several individual PCFG-derived metrics, such as Grammar Entropy (AUROC 0.7087), Max Entropy (AUROC 0.6851), and Spectral Factor/Radius (AUROC 0.6800), demonstrate better discriminative power than the self-consistency metrics (AUROCs 0.60-0.62). This suggests that for DeepSeek-v3 on this task, intrinsic structural characteristics of the generated SMT are more indicative of correctness than its consistency with textual outputs. While Ensemble Simple yields the highest relative error reduction (35.14%), this comes at the cost of a high abstention rate (Opt.T 0.50), indicating practical trade-offs in selective prediction.

The UQ performance for o3-mini on ProofWriter is remarkably high, demonstrating the strong potential of PCFG-based metrics in formal reasoning contexts. Numerous individual metrics, including Grammar Entropy, Perplexity (AUROC 0.9194), Renyi Entropy (0.5) (AUROC 0.9301), Average Rules / NT

Table 12: Uncertainty Quantification for DeepSeek-v3 on StrategyQA: Ensemble ML leads with an AUROC of 0.7709. Several individual PCFG-based metrics like Grammar Entropy (AUROC 0.7087) and Max Entropy (AUROC 0.6851) show reasonable efficacy, outperforming self-consistency measures for this model.

| Metric | AUROC | ECE | Brier | AURC | Opt.T | Err@T | RelErrRed |
|---|---|---|---|---|---|---|---|
| Grammar Entropy | 0.7087 | 0.1575 | 0.2302 | 0.2097 | 0.05 | 0.3474 | 0.0612 |
| Perplexity | 0.6122 | 0.1601 | 0.2641 | 0.2497 | 0.5 | 0.28 | 0.2432 |
| KL Divergence | 0.5723 | 0.1393 | 0.2322 | 0.2878 | 0.05 | 0.3579 | 0.0327 |
| NSUI | 0.5997 | 0.0781 | 0.2191 | 0.2672 | 0.1 | 0.3222 | 0.1291 |
| Renyi Ent (2) | 0.6195 | 0.1622 | 0.2679 | 0.2429 | 0.45 | 0.2727 | 0.2629 |
| Renyi Ent (0.5) | 0.6126 | 0.1623 | 0.2626 | 0.2517 | 0.5 | 0.28 | 0.2432 |
| Max Ent | 0.6851 | 0.0473 | 0.2099 | 0.2271 | 0.1 | 0.3222 | 0.1291 |
| Ent Ratio | 0.5311 | 0.1336 | 0.2538 | 0.3306 | 0.05 | 0.3684 | 0.0043 |
| Spectral Factor | 0.68 | 0.0992 | 0.2236 | 0.2365 | 0.05 | 0.3474 | 0.0612 |
| Spectral Radius | 0.68 | 0.0686 | 0.2148 | 0.2365 | 0.05 | 0.3474 | 0.0612 |
| # Nonterminals | 0.6115 | 0.1329 | 0.2469 | 0.2547 | 0.45 | 0.2909 | 0.2138 |
| # Rules | 0.6197 | 0.1109 | 0.2252 | 0.2583 | 0.15 | 0.3176 | 0.1415 |
| Avg Rules / NT | 0.6021 | 0.0902 | 0.226 | 0.2616 | 0.05 | 0.3579 | 0.0327 |
| Avg RHS Len | 0.5122 | 0.1753 | 0.2712 | 0.3279 | 0.1 | 0.3444 | 0.0691 |
| Max Branch Factor | 0.618 | 0.145 | 0.227 | 0.2688 | 0.05 | 0.3474 | 0.0612 |
| Rule Dist Mean | 0.6021 | 0.1811 | 0.2534 | 0.2616 | 0.05 | 0.3579 | 0.0327 |
| Rule Dist StdDev | 0.5281 | 0.1251 | 0.2573 | 0.3116 | 0.5 | 0.32 | 0.1351 |
| Rule Dist Skew | 0.6036 | 0.1431 | 0.2489 | 0.261 | 0.1 | 0.3444 | 0.0691 |
| Rule Dist Kurtosis | 0.5787 | 0.172 | 0.26 | 0.2754 | 0.05 | 0.3579 | 0.0327 |
| Self Consistency Text | 0.6017 | 0.2874 | 0.3048 | 0.2882 | 0.1 | 0.3444 | 0.0691 |
| Self Consistency SMT | 0.6203 | 0.2318 | 0.2745 | 0.2513 | 0.1 | 0.3444 | 0.0691 |
| Ensemble Average | 0.6795 | 0.1214 | 0.2077 | 0.2182 | 0.1 | 0.3222 | 0.1291 |
| Ensemble Weighted | 0.7211 | 0.1257 | 0.2135 | 0.1989 | 0.05 | 0.3474 | 0.0612 |
| Ensemble ML | 0.7709 | 0.0877 | 0.1968 | 0.1847 | 0.05 | 0.3368 | 0.0896 |
| Ensemble Simple | 0.6401 | 0.1763 | 0.2514 | 0.2483 | 0.5 | 0.24 | 0.3514 |

(AUROC 0.9301), and various rule distribution statistics, achieve AUROC scores exceeding 0.90. More impressively, their AURC values are exceptionally low (e.g., 0.0008 for Grammar Entropy), translating to a 100% relative error reduction by abstaining on a small fraction of samples (e.g., 10%). This strongly supports the hypothesis that syntactic irregularities in generated formal artifacts are highly indicative of underlying semantic errors when the task aligns well with the formal language. Ensemble methods elevate this performance to near-perfection (Ensemble Average AUROC 0.9949). Despite the excellent discrimination, some metrics show high ECE values (e.g., Grammar Entropy ECE 0.4419), suggesting that while they can effectively rank outputs by correctness likelihood, their raw scores may not be perfectly calibrated across the entire probability spectrum.

The UQ results for Gemini 2.0 Flash Lite on ProofWriter present a mixed picture, contrasting with o3-mini's strong performance on the same task. Many individual PCFG-derived metrics demonstrate weak discriminative ability, with AUROC scores often between 0.50 and 0.59 (e.g., Grammar Entropy at 0.5380, Spectral Radius at 0.5011). However, SMT Self Consistency stands out as a significantly stronger individual performer with an AUROC of 0.7364. Ensemble methods, particularly Ensemble ML, achieve the best overall performance (AUROC 0.7631, AURC 0.0823), leading to a 14.61% relative error reduction when abstaining on 5% of the samples. This suggests that for Gemini 2.0 Flash Lite on ProofWriter, the structural variations in its SMT outputs are less consistently tied to semantic correctness compared to o3-mini. Instead, behavioral consistency (specifically, how its SMT outputs align with each other across multiple generations) and learned patterns across a combination of (often individually weaker) signals provide more reliable error detection.

Table 13: Uncertainty Quantification for o3-mini on ProofWriter: PCFG-derived metrics achieve exceptional discriminative power (e.g., Grammar Entropy AUROC 0.9301, AURC 0.0008), enabling near-perfect error detection with minimal abstention (100% RelErrRed at Opt.T 0.10). Ensemble methods (e.g., Ensemble Average AUROC 0.9949) further refine this, confirming that SMT structural properties are extremely strong predictors of correctness for o3-mini on this formal reasoning task.

| Metric | AUROC | ECE | Brier | AURC | Opt.T | Err@T | RelErrRed |
|---|---|---|---|---|---|---|---|
| | | | **ProofWriter** | | | | |
| Grammar Entropy | 0.9301 | 0.4419 | 0.25 | 0.0008 | 0.1000 | 0.0000 | 1.0000 |
| Perplexity | 0.9194 | 0.5358 | 0.3515 | 0.0008 | 0.1000 | 0.0000 | 1.0000 |
| KL Divergence | 0.5108 | 0.5167 | 0.326 | 0.0074 | 0.0000 | 0.0106 | 0.0000 |
| NSUI | 0.5645 | 0.571 | 0.3843 | 0.0084 | 0.0000 | 0.0106 | 0.0000 |
| Renyi Ent (2) | 0.8871 | 0.5405 | 0.3598 | 0.0013 | 0.1500 | 0.0000 | 1.0000 |
| Renyi Ent (0.5) | 0.9301 | 0.4724 | 0.2879 | 0.0008 | 0.1000 | 0.0000 | 1.0000 |
| Max Ent | 0.9086 | 0.7198 | 0.555 | 0.0013 | 0.1000 | 0.0000 | 1.0000 |
| Ent Ratio | 0.586 | 0.5714 | 0.3764 | 0.0055 | 0.4500 | 0.0000 | 1.0000 |
| Spectral Factor | 0.7473 | 0.3458 | 0.2247 | 0.0032 | 0.3000 | 0.0000 | 1.0000 |
| Spectral Radius | 0.7473 | 0.3545 | 0.2305 | 0.0032 | 0.3000 | 0.0000 | 1.0000 |
| # Nonterminals | 0.8011 | 0.4267 | 0.2397 | 0.0019 | 0.2000 | 0.0000 | 1.0000 |
| # Rules | 0.5108 | 0.4186 | 0.2201 | 0.0084 | 0.0000 | 0.0106 | 0.0000 |
| Avg Rules / NT | 0.9301 | 0.5393 | 0.3499 | 0.0008 | 0.1000 | 0.0000 | 1.0000 |
| Avg RHS Len | 0.8011 | 0.6535 | 0.5086 | 0.0026 | 0.2500 | 0.0000 | 1.0000 |
| Max Branch Factor | 0.5914 | 0.2979 | 0.1377 | 0.0055 | 0.4500 | 0.0000 | 1.0000 |
| Rule Dist Mean | 0.9301 | 0.4945 | 0.3034 | 0.0008 | 0.1000 | 0.0000 | 1.0000 |
| Rule Dist StdDev | 0.5108 | 0.5555 | 0.3723 | 0.0074 | 0.5000 | 0.0000 | 1.0000 |
| Rule Dist Skew | 0.9301 | 0.4923 | 0.2987 | 0.0008 | 0.1000 | 0.0000 | 1.0000 |
| Rule Dist Kurtosis | 0.586 | 0.5107 | 0.3115 | 0.0055 | 0.4500 | 0.0000 | 1.0000 |
| Self Consistency Text | 0.899 | 0.0423 | 0.028 | 0.002 | 0.0500 | 0.0105 | 0.4684 |
| Self Consistency SMT | 0.7121 | 0.8501 | 0.7764 | 0.0025 | 0.1500 | 0.0000 | 1.0000 |
| Ensemble Average | 0.9949 | 0.3356 | 0.1414 | 0.0005 | 0.0500 | 0.0000 | 1.0000 |
| Ensemble Weighted | 0.9785 | 0.4612 | 0.2566 | 0.0003 | 0.0500 | 0.0000 | 1.0000 |
| Ensemble ML | 0.9892 | 0.0572 | 0.028 | 0.0003 | 0.0500 | 0.0000 | 1.0000 |
| Ensemble Simple | 0.9355 | 0.4419 | 0.2582 | 0.0008 | 0.1000 | 0.0000 | 1.0000 |

Table 14: Uncertainty Quantification for Gemini 2.0 Flash Lite on ProofWriter: Performance is moderate; SMT Self Consistency (AUROC 0.7364) and Ensemble ML (AUROC 0.7631) are the strongest UQ signals. Most individual PCFG structural metrics show weak discriminative power (many AUROCs 0.50-0.59), indicating that for this model on ProofWriter, behavioral consistency (SMT-based) and learned combinations are more indicative of correctness than raw SMT syntactic properties alone.

| Metric | AUROC | ECE | Brier | AURC | Opt.T | Err@T | RelErrRed |
|---|---|---|---|---|---|---|---|
| Grammar Entropy | 0.5380 | 0.3185 | 0.2869 | 0.1405 | 0.4500 | 0.1667 | 0.1081 |
| Perplexity | 0.5934 | 0.3888 | 0.3182 | 0.1267 | 0.1000 | 0.1742 | 0.0680 |
| KL Divergence | 0.5164 | 0.3080 | 0.2797 | 0.1573 | 0.0000 | 0.1869 | 0.0000 |
| NSUI | 0.5243 | 0.3186 | 0.2642 | 0.1514 | 0.1500 | 0.1845 | 0.0125 |
| Renyi Ent (2) | 0.5996 | 0.4102 | 0.3368 | 0.1285 | 0.1000 | 0.1742 | 0.0680 |
| Renyi Ent (0.5) | 0.5933 | 0.4401 | 0.3581 | 0.1258 | 0.1000 | 0.1742 | 0.0680 |
| Max Ent | 0.5417 | 0.3503 | 0.3045 | 0.1420 | 0.5000 | 0.1717 | 0.0811 |
| Ent Ratio | 0.5177 | 0.3943 | 0.3426 | 0.1548 | 0.2000 | 0.1835 | 0.0178 |
| Spectral Factor | 0.5011 | 0.5048 | 0.4157 | 0.1578 | 0.0000 | 0.1869 | 0.0000 |
| Spectral Radius | 0.5011 | 0.3930 | 0.3172 | 0.1578 | 0.0000 | 0.1869 | 0.0000 |
| # Nonterminals | 0.5167 | 0.4838 | 0.4215 | 0.1672 | 0.1000 | 0.1854 | 0.0079 |
| # Rules | 0.5549 | 0.2422 | 0.2315 | 0.1370 | 0.4000 | 0.1610 | 0.1383 |
| Avg Rules / NT | 0.5840 | 0.2790 | 0.2656 | 0.1301 | 0.3000 | 0.1377 | 0.2632 |
| Avg RHS Len | 0.5631 | 0.3413 | 0.2906 | 0.1480 | 0.1000 | 0.1685 | 0.0981 |
| Max Branch Factor | 0.5745 | 0.2189 | 0.2293 | 0.1355 | 0.3000 | 0.1522 | 0.1857 |
| Rule Dist Mean | 0.5838 | 0.4368 | 0.3713 | 0.1301 | 0.3000 | 0.1377 | 0.2632 |
| Rule Dist StdDev | 0.5144 | 0.4474 | 0.3795 | 0.1559 | 0.3500 | 0.1797 | 0.0384 |
| Rule Dist Skew | 0.5844 | 0.4511 | 0.3779 | 0.1313 | 0.3000 | 0.1522 | 0.1857 |
| Rule Dist Kurtosis | 0.5044 | 0.1437 | 0.1913 | 0.1726 | 0.4000 | 0.1610 | 0.1383 |
| Self Consistency Text | 0.5525 | 0.2283 | 0.2419 | 0.1376 | 0.4500 | 0.1545 | 0.1866 |
| Self Consistency SMT | 0.7364 | 0.3535 | 0.2751 | 0.1031 | 0.2000 | 0.1062 | 0.4408 |
| Ensemble Average | 0.6140 | 0.3922 | 0.3192 | 0.1240 | 0.1000 | 0.1722 | 0.0936 |
| Ensemble Weighted | 0.7235 | 0.3327 | 0.2539 | 0.1035 | 0.1000 | 0.1404 | 0.2484 |
| Ensemble ML | 0.7631 | 0.2897 | 0.2229 | 0.0823 | 0.0500 | 0.1596 | 0.1461 |
| Ensemble Simple | 0.6476 | 0.3867 | 0.3039 | 0.1071 | 0.2000 | 0.1519 | 0.1871 |

## C.3 Detailed Performance of SMT-Based Uncertainty Metrics for Text-Answer Prediction

This section evaluates the efficacy of uncertainty quantification (UQ) metrics derived from SMT-LIB generations in predicting the correctness of the SMT results with the corresponding textual answers. The goal is to identify when the formalization (SMT output) aligns or diverges from the model's natural language reasoning output (textual answer). On StrategyQA, o3 mini had 100% agreement between SMT and text answers, so UQ analysis for SMT-Text consistency prediction was not applicable for that specific model-dataset pair as there were no disagreements to predict. Results for other cases are detailed below.

For DeepSeek R1 on StrategyQA, we assesses how well metrics derived from its SMT generations can predict alignment with its textual answers. The results are strong: ensemble methods integrating these SMT features, such as Ensemble Weighted (AUROC 0.8494) and Ensemble Average (AUROC 0.8183), are highly effective. Notably, Text Self Consistency (AUROC 0.8245) is a top individual performer, suggesting that instability in textual outputs often correlates with SMT-Text divergence. Among metrics purely derived from SMT structure, Grammar Entropy (AUROC 0.7609) is noteworthy, achieving a 100% relative error reduction in identifying SMT-Text disagreements if one abstains on 45% of cases. This performance in predicting SMT-Text consistency is robust and highlights that both SMT structural integrity and textual stability are key indicators. The AURC values are generally very low for top performers (e.g., 0.0155 for Ensemble Weighted), indicating high utility in selectively flagging potential cross-modal disagreements.

Table 15: UQ for SMT-Text Consistency (DeepSeek R1, StrategyQA): SMT-derived metrics, especially ensembles (Ensemble Weighted AUROC 0.8494), effectively predict SMT-Text answer agreement. Text Self Consistency (AUROC 0.8245) is a strong predictor, while SMT-derived Grammar Entropy (AUROC 0.7609) also shows good utility, enabling high error reduction (100% RelErrRed at Opt.T 0.45) in identifying SMT-Text divergences.

| Metric | AUROC | ECE | Brier | AURC | Opt.T | Err@T | RelErrRed |
|---|---|---|---|---|---|---|---|
| Grammar Entropy | 0.7609 | 0.4617 | 0.2968 | 0.0216 | 0.4500 | 0.0000 | 1.0000 |
| Perplexity | 0.6211 | 0.3789 | 0.2640 | 0.0361 | 0.5000 | 0.0204 | 0.7114 |
| KL Divergence | 0.5776 | 0.4408 | 0.2855 | 0.0443 | 0.3000 | 0.0580 | 0.1801 |
| NSUI | 0.5963 | 0.2529 | 0.1433 | 0.0462 | 0.1000 | 0.0562 | 0.2055 |
| Renyi Ent (2) | 0.6242 | 0.3980 | 0.2746 | 0.0378 | 0.5000 | 0.0204 | 0.7114 |
| Renyi Ent (0.5) | 0.6149 | 0.3942 | 0.2755 | 0.0376 | 0.5000 | 0.0408 | 0.4227 |
| Max Ent | 0.7174 | 0.3994 | 0.2341 | 0.0262 | 0.0500 | 0.0638 | 0.0973 |
| Ent Ratio | 0.5217 | 0.5047 | 0.3529 | 0.0543 | 0.3000 | 0.0580 | 0.1801 |
| Spectral Factor | 0.5481 | 0.1533 | 0.0927 | 0.0640 | 0.1500 | 0.0476 | 0.3265 |
| Spectral Radius | 0.5481 | 0.2067 | 0.1166 | 0.0640 | 0.1500 | 0.0476 | 0.3265 |
| # Nonterminals | 0.5264 | 0.5679 | 0.4237 | 0.0557 | 0.5000 | 0.0408 | 0.4227 |
| # Rules | 0.6087 | 0.3083 | 0.1839 | 0.0396 | 0.4000 | 0.0508 | 0.2809 |
| Avg Rules / NT | 0.7034 | 0.4068 | 0.2532 | 0.0273 | 0.0500 | 0.0638 | 0.0973 |
| Avg RHS Len | 0.5637 | 0.2491 | 0.1566 | 0.0544 | 0.1000 | 0.0562 | 0.2055 |
| Max Branch Factor | 0.6801 | 0.3325 | 0.2042 | 0.0324 | 0.0500 | 0.0638 | 0.0973 |
| Rule Dist Mean | 0.7034 | 0.5352 | 0.3733 | 0.0273 | 0.0500 | 0.0638 | 0.0973 |
| Rule Dist StdDev | 0.6056 | 0.6755 | 0.5361 | 0.0413 | 0.2000 | 0.0506 | 0.2839 |
| Rule Dist Skew | 0.6848 | 0.4800 | 0.3260 | 0.0309 | 0.0500 | 0.0638 | 0.0973 |
| Rule Dist Kurtosis | 0.5311 | 0.2521 | 0.1588 | 0.0534 | 0.1000 | 0.0674 | 0.0465 |
| Self Consistency Text | 0.8245 | 0.1002 | 0.0751 | 0.0155 | 0.0500 | 0.0319 | 0.5486 |
| Self Consistency SMT | 0.7570 | 0.2062 | 0.1373 | 0.0268 | 0.0500 | 0.0532 | 0.2477 |
| Ensemble Average | 0.8183 | 0.4695 | 0.3058 | 0.0180 | 0.2500 | 0.0135 | 0.8089 |
| Ensemble Weighted | 0.8494 | 0.3256 | 0.1711 | 0.0155 | 0.0500 | 0.0319 | 0.5486 |
| Ensemble ML | 0.8245 | 0.3084 | 0.2003 | 0.0170 | 0.2000 | 0.0380 | 0.4629 |
| Ensemble Simple | 0.6957 | 0.4363 | 0.2748 | 0.0316 | 0.1000 | 0.0562 | 0.2055 |

When predicting SMT-Text consistency for DeepSeek v3 on StrategyQA, UQ metrics based on SMT generations prove highly effective. The Ensemble ML approach, which learns from various SMT-derived features, achieves an impressive AUROC of 0.8517 and offers a 55.56% relative error reduction in spotting SMT-Text disagreements when abstaining on 25% of samples. Good individual predictors include SMT-derived Grammar Entropy (AUROC 0.7354) and SMT Self Consistency (AUROC 0.7116). This demonstrates that for DeepSeek v3, deviations from typical SMT structure (signaled by grammar entropy) or inconsistencies in the SMT generation process itself are strong indicators that the SMT output might not align with the model's textual answer. The low AURC (0.0573) for Ensemble ML highlights its practical utility. This task of predicting internal consistency (SMT-Text) shows strong signals, comparable to or even clearer (e.g. for Ensemble ML) than predicting SMT-Ground Truth correctness for this model on the same dataset.

For Gemini Flash 2.0 Lite on StrategyQA, the task of predicting SMT-Text consistency reveals a standout individual metric: Rule Distribution Kurtosis from the SMT generations achieves a very high AUROC of 0.8695. This is a particularly interesting finding, as it suggests that the "tailedness" or outlier presence

Table 16: UQ for SMT-Text Consistency (DeepSeek v3, StrategyQA): Ensemble ML using SMT-derived features shows excellent performance (AUROC 0.8517) in predicting SMT-Text agreement, with a 55.56% relative error reduction. SMT-derived Grammar Entropy (AUROC 0.7354) and SMT Self Consistency (AUROC 0.7116) also serve as solid individual predictors, indicating that atypical SMT structures and generation instability can flag potential SMT-Text divergences.

| Metric | AUROC | ECE | Brier | AURC | Opt.T | Err@T | RelErrRed |
|---|---|---|---|---|---|---|---|
| Grammar Entropy | 0.7354 | 0.3058 | 0.2551 | 0.097 | 0.0500 | 0.1789 | 0.1479 |
| Perplexity | 0.6721 | 0.3282 | 0.2718 | 0.1205 | 0.3000 | 0.1143 | 0.4558 |
| KL Divergence | 0.5335 | 0.1195 | 0.178 | 0.1633 | 0.0500 | 0.2000 | 0.0476 |
| NSUI | 0.5973 | 0.1768 | 0.1919 | 0.1411 | 0.0500 | 0.1895 | 0.0977 |
| Renyi Ent (2) | 0.6799 | 0.3403 | 0.2808 | 0.116 | 0.3000 | 0.1000 | 0.5238 |
| Renyi Ent (0.5) | 0.6667 | 0.3333 | 0.2751 | 0.1223 | 0.3500 | 0.1077 | 0.4872 |
| Max Ent | 0.6353 | 0.1309 | 0.1738 | 0.1247 | 0.1000 | 0.1889 | 0.1005 |
| Ent Ratio | 0.6154 | 0.4091 | 0.3307 | 0.1446 | 0.5000 | 0.1000 | 0.5238 |
| Spectral Factor | 0.7034 | 0.1254 | 0.158 | 0.1206 | 0.1000 | 0.1667 | 0.2063 |
| Spectral Radius | 0.7034 | 0.1896 | 0.1752 | 0.1206 | 0.1000 | 0.1667 | 0.2063 |
| # Nonterminals | 0.5549 | 0.2271 | 0.2455 | 0.148 | 0.0500 | 0.2000 | 0.0476 |
| # Rules | 0.5675 | 0.2025 | 0.2077 | 0.1485 | 0.1000 | 0.1778 | 0.1534 |
| Avg Rules / NT | 0.6034 | 0.1393 | 0.1854 | 0.1399 | 0.0500 | 0.1895 | 0.0977 |
| Avg RHS Len | 0.5208 | 0.1254 | 0.1818 | 0.1972 | 0.0500 | 0.2000 | 0.0476 |
| Max Branch Factor | 0.5937 | 0.1563 | 0.192 | 0.1457 | 0.1000 | 0.1889 | 0.1005 |
| Rule Dist Mean | 0.6034 | 0.321 | 0.2742 | 0.1399 | 0.0500 | 0.1895 | 0.0977 |
| Rule Dist StdDev | 0.6281 | 0.2226 | 0.2221 | 0.1445 | 0.3000 | 0.1571 | 0.2517 |
| Rule Dist Skew | 0.5986 | 0.3057 | 0.2619 | 0.1406 | 0.0500 | 0.1895 | 0.0977 |
| Rule Dist Kurtosis | 0.66 | 0.0731 | 0.1576 | 0.1256 | 0.0500 | 0.1895 | 0.0977 |
| Self Consistency Text | 0.5778 | 0.1874 | 0.2008 | 0.1754 | 0.1500 | 0.1765 | 0.1597 |
| Self Consistency SMT | 0.7116 | 0.1662 | 0.1909 | 0.1054 | 0.1000 | 0.1778 | 0.1534 |
| Ensemble Average | 0.7064 | 0.1876 | 0.1831 | 0.0983 | 0.1000 | 0.1667 | 0.2063 |
| Ensemble Weighted | 0.7709 | 0.234 | 0.1971 | 0.0798 | 0.0500 | 0.1895 | 0.0977 |
| Ensemble ML | 0.8517 | 0.1861 | 0.1703 | 0.0573 | 0.2500 | 0.0933 | 0.5556 |
| Ensemble Simple | 0.6727 | 0.3021 | 0.2567 | 0.1119 | 0.1500 | 0.1765 | 0.1597 |

in the distribution of PCFG rules used during SMT generation is a very strong signal of whether the SMT output will align with the textual answer for this model. This metric's performance surpasses many other individual PCFG metrics (e.g., Grammar Entropy AUROC 0.6622, Perplexity AUROC 0.7212). Ensemble methods, like Ensemble Weighted (AUROC 0.8070) and Ensemble Average (AUROC 0.7927), provide robust overall performance, leveraging combinations of signals. The strong performance of kurtosis aligns with the our discussion about "syntactic fingerprints" and how atypical SMT patterns (like bimodal distributions captured by kurtosis) can signal reasoning issues or misalignments. The AURC for Kurtosis (0.4448) suggests that while discriminative, its practical utility in terms of risk reduction might require careful thresholding, achieving a 30.56% error reduction at a 50% abstention rate.

The results for o3-mini on ProofWriter for predicting SMT-Text consistency are exceptional. Ensemble ML and Ensemble Weighted methods achieve perfect AUROC scores of 1.0000, signifying an ability to flawlessly distinguish SMT outputs that align with textual answers from those that diverge. This allows for a 100% relative error reduction with a very low 5% abstention rate. Beyond ensembles, many individual PCFG metrics derived from the SMT generations show extremely high predictive capabilities. For instance, Ent Ratio (AUROC 0.9677), Rule Dist Kurtosis (AUROC 0.9462), Max Ent (AUROC 0.9355), and NSUI (AUROC 0.9355) are all remarkably strong predictors, each achieving 100% relative error reduction at their respective optimal thresholds. This indicates that for o3-mini, particularly on a formal reasoning task like ProofWriter, the structural and probabilistic characteristics of its SMT generations are almost perfectly indicative of whether its formal and textual reasoning pathways are aligned. The exceptionally low AURC values (e.g., 0.0003 for Ensemble ML) further emphasize the practical certainty offered by these UQ measures in this context. This level of predictability for SMT-Text consistency is even more pronounced than some of the SMT-Ground Truth prediction results for this model, demonstrating the power of SMT features for diagnosing internal reasoning coherence.

Table 17: UQ for SMT-Text Consistency (Gemini Flash 2.0 Lite, StrategyQA): Rule Distribution Kurtosis (AUROC 0.8695) from SMT generations is an exceptionally strong individual predictor of SMT-Text agreement, significantly outperforming other PCFG metrics. Ensemble methods (e.g., Ensemble Weighted AUROC 0.8070) also perform well. This highlights a specific SMT structural feature as a key indicator of cross-modal alignment for this model.

| Metric | AUROC | ECE | Brier | AURC | Opt.T | Err@T | RelErrRed |
|---|---|---|---|---|---|---|---|
| Grammar Entropy | 0.6622 | 0.1277 | 0.2095 | 0.5689 | 0.1000 | 0.6889 | 0.0432 |
| Perplexity | 0.7212 | 0.3255 | 0.2849 | 0.5273 | 0.0500 | 0.7053 | 0.0205 |
| KL Divergence | 0.5486 | 0.2387 | 0.263 | 0.6341 | 0.1000 | 0.7000 | 0.0278 |
| NSUI | 0.6741 | 0.1195 | 0.1624 | 0.5221 | 0.5000 | 0.6600 | 0.0833 |
| Renyi Ent (2) | 0.7624 | 0.3192 | 0.2624 | 0.5037 | 0.1000 | 0.6889 | 0.0432 |
| Renyi Ent (0.5) | 0.6994 | 0.2766 | 0.2619 | 0.5438 | 0.0500 | 0.7053 | 0.0205 |
| Max Ent | 0.5982 | 0.3401 | 0.3083 | 0.59 | 0.3000 | 0.6714 | 0.0675 |
| Ent Ratio | 0.5511 | 0.1641 | 0.2511 | 0.6346 | 0.2500 | 0.6667 | 0.0741 |
| Spectral Factor | 0.6538 | 0.0943 | 0.1677 | 0.5316 | 0.5000 | 0.6600 | 0.0833 |
| Spectral Radius | 0.6538 | 0.1648 | 0.1703 | 0.5316 | 0.5000 | 0.6600 | 0.0833 |
| # Nonterminals | 0.5166 | 0.418 | 0.3958 | 0.6509 | 0.3500 | 0.6769 | 0.0598 |
| # Rules | 0.5749 | 0.4256 | 0.3818 | 0.6252 | 0.1000 | 0.6889 | 0.0432 |
| Avg Rules / NT | 0.6565 | 0.2913 | 0.2761 | 0.5923 | 0.2500 | 0.6400 | 0.1111 |
| Avg RHS Len | 0.6014 | 0.4855 | 0.442 | 0.6098 | 0.0500 | 0.7053 | 0.0205 |
| Max Branch Factor | 0.5818 | 0.5167 | 0.4747 | 0.6313 | 0.0500 | 0.7053 | 0.0205 |
| Rule Dist Mean | 0.6565 | 0.2197 | 0.228 | 0.5923 | 0.2500 | 0.6400 | 0.1111 |
| Rule Dist StdDev | 0.7183 | 0.3136 | 0.2807 | 0.5455 | 0.1500 | 0.6706 | 0.0686 |
| Rule Dist Skew | 0.6796 | 0.1783 | 0.2199 | 0.572 | 0.2500 | 0.6400 | 0.1111 |
| Rule Dist Kurtosis | 0.8695 | 0.3187 | 0.2412 | 0.4448 | 0.5000 | 0.5000 | 0.3056 |
| Self Consistency Text | 0.535 | 0.2788 | 0.2616 | 0.6064 | 0.3500 | 0.6462 | 0.1026 |
| Self Consistency SMT | 0.7505 | 0.538 | 0.4357 | 0.4822 | 0.5000 | 0.5600 | 0.2222 |
| Ensemble Average | 0.7927 | 0.1531 | 0.1702 | 0.4848 | 0.4000 | 0.5833 | 0.1898 |
| Ensemble Weighted | 0.807 | 0.1673 | 0.1632 | 0.4718 | 0.3000 | 0.6143 | 0.1468 |
| Ensemble ML | 0.7946 | 0.1308 | 0.1592 | 0.4784 | 0.5000 | 0.5400 | 0.2500 |
| Ensemble Simple | 0.7584 | 0.0796 | 0.1568 | 0.4929 | 0.1500 | 0.6824 | 0.0523 |

Table 18: UQ for SMT-Text Consistency (o3-mini, ProofWriter): SMT-derived UQ metrics demonstrate outstanding performance, with Ensemble ML and Ensemble Weighted achieving perfect AUROC (1.0000) in predicting SMT-Text agreement. Numerous individual PCFG metrics, such as Ent Ratio (AUROC 0.9677) and Rule Dist Kurtosis (AUROC 0.9462), are also exceptionally effective, enabling complete identification of SMT-Text inconsistencies with minimal abstention. This underscores a very strong link between SMT formalization properties and cross-modal reasoning alignment for o3-mini on this formal task.

| Metric | AUROC | ECE | Brier | AURC | Opt.T | Err@T | RelErrRed |
|---|---|---|---|---|---|---|---|
| Grammar Entropy | 0.8602 | 0.4419 | 0.2542 | 0.0013 | 0.15 | 0.00 | 1.00 |
| Perplexity | 0.9032 | 0.4429 | 0.2616 | 0.0013 | 0.10 | 0.00 | 1.00 |
| KL Divergence | 0.6667 | 0.5167 | 0.3238 | 0.0039 | 0.35 | 0.00 | 1.00 |
| NSUI | 0.9355 | 0.4077 | 0.2172 | 0.0008 | 0.10 | 0.00 | 1.00 |
| Renyi Ent (2) | 0.9032 | 0.4382 | 0.2580 | 0.0013 | 0.10 | 0.00 | 1.00 |
| Renyi Ent (0.5) | 0.8925 | 0.5063 | 0.3246 | 0.0013 | 0.15 | 0.00 | 1.00 |
| Max Ent | 0.9355 | 0.2589 | 0.1029 | 0.0008 | 0.10 | 0.00 | 1.00 |
| Ent Ratio | 0.9677 | 0.4074 | 0.2082 | 0.0003 | 0.05 | 0.00 | 1.00 |
| Spectral Factor | 0.5269 | 0.6329 | 0.5061 | 0.0084 | 0.00 | 0.01 | 0.00 |
| Spectral Radius | 0.5269 | 0.6243 | 0.4953 | 0.0084 | 0.00 | 0.01 | 0.00 |
| # Nonterminals | 0.6505 | 0.5520 | 0.3650 | 0.0047 | 0.40 | 0.00 | 1.00 |
| # Rules | 0.5108 | 0.4186 | 0.2201 | 0.0064 | 0.50 | 0.00 | 1.00 |
| Avg Rules / NT | 0.8011 | 0.4394 | 0.2554 | 0.0026 | 0.25 | 0.00 | 1.00 |
| Avg RHS Len | 0.5054 | 0.6535 | 0.5116 | 0.0074 | 0.50 | 0.00 | 1.00 |
| Max Branch Factor | 0.7419 | 0.6809 | 0.5100 | 0.0032 | 0.30 | 0.00 | 1.00 |
| Rule Dist Mean | 0.8011 | 0.4842 | 0.2971 | 0.0026 | 0.25 | 0.00 | 1.00 |
| Rule Dist StdDev | 0.8710 | 0.4232 | 0.2392 | 0.0013 | 0.15 | 0.00 | 1.00 |
| Rule Dist Skew | 0.8172 | 0.4864 | 0.2964 | 0.0019 | 0.20 | 0.00 | 1.00 |
| Rule Dist Kurtosis | 0.9462 | 0.5107 | 0.3056 | 0.0008 | 0.10 | 0.00 | 1.00 |
| Self Consistency Text | 0.7050 | 0.9352 | 0.9023 | 0.0030 | 0.30 | 0.00 | 1.00 |
| Self Consistency SMT | 0.7100 | 0.8600 | 0.7863 | 0.0030 | 0.30 | 0.00 | 1.00 |
| Ensemble Average | 0.9300 | 0.3560 | 0.1741 | 0.0007 | 0.10 | 0.00 | 1.00 |
| Ensemble Weighted | 1.0000 | 0.4231 | 0.2375 | 0.0003 | 0.05 | 0.00 | 1.00 |
| Ensemble ML | 1.0000 | 0.0496 | 0.0199 | 0.0003 | 0.05 | 0.00 | 1.00 |
| Ensemble Simple | 0.6667 | 0.4419 | 0.2661 | 0.0039 | 0.35 | 0.00 | 1.00 |

# D  Supplementary Experimental Details

The comprehensive PCFG analysis underpinning our uncertainty quantification was conducted on a focused set of benchmarks. Specifically, for 100 questions each from the StrategyQA, ProofWriter, and ProntoQA datasets, a corpus of $N_{SMT}=100$ SMT-LIB v2 program samples per question was generated. The FOLIO dataset was excluded from this detailed PCFG study due to challenges in obtaining consistently robust SMT formalizations from the evaluated LLMs. Each SMT program within these corpora was parsed using an ANTLR-based parser to extract its constituent production rules. For the generation of these primary SMT samples used in uncertainty quantification (distinct from the temperature ablation study), LLM sampling temperature was maintained at its default setting to promote more deterministic outputs, with up to 10 generation attempts per SMT sample to ensure corpus completeness.

For each of the selected questions, a unique PCFG was induced from its corresponding 100 SMT samples. Rule probabilities within these per-question PCFGs were estimated via Maximum Likelihood Estimation (MLE), incorporating Lidstone smoothing (specifically, Laplace smoothing with $\beta_s=1$) to manage unseen production rules. Beyond the metrics detailed in the main methodology, specific configurations included the computation of Rényi entropy for orders $\alpha=0.5$ and $\alpha=2.0$ (Collision Entropy).

The evaluation framework for the derived uncertainty metrics incorporated specific settings. Expected Calibration Error (ECE) was calculated using 10 discretization bins for confidence scores. In the analysis of selective prediction utility (error vs. abstention), optimal abstention thresholds were determined by targeting maximum relative error reduction while considering abstention levels up to a practical maximum of 50%. For our Ensemble ML predictor, a Logistic Regression model was employed, configured with balanced class weights and trained for up to 10,000 iterations on scaled features derived from the suite of PCFG uncertainty metrics.

# E  SMT Error Ratios vs Text Error Ratios

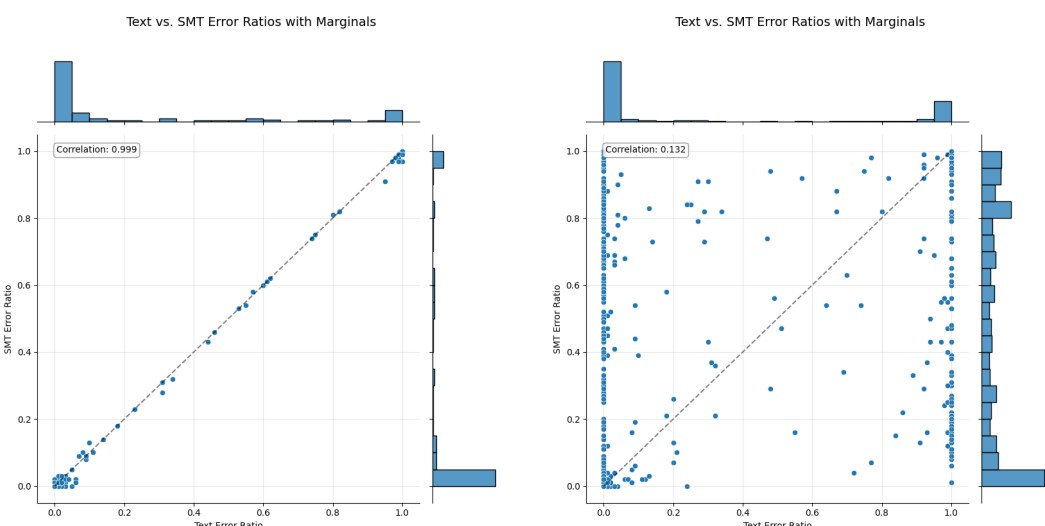

Figure 14: SMT vs. Text Error Ratio Analysis for o3-mini: Illustrates well-calibrated SMT generation, indicated by a strong correlation between SMT and Text error patterns.

Figure 15: SMT vs. Text Error Ratio Analysis for Gemini Flash 2.0: Depicts less-calibrated SMT generation, evidenced by a weaker correlation between SMT and Text error patterns.

The figures juxtapose SMT versus Text error ratios (with marginals) for o3-mini (Fig. 14) and Gemini Flash 2.0 (Fig. 15); *the Text error ratio is defined as the proportion of incorrect direct textual answers from the LLM per question out of the many samples, while the SMT error ratio is the proportion of incorrect answers derived from its SMT-LIB formalizations.* O3-mini exhibits a notable correlation between its SMT and Text error distributions, characteristic of a *well-calibrated SMT generation process* where formalization errors tend to align with textual reasoning errors. In contrast, Gemini Flash 2.0 shows a weaker correlation,

suggesting its SMT generations may introduce errors or exhibit patterns less consistently coupled with its textual output, indicative of *poorer calibration*. This comparative error ratio analysis is valuable for assessing the fidelity of an LLM's autoformalization. Strong SMT-Text error correlation implies that the SMT modality can be a more reliable indicator of the LLM's general reasoning tendencies for a problem, making SMT-derived uncertainty metrics potentially more transferable. Poor correlation, however, signals a divergence between textual reasoning and formalization, cautioning against using SMT outputs as direct proxies without careful consideration of modality-specific error sources and motivating efforts towards better SMT-Text reasoning alignment.

## F    Qualitative Analysis

Beyond quantitative uncertainty metrics, the PCFG framework, by its nature of parsing and structuring program ensembles, lends itself to a nuanced qualitative analysis of LLM-generated formal artifacts. Initial explorations can focus on broad characteristics such as the distribution of SMT-LIB sorts (datatypes) employed or the prevalent logical fragments (e.g., 'QF_LIA', 'QF_AUFBV') selected by the LLM for a given problem class. However, a more profound understanding of an LLM's formalization strategy emerges from a detailed examination of substructures, like the `assert` statements, which constitute the semantic core of an SMT program by stipulating the conditions and axioms for the solver. Our PCFG-based analysis of these assertions, and the logical architectures therein, reveals critical patterns in how LLMs attempt to translate natural language problem specifications into rigorous, machine-interpretable logic.

When an LLM generates multiple SMT program samples for a single natural language input, the per-problem induced PCFG captures a distribution over grammatical structures. This distribution inherently models the LLM's normative formalization pathways alongside its idiosyncratic variations, particularly in the construction of `assert` statements and their nested logical terms—including quantifiers (`forall`, `exists`), logical connectives (`=>`, `and`, `or`, `not`), and predicate applications. Analyzing the probabilities and diversity of production rules within these PCFGs allows for the identification and interpretation of several key types of divergences and tendencies in formal specification:

**Formalization Aliasing and Representational Stability**    A core aspect of a problem specification may elicit syntactically diverse, yet ideally logically equivalent, `assert` statements across an ensemble of LLM generations. For instance, an implication $A \Rightarrow B$ might be directly asserted or rendered as $\neg A \lor B$. The PCFG reflects such syntactic polymorphism through multiple, lower-probability rule sequences mapping to the same underlying semantic constraint. A high degree of such variability for asserting fundamental problem axioms often signals the LLM's lack of a converged or canonical formalization strategy, potentially indicating uncertainty or representational underspecification for that particular logical construct.

**Variance in Logical Decomposition and Structural Complexity**    The PCFG rule sets unveil the LLM's implicit preferences regarding the structural complexity and granularity of asserted logical terms. For a given problem, some SMT samples might employ deeply nested quantifiers and connectives within a monolithic `assert` statement. In contrast, other samples might exhibit a preference for flatter, more direct assertions or decompose a complex axiom into several simpler, conjoined `assert` statements. This divergence in logical decomposition strategies is captured by differing rule probabilities and derivation depths within the PCFG, pointing to variations in the LLM's approach to abstraction and information chunking during the formalization process.

**Identification of Atypical or Anomalous Assertions**    Occasionally, an LLM may generate `assert` statements possessing highly unusual or infrequent syntactic structures relative to the typical formalizations observed for a given problem context or across a dataset. The PCFG methodology inherently highlights these as low-probability production rules or derivations. Qualitative inspection of the SMT code corresponding to these rare assertion patterns can uncover unique, potentially innovative, or conversely, flawed and overly convoluted ways the LLM attempts to axiomatize specific constraints, offering insights into its error modes or its capacity for novel formal expression.

**Semantic Divergence in Axiomatization**    More critically, divergences can be semantic rather than merely syntactic, leading to logically distinct problem formalizations from the same natural language input. Such semantic drift often manifests as significantly different asserted terms within `assert` statements, pointing to LLM misinterpretation, unresolved ambiguity, or flaws in its inferential reasoning. For example, if an input "All engineers use LaTeX" is ambiguously formalized, one SMT sample might correctly

assert (forall ((x Engineer)) (usesLaTeX x)), while a semantically divergent sample might erroneously assert (forall ((x User)) (implies (usesLaTeX x) (isEngineer x))). The PCFG rules governing the predicates, variables, and logical structure of terms within these assertions would markedly differ, directly reflecting this semantic incongruity and providing a diagnostic trace.

**Fidelity in Representing Ground Facts** For declarative factual statements present in the input (e.g., "Constantine is a logician"), the LLM is expected to consistently assert the corresponding ground fact in a stable manner. If the PCFG reveals multiple, conflicting, or unstable rule applications for asserting properties of specific entities (e.g., some derivations asserting `(isLogician constantine)` while others, for the same conceptual input fact, generate `(isPhilosopher constantine)` or vary the predicate structure), this indicates a deficiency in the LLM's fidelity in extracting and consistently formalizing elementary factual information, pointing to potential grounding issues.

| Program 1 | Program 2 | Program 3 | Program 4 |
|---|---|---|---|
| `(set-logic QF_LIA)`
`(declare-const rela-`
`tion_count_G Int)`
`(declare-const rela-`
`tion_count_J Int)`
`(assert (> relation_count_G`
`relation_count_J))`
`(assert (>= relation_count_G`
`1))`
`(assert (>= relation_count_J`
`0))`
`(check-sat)`
`(get-model)` | `(set-logic QF_LIA)`
`(declare-const GC Int)`
`(declare-const JC Int)`
`(assert (> GC 0))`
`(assert (> JC 0))`
`(assert (> GC JC))`
`(assert (>= GC 10))`
`(assert (>= JC 1))`
`(check-sat)`
`(get-model)` | `(set-logic QF_LIA)`
`(declare-fun people_genghis ()`
`Int)`
`(declare-fun people_caesar ()`
`Int)`
`(assert (>= people_genghis`
`0))`
`(assert (>= people_caesar 0))`
`(assert (> people_genghis peo-`
`ple_caesar))`
`(assert (>= people_genghis`
`1000000))`
`(assert (<= people_caesar`
`500000))`
`(check-sat)`
`(get-model)` | `(set-logic QF_LIA)`
`(declare-const KN Int)`
`(declare-const CA Int)`
`(assert (> KN CA))`
`(assert (= KN 16))`
`(assert (= CA 1))`
`(check-sat)`
`(get-model)` |

Table 19: Divergent LLM Formalizations of a StrategyQA Problem: Sample SMT-LIB outputs illustrating varied axiomatization strategies, with `assert` statements highlighted (blue). Such variations are central to the qualitative PCFG analysis discussed.

# G Computational Efficiency

The overhead of our uncertainty quantification pipeline is minimal and does not constitute a performance bottleneck. Table 20 presents comprehensive runtime and memory profiles for the key computational components. Z3 solver execution on individual SMT programs averages 10ms with minimal memory footprint (24.7MB mean). PCFG construction from 100 SMT samples requires approximately 1.3 seconds, while subsequent uncertainty metric calculation adds only 0.4 seconds. These measurements confirm that our experimental pipeline is dominated by LLM inference time rather than formal verification or uncertainty quantification overhead. All experiments employed a 30-second timeout for Z3 execution to handle edge cases where malformed SMT programs might cause solver delays, though such timeouts were rarely triggered in practice.

Table 20: Runtime and memory profiles for uncertainty quantification pipeline components. All measurements represent averages across thousands of executions on our experimental infrastructure.

| Component | Mean | Median | P90 | P95 | Std Dev | Peak Mem |
|---|---|---|---|---|---|---|
| Z3 per SMT program | 10ms | 10ms | 11ms | 11ms | 1ms | 26.4MB |
| PCFG construction (100 SMT) | 1.297s | 1.286s | 1.454s | 1.527s | 0.131s | 161.9MB |
| UQ metrics calculation | 0.408s | 0.402s | 0.423s | 0.431s | 0.023s | 118.9MB |

# H Discussion around Risk Thresholds

**Calibration and Risk Management:** The relationship between typicality and correctness isn't straightforward; metrics with superior discriminative ability often exhibit poor calibration, indicating anomaly magnitude doesn't linearly predict error probability. This calibration challenge necessitates careful deployment considerations. Our evaluation explicitly addresses this through multiple complementary metrics: Expected Calibration Error (ECE) measures how well predicted confidence scores align with

empirical accuracy; Brier scores quantify probabilistic prediction quality; and the Area Under the Risk-Coverage Curve (AURC) evaluates practical risk mitigation through selective abstention. For instance, on ProofWriter with o3-mini, our Ensemble ML method achieves both excellent discrimination (AUROC=0.9892) and strong calibration (ECE=0.0572, Brier=0.0280), demonstrating that well-calibrated uncertainty estimates are achievable through careful metric fusion.

The selective verification framework inherently provides risk management: engineers can adjust abstention thresholds along the Risk-Coverage curve based on application requirements—choosing low error rates with higher abstention for safety-critical domains, or higher coverage with moderate error rates for less critical applications. Our AURC metric summarizes this tradeoff across all possible thresholds, with our ensemble methods consistently achieving lower AURC values than individual metrics, indicating superior risk-coverage profiles. Importantly, our approach is conservative by design, erring toward caution by abstaining when uncertain rather than providing false confidence. When our UQ metrics themselves exhibit miscalibration, the multi-metric evaluation strategy (AUROC, ECE, Brier, AURC) provides redundancy to catch such cases—no single metric failure can compromise the entire uncertainty quantification pipeline.

# I   Extensibility to Prolog: Pilot Study

To demonstrate that our PCFG-based uncertainty quantification framework generalizes beyond SMT-LIB to other formal languages, we conducted a pilot study applying our methodology to Prolog programs. This extensibility validation is critical for establishing that our approach addresses fundamental aspects of LLM-generated formal artifacts rather than being narrowly tailored to SMT-specific properties.

## I.1   Methodology

Our implementation leverages the modular design of our framework: the PCFG construction pipeline uses ANTLR for parsing, which supports any language with a defined grammar specification. We utilized the Prolog.g4 grammar from the ANTLR-v4 public repository, requiring only minimal adapter code to integrate Prolog parsing into our existing uncertainty quantification pipeline. This demonstrates that extending to new formal languages requires engineering effort proportional to grammar complexity rather than fundamental algorithmic redesign.

## I.2   Example: Logical Statement Translation

We converted the example from Figure 1 of the main paper—"Everyone who studies math or physics and works hard will succeed"—into five semantically distinct Prolog formalizations that mirror the logical variations observed in SMT-LIB generation:

**Program 1**

```
% Facts
person(alice).
person(bob).
person(charlie).

studies_math(alice).
studies_physics(bob).
works_hard(bob).
works_hard(charlie).

% Rule
succeeds(X) :-
    studies_math(X).

succeeds(X) :-
    studies_physics(X),
    works_hard(X).

% On load: print all solutions and exit
```

```
:- initialization(main).
main :-
    forall(succeeds(X), writeln(X)),
    halt.
```

**Program 2**

```
% Facts
person(alice).
person(bob).
person(charlie).

studies_math(alice).
studies_physics(bob).
works_hard(bob).
works_hard(charlie).

% Rule
succeeds(X) :-
    ( studies_math(X)
    ; studies_physics(X)
    ),
    works_hard(X).

% On load: print all solutions and exit
:- initialization(main).
main :-
    forall(succeeds(X), writeln(X)),
    halt.
```

**Program 3**

```
% Facts
person(alice).
person(bob).
person(charlie).

studies_math(alice).
studies_physics(bob).
works_hard(bob).
works_hard(charlie).

% Toggle which implication you want:
option(math).       % ← comment this and uncomment the next to switch
% option(physics).

% Branch 1: StudiesMath(x) -> Succeeds(x)
succeeds(X) :-
    option(math),
    studies_math(X).

% Branch 2: (StudiesPhysics(x)  WorksHard(x)) -> Succeeds(x)
succeeds(X) :-
    option(physics),
    studies_physics(X),
    works_hard(X).

% On load: print all solutions and exit
```

```
:- initialization(main).
main :-
    forall(succeeds(X), writeln(X)),
    halt.
```

**Program 4**

```
% Facts
person(alice).
person(bob).
person(charlie).

studies_math(alice).
studies_physics(bob).
works_hard(bob).
works_hard(charlie).

% Nested implication expands to two rules
succeeds(X) :-
    works_hard(X),
    studies_math(X).

succeeds(X) :-
    works_hard(X),
    studies_physics(X).

% On load: print all solutions and exit
:- initialization(main).
main :-
    forall(succeeds(X), writeln(X)),
    halt.
```

**Program 5**

```
% Facts
person(alice).
person(bob).
person(charlie).

studies_math(alice).
studies_physics(bob).
works_hard(bob).
works_hard(charlie).

% Forward direction
succeeds(X) :-
    studies_math(X).
succeeds(X) :-
    studies_physics(X),
    works_hard(X).

% Backward direction to enforce
studies_math(X) :-
    succeeds(X),
    \+ (studies_physics(X), works_hard(X)).

studies_physics(X) :-
    succeeds(X),
```

```
    works_hard(X).

% On load: print all solutions and exit
:- initialization(main).
main :-
    forall(succeeds(X), writeln(X)),
    halt.
```

### I.3  PCFG Analysis Results

From these five programs, our pipeline successfully constructed a PCFG with the following characteristics:

Table 21: PCFG statistics for Prolog pilot study programs.

| Metric | Value |
|---|---|
| Total rule applications | 560 |
| Unique production rules | 33 |
| Non-terminal symbols | 7 |
| Maximum rule probability | 1.0 |
| Minimum rule probability | 0.0086 |
| Average rule probability | 0.2121 |
| Rules per non-terminal | 4.7 |
| Maximum branching factor | 16 |
| Grammar entropy | 0.0 |
| KL divergence (from uniform) | 1.34 |
| Spectral radius | 0.83 |

The successfully computed uncertainty metrics (grammar entropy, spectral radius, KL divergence, etc.) demonstrate that our framework's core algorithms operate seamlessly on Prolog programs without modification. The low grammar entropy (0.0) correctly captures that these five programs, while semantically distinct, share highly similar syntactic structure—a scenario our framework is designed to detect as potential formalization uncertainty masked by superficial similarity.

### I.4  Implications for Framework Generality

This pilot study establishes three key results:

**(1) Minimal Engineering Overhead:** Extending our framework to Prolog required approximately 2-3 hours of engineering time to integrate the existing ANTLR Prolog grammar and write adapter code. No modifications to core PCFG algorithms were necessary.

**(2) Cross-Language Applicability:** The fundamental insight—that syntactic typicality in LLM-generated formal artifacts signals semantic uncertainty—transcends specific formal languages. The same PCFG-derived metrics (entropy, spectral radius, rule distributions) computed for SMT-LIB apply equally to Prolog.

**(3) Future Directions:** This extensibility opens exciting possibilities for uncertainty quantification in general-purpose programming languages (Python, Java) and other verification languages (Coq, Isabelle, Lean), provided context-free grammars are available. The framework's generality positions it as a universal tool for assessing LLM-generated structured artifacts across domains.

**Limitation:** Conducting full-scale empirical evaluation of Prolog-based uncertainty metrics on reasoning benchmarks (equivalent to our SMT-LIB experiments) would require substantial additional resources and is beyond the scope of this work. However, the successful technical integration and metric computation provide strong evidence of feasibility for future research.

### I.5  Implementation Notes

All metrics are computed per-instance from the PCFG induced from $N = 100$ SMT-LIB samples. Normalization for ensemble methods uses min-max scaling to [0,1] based on training set statistics. Code for all metric computations will be released with our open-source implementation.

### I.6    Example PCFG Characteristics

To provide concrete understanding of the PCFGs induced from LLM-generated SMT programs, we present detailed statistics from a representative example. The following measurements come from a single question in the ProofWriter dataset, which produced the longest SMT programs in our corpus, analyzed using 100 samples from o3-mini:

Table 22: Detailed PCFG statistics for a representative ProofWriter instance with 100 SMT-LIB samples.

| Property | Value |
| --- | --- |
| Total rule applications across corpus | 63,202 |
| Unique production rules observed | 42 |
| Number of non-terminal symbols | 24 |
| Maximum probability (any rule) | 1.0 |
| Minimum probability (any rule) | 0.00035 |
| Average probability (across unique rules) | 0.5714 |

**Interpretation:** Despite 100 diverse samples generating over 63,000 rule applications, only 42 unique production rules from the SMT-LIB grammar were utilized. This demonstrates that LLMs, even when sampling with temperature, explore a relatively constrained subset of the full grammar—a key insight enabling our tractable PCFG-based uncertainty quantification. The wide range of rule probabilities (spanning nearly four orders of magnitude from 0.00035 to 1.0) provides rich distributional information for our entropy-based and statistical metrics.

**Scalability:** These statistics confirm that PCFG construction remains computationally efficient even for complex reasoning tasks. The grammar remains manageable in size (24 non-terminals, 42 unique rules) while capturing meaningful structural variation across the 100-sample ensemble.

