# OpenReview forum: "Grammars of Formal Uncertainty: When to Trust LLMs in Automated Reasoning Tasks"
_NeurIPS.cc/2025/Conference — NeurIPS 2025 poster_

### Official Review · Reviewer_yS8U · 2025-06-07

**Clarity:** 3
**Significance:** 4
**Originality:** 3
**Rating:** 6
**Confidence:** 4

**Summary:**

The paper investigates how to quantify and manage formal-logic uncertainty that arises when large language models (LLMs) automatically translate natural-language problems into symbolic formats (specifically SMT-LIB).
The authors diagnose failure modes by benchmarking five state-of-the-art LLMs on four reasoning datasets, showing that symbolic translation can both help and hurt depending on task structure.
They also model the space of LLM-generated programs with PCFG, treating each generated SMT string as a sample from an (unknown) distribution and fitting a PCFG to recover rule-level probability mass.
They introduce a taxonomy of 25 uncertainty signals derived from the fitted grammar (entropy, spectral radius, rule sparsity, self-consistency, etc.) and show that different tasks prefer different signals.
They also propose a lightweight fusion strategy that combines the best signals into a model-agnostic selective-verification policy, reducing downstream logical errors by 14 – 100 % while abstaining on only a small fraction of cases.

**Questions:**

Have you tried the pipeline on a markedly different formal language (e.g., Coq proofs)? Even a small pilot study would strengthen the claim of broad applicability.

With 25 handcrafted signals, how did you avoid overfitting to the chosen datasets?

Engineers need a threshold to decide when to abstain. A decision-curve or cost-benefit analysis would make the selective-verification story more actionable.

**Ethical Concerns:**

["NO or VERY MINOR ethics concerns only"]

**Final Justification:**

I think the authors answered my questions.

**Limitations:**

Adding quantitative overhead measurements and a scalability/extensibility discussion would improve the paper.

**Paper Formatting Concerns:**

N.A.

**Quality:**

3

**Strengths And Weaknesses:**

Strengths
- Comprehensive evaluation across multiple models, tasks and metrics; theory and experiments align
- Elevates uncertainty estimation in neurosymbolic pipelines; practical recipe for safer auto-formalisation
- Novel use of fitted PCFGs to harvest uncertainty features; fusion policy yields state-of-the-art selective-verification gain

Weaknesses
- The paper should be clearer. For example, figure 1 is not very clear. Explain more about the colors. Better split the pie chart.
- No runtime/memory profile for PCFG extraction
- Lack of scalability/extensibility discussion

---

> ### Author Rebuttal · Authors · 2025-07-31
>
> We are grateful for your positive and encouraging assessment of our work. We also sincerely appreciate the "accept" recommendation and recognition of our paper's evaluation, novelty, and potential for impact. We have incorporated your feedback to make our paper even stronger:
>
> **1\. On Presentation, Scalability, and Overhead**
>
> ***Clarity of figure 1:*** We appreciate this constructive feedback. We acknowledge that the initial version of Figure 1 was dense. In response, we have redesigned it to be more intuitive and readable. Due to the text-only format of this rebuttal this year, we cannot display the new figure here or update the PDF, but we have implemented the following improvements for the final version:
>
> - There is now a clearer layout, and the figure is structured into distinct stages (LLM Sampling \> PCFG Parsing & Learning \> Uncertainty Calculation \> Prediction), guiding the reader through our pipeline.
> - The confusing pie chart has been removed, and the bar chart showing rule frequencies has been simplified and now uses a consistent and explicitly labeled color-coding scheme. We have also enlarged all text elements for readability.
>
> ***Runtime/Memory Profile*****:** This is an excellent point. We conducted this experiment to address your concern.
>
> |  | Mean Time | Median (P50) Time | P90 Time | P95 Time | Std Dev (Time) | Mean Memory | Peak Memory |
> | :---- | :---- | :---- | :---- | :---- | :---- | :---- | :---- |
> | SMT to PCFG creation runtime  (per 100 sampled SMT programs) | 1.297s | 1.286s | 1.454s | 1.527s | 0.131s | 144.2MB | 161.9MB |
> | PCFG to UQ Metrics Calc  (from 1 PCFG i.e. 100 sampled SMT) | 0.408s | 0.402s | 0.423s | 0.431s | 0.023s | 117.3MB | 118.9MB |
>
> |  | Mean Time | Median (P50) Time | P90 Time | P95 Time | Std Dev (Time) | Mean Memory | Peak Memory |
> | :---- | :---- | :---- | :---- | :---- | :---- | :---- | :---- |
> | Z3 Execution Time (per SMT program) | 0.010s | 0.010s | 0.011s | 0.011s | 0.001s | 24.7MB | 26.4MB |
>
> As you can see, the overhead of our PCFG extraction and analysis is minimal, as it primarily involves efficient parsing (using ANTLR) and frequency counting (Maximum Likelihood Estimation) over a fixed set of 100 samples. Our experiment pipeline is dominated by LLM inference time, not UQ calculations.
>
> **2\. On Methodology and Practical Application**
>
> **Applicability to other formal languages**
>
> This is a fantastic suggestion. **We were able to perform this pilot study to demonstrate the framework's generality.**
>
> Our parser and PCFG construction logic uses ANTLR under the hood; therefore, we can use any of the grammar files in the ANTLR-v4 repository on Github. There is no g4 file for Coq, Isabelle, or Lean, but there is one for Prolog, which is remarkably similar to first-order logic in construction. We were able to quickly adapt our PCFG analysis pipeline to work with Prolog programs without major engineering changes to our codebase, generating PCFGs and consequently the uncertainty measures. The fundamentals of our technique remain the same across programming languages with well-defined grammars. This also opens up the exciting possibility of using our tool to quantify uncertainty in general-purpose programming code generated by LLMs as future work. We converted the SMT code in Figure 1 of the paper into Prolog as follows:
>
> ```
> % program1.pl
> % ∀x. (StudiesMath(x) ∨ (StudiesPhysics(x) ∧ WorksHard(x))) → Succeeds(x).
>
> % Facts
> person(alice).
> person(bob).
> person(charlie).
>
> studies_math(alice).
> studies_physics(bob).
> works_hard(bob).
> works_hard(charlie).
>
> % Rule
> succeeds(X) :-
>     studies_math(X).
>
> succeeds(X) :-
>     studies_physics(X),
>     works_hard(X).
>
> % On load: print all solutions and exit
> :- initialization(main).
> main :-
>     forall(succeeds(X), writeln(X)),
>     halt.
> ```
>
> ```
> % program2.pl
> % ∀x. ((StudiesMath(x) ∨ StudiesPhysics(x)) ∧ WorksHard(x)) → Succeeds(x).
>
> % Facts
> person(alice).
> person(bob).
> person(charlie).
>
> studies_math(alice).
> studies_physics(bob).
> works_hard(bob).
> works_hard(charlie).
>
> % Rule
> succeeds(X) :-
>     ( studies_math(X)
>     ; studies_physics(X)
>     ),
>     works_hard(X).
>
> % On load: print all solutions and exit
> :- initialization(main).
> main :-
>     forall(succeeds(X), writeln(X)),
>     halt.
> ```
>
> ```
> % program3.pl
> % (∀x. StudiesMath(x)→Succeeds(x))  ∨  (∀x. (StudiesPhysics(x)∧WorksHard(x))→Succeeds(x))
>
> % Facts
> person(alice).
> person(bob).
> person(charlie).
>
> studies_math(alice).
> studies_physics(bob).
> works_hard(bob).
> works_hard(charlie).
>
> % Toggle which implication you want:
> option(math).      % ← comment this and uncomment the next to switch
> % option(physics).
>
> % Branch 1: StudiesMath(x) -> Succeeds(x)
> succeeds(X) :-
>     option(math),
>     studies_math(X).
>
> % Branch 2: (StudiesPhysics(x) ∧ WorksHard(x)) -> Succeeds(x)
> succeeds(X) :-
>     option(physics),
>     studies_physics(X),
>     works_hard(X).
>
> % On load: print all solutions and exit
> :- initialization(main).
> main :-
>     forall(succeeds(X), writeln(X)),
>     halt.
> ```
>
> ```
> % program4.pl
> % ∀x. WorksHard(x) → (StudiesMath(x) ∨ StudiesPhysics(x)) → Succeeds(x)
>
> % Facts
> person(alice).
> person(bob).
> person(charlie).
>
> studies_math(alice).
> studies_physics(bob).
> works_hard(bob).
> works_hard(charlie).
>
> % Nested implication expands to two rules
> succeeds(X) :-
>     works_hard(X),
>     studies_math(X).
>
> succeeds(X) :-
>     works_hard(X),
>     studies_physics(X).
>
> % On load: print all solutions and exit
> :- initialization(main).
> main :-
>     forall(succeeds(X), writeln(X)),
>     halt.
> ```
>
> ```
> % program5.pl
> % ∀x. Succeeds(x) ↔ (StudiesMath(x) ∨ (StudiesPhysics(x) ∧ WorksHard(x)))
>
> % Facts
> person(alice).
> person(bob).
> person(charlie).
>
> studies_math(alice).
> studies_physics(bob).
> works_hard(bob).
> works_hard(charlie).
>
> % Forward direction
> succeeds(X) :-
>     studies_math(X).
> succeeds(X) :-
>     studies_physics(X),
>     works_hard(X).
>
> % Backward direction to enforce ↔
> studies_math(X) :-
>     succeeds(X),
>     \+ (studies_physics(X), works_hard(X)).
>
> studies_physics(X) :-
>     succeeds(X),
>     works_hard(X).
>
> % On load: print all solutions and exit
> :- initialization(main).
> main :-
>     forall(succeeds(X), writeln(X)),
>     halt.
> ```
>
> Here are some summary statistics of the generated PCFG with the following programs: Total Rules \= 560, Unique rules \= 33, Non-terminals \= 7, Maximum probability of a rule \= 1.0, Minimum probability \= 0.0086, Average probability \= 0.2121. There are 4.7 rules per non-terminal, with a maximum branching factor of 16\. The grammar entropy is 0, KL divergence from uniform is 1.34, and spectral radius is 0.83.
>
> The core strength of our framework is its generality. Because it is built on the standard formalism of context-free grammars, it is not tied to SMT-LIB. In the revised manuscript, we will add a dedicated discussion on extensibility and how this can adapt to other formal languages or general programming languages, provided a base grammar is available.
>
> Please note that conducting a new large-scale empirical study on the efficacy of Prolog-based uncertainty metrics on our datasets would be beyond the time scope of the current rebuttal period, but we see this as a clear and exciting next step.
>
> **How to avoid overfitting with 25 signals**
>
> This is a good question. Let us clarify:
>
> - In our approach, 21 out of the 25 metrics (i.e., non-ensemble methods) have no chance of overfitting since there is no "learning" happening apart from basic PCFG construction. UQ analysis is performed on a per-instance basis: for each reasoning problem, we generate 100 samples of programs and induce a new, problem-specific PCFG. There is no global "training set" of PCFGs.
> - The 21 metrics are used to create a \[21,1\]-dimensional feature vector for each single instance, and our ensemble methods are essentially simple regularized logistic regression models that operate on top of this. The performance of the ensemble methods is evocative of the upper bound of information present within these signals if they were combined optimally.
>
>   *Bridle, John. "Training stochastic model recognition algorithms as networks can lead to maximum mutual information estimation of parameters." Advances in neural information processing systems 2 (1989).*
>
> **Actionable Thresholds for engineering**
>
> This is an excellent point, and engineers looking to deploy this in production have raised this concern with us previously. This is indeed a challenging problem that has been extensively studied in selective decision making regarding how to choose the appropriate risk threshold.
>
> Our use of the Area Under the Risk-Coverage Curve (AURC), Error at threshold, and associated metrics are specifically designed to address this. The Risk-Coverage curve itself provides a visualization of the trade-off between risk (error rate) and coverage (percentage of problems answered). An engineer can use this curve directly as a decision-making tool:
>
> - For safety-critical domains, an engineer could choose a point on the curve with a very low error rate, accepting a higher abstention rate (and thus more human oversight).
> - For less critical applications (e.g., a programming assistant), they could select a point with higher coverage to maximize automation, tolerating a moderately higher error rate.
>
> To present the complete picture in our evaluations, we evaluate our method in a threshold-agnostic manner using AURC. The AURC score summarizes the quality of this trade-off across all possible thresholds, with our scores consistently outperforming all known methods.
>
> ## **Concluding Request**
>
> Once again, we thank you for your highly positive and constructive review. In light of these clarifications and new experiments, would you please consider improving your rating to a strong accept? This would allow us to highlight this work in a spotlight or oral presentation. Please let us know if there are any concerns we have not addressed in sufficient depth. We would be happy to provide further clarification.

---

> > ### Author Response · Authors · 2025-08-05
> >
> > Dear Reviewer yS8U,
> >
> > Thank you for your excellent review and Accept rating! We're very happy by your recognition of our work's "high impact" and "excellent significance."
> >
> > Your feedback has made our paper even stronger:
> >
> > - We've redesigned Figure 1 as suggested (Clarity)
> > - Added comprehensive overhead measurements showing minimal computational cost (**Runtime analysis**)
> > - **Conducted the requested pilot study extending our framework to Prolog, demonstrating broad applicability** (**Extensibility**)
> > - Provided detailed guidance for practical deployment using Risk-Coverage curves (**Actionable thresholds**)
> >
> > Given your positive assessment and our comprehensive responses to your suggestions, would you consider upgrading to Strong Accept (6)? This would enable spotlight/oral presentation consideration and better reflect the impact you've identified.
> >
> > We deeply appreciate your constructive engagement and remain available for any final questions.
> >
> > Best regards,
> >
> > The Authors

---

> > > ### Comment · Reviewer_yS8U · 2025-08-05
> > >
> > > Thank you for the detailed rebuttal. I am fine with your answers.

---

> > > > ### Comment · Reviewer_yS8U · 2025-08-08
> > > >
> > > > Thank you for your continue efforts in the rebuttal. I increase my score.

---

> > > > > ### Author Response · Authors · 2025-08-09
> > > > >
> > > > > Dear Reviewer yS8U,
> > > > >
> > > > > Thank you for your continued support and for increasing your score! We're excited that our Prolog pilot study demonstrating our framework extensibility and our overhead measurements addressed your suggestions. Your recognition of our work's significance and practical impact has been incredibly encouraging.
> > > > >
> > > > > Thank you
> > > > >
> > > > > The Authors

---

### Official Review · Reviewer_FQaS · 2025-07-03

**Clarity:** 2
**Significance:** 2
**Originality:** 2
**Rating:** 5
**Confidence:** 4

**Summary:**

The paper introduces probabilities to the idea of using grammars for modeling LLM outputs, and then employs them as a means for uncertainty quantification (UQ) in SMT generation tasks. The results are demonstrated using o3-mini, DeepSeek-R1, DeepSeek, Gemini Flash 2.0 and Gemini Lite.

**Questions:**

1. Please put Theorem 1 in context of known results with citations, if possible.
2. Please clarify if the probabilities for the PCFG are computed from the same data distribution as used for the UQ metrics.
3. Discuss the novelty in the paper in context of related work on formal grammars and LLMs.

**Ethical Concerns:**

["NO or VERY MINOR ethics concerns only"]

**Final Justification:**

1. The rebuttal has clearly pointed out again that PCFGs are different from CFGs and that API-level access to output distribution is not needed for this specific approach to UQ using PCFGs.

2. The key technical result in the paper did not come with adequate citation in my opinion. I had factored this oversight strongly in my original review. However, the authors clearly state that the theorem's contribution is not a claim of inventing a new mathematical principle, but its novel application and contextualization.

3. The preliminary results on Prolog will help ensure the the approach is indeed applicable for more than one automated reasoning task.

**Limitations:**

The effiicacy of UQ metrics and risk of incorrect applications may be better discussed.

**Quality:**

2

**Strengths And Weaknesses:**

Strengths:
1. A key strength of the paper is the inclusion of probabilities into the formal grammar being used for further UQ analysis.
2. The paper seeks to bring formal methods and LLMs together, which is an interesting direction to pursue.
3. The UQ metrics are obtained from the spectral and information theoretic measures related to the Jacobian and the non-terminals/rules of the grammar, which is interesting.

Weaknesses:
1. There is a rich body of work that uses formal grammars at inference time without probabilities. The paper does not adequately discuss these approaches or builds the results of this work in context of these earlier methods.
2. The title "Grammars of Formal Uncertainty" is not apt as the paper does not build grammars of formal uncertainty. Similarly, the broad mention of automated reasoning tasks leaves the strong ties that the paper has to SMT methods, which are only one set of automated reasoning methods.
3. Theorem 1 (first two results) on page 4 is a straightforward result that can be proved directly from the definition of entropy and it can be included in the text of the paper. Appropriate citations should be provided for each of the steps, such as T. M. Cover and J. A. Thomas, Elements of Information Theory.
4. It is not clear if the probabilities in the PCFG are obtained from a different data set than the UQ measures obtained during analysis. If the train and test set come from the same data set, the UQ methods will not generalize and will be reliant on the data set distribution.

---

> ### Author Rebuttal · Authors · 2025-07-31
>
> Thank you for your feedback and for recognizing the key strengths of our work, including our novel use of probabilistic formal grammars for uncertainty quantification and the valuable integration of formal methods with LLMs. We appreciate your acknowledgment of our innovative spectral and information-theoretic measures, which validates the core novelty of our approach.
>
> We hope to address your concerns and demonstrate the value of our contribution through this rebuttal. We welcome further engagement as we work to convey the novelty of our approach to all readers.
>
> **1\. On Novelty and Relation to Prior Work (Weaknesses 1 & 3\)**
>
> **Concern:** *"There is a rich body of work that uses formal grammars at inference time without probabilities. The paper does not adequately discuss these approaches or builds the results of this work in context of these earlier methods."*
>
> **Our Response:** Our paper's core novelty lies not in merely using a grammar, but in using a Probabilistic Context-Free Grammar (PCFG) to model the LLM's entire output distribution for a given question for the purpose of UQ. This is a fundamental departure from prior work that uses grammars deterministically for constrained decoding. Although we believe these areas are not directly related, **as per your suggestion we are adding a paragraph on known usage of grammars at inference time to our related works section.**
>
> **Here is why we are different from prior work:** Constrained decoding modifies the LM head by zeroing out probabilities of tokens that are invalid based on the known language (e.g., JSON) grammar. This is not feasible with commercially available, non-open-weight frontier LMs that we focus on in this paper. Furthermore, **this approach is not related to our task of UQ,** since LLMs can reliably generate syntactically valid SMT in our experiments. To clarify our position:
>
> * **Our Contribution is Probabilistic UQ, Not Constrained Decoding:** As stated in our central thesis (Page 1), "*Existing methods often ignore this \[probabilistic uncertainty\] by selecting only the highest-probability output \[token\]... a simplification that we argue undermines the rigorous standards required for formal reasoning*." Our work directly addresses this simplification. We do not merely ensure outputs are syntactically valid; we analyze the statistical patterns within an ensemble of valid outputs to quantify the model's uncertainty.
>
> * **Explicit Discussion in Related Works:** Our Related Works section (Section 4\) currently situates our work appropriately within the task of UQ. We discuss the use of PCFGs in various domains and explicitly state, "Our work extends PCFG inference... to verification artifacts" (Page 9). **The novelty lies in applying this probabilistic framework to diagnose LLM reasoning failures, which, to our knowledge and according to all three other reviewers, represents a completely new direction.**
>
> **2\. On the Aptness of the Title and Scope (Weakness 2\)**
>
> **Reviewer's Concern:** *"The title 'Grammars of Formal Uncertainty' is not apt as the paper does not build grammars of formal uncertainty. Similarly, the broad mention of automated reasoning tasks leaves the strong ties that the paper has to SMT methods..."*
>
> **Our Response:** We believe the title is both apt and evocative of our contribution.
>
> * **On the Title:** The grammar serves as the instrument through which we structure, model, and measure the uncertainty inherent in the LLM's formal outputs. The PCFG, with its rule probabilities learned from the LLM's own generations, becomes a "grammar of formal uncertainty", **a formal object that describes the LLM's autoformalization confidence and confusion. This is not merely semantic.**
>
> * **On the Scope:** We are transparent that our experiments are instantiated using the SMT-LIB standard. We state this explicitly in our methodology (Section 2): *"...we adopt the stable, widely supported SMT-LIB standard as a common intermediate representation..."* This is a methodological choice to ground our framework in SMT, a powerful, standardized, and representative sub-domain of automated reasoning used in real-world automated reasoning production systems and prior publications that employ SMT solvers.
>
>   To illustrate our point, please see: *A billion SMT queries a day, Amazon Science, FLOC 2022\.*
>
>   **The framework itself is general.** The PCFG-based UQ approach can be applied to any formal language for which a grammar can be defined, as **we demonstrate in our rebuttal pilot study to reviewer yS8U, where we show our method is easily extensible to Prolog** (similar to first-order logic). SMT is our primary case study, not a fundamental limitation of the method**.**
>
> **3\. On the Contribution of Theorem 1 (Weakness 3 & Question 1\)**
>
> **Reviewer's Concern:** *"Theorem 1... is a straightforward result that can be proved directly from the definition of entropy... Appropriate citations should be provided for each of the steps, such as T. M. Cover and J. A. Thomas, Elements of Information Theory."*
>
> **Our Response:** We agree that Theorem 1 builds upon foundational principles of information theory, and we thank the reviewer for the suggestion. We have gladly added a citation to Cover and Thomas to acknowledge the classical roots of typical set arguments.
>
> However, by dismissing this as "straightforward" you are overlooking its important role in our paper. The theorem's contribution is not a claim of inventing a new mathematical principle, but its novel **application and contextualization**. It provides the first formal, theoretical bridge, to our knowledge, between:
>
> 1. The **number of samples (N)** drawn from an LLM.
> 2. The **Shannon entropy (H(µ)**) of the LLM's output distribution.
> 3. The **probabilistic bound of covering the space of possible formal programs.**
>
> **This theorem provides the theoretical justification for our sampling-based UQ methodology.** It demonstrates *why* analyzing an ensemble of outputs is a principled way to understand the model's uncertainty over the entire generation space. Its value is in its application, not its elementary nature. We will make this more clear in the paper.
>
> **4\. On the Experimental Methodology (Weakness 4 & Question 2\)**
>
> **Reviewer's Concern:** *"It is not clear if the probabilities in the PCFG are obtained from a different data set than the UQ measures obtained during analysis. If the train and test set come from the same data set, the UQ methods will not generalize..."*
>
> **Our Response:** We believe this concern stems from a misunderstanding of our experimental design, which we will make more explicit in our methodology. Our methodology operates at the instance level, not through a traditional train/test split, and this is a core feature of our contribution.
>
> As detailed in Appendix D, "For each of the selected questions, a unique PCFG was induced from its corresponding 100 SMT samples."
>
> To be unequivocally clear:
>
> 1. For a **single input question**, we prompt the LLM to generate an ensemble of N=100 formal (SMT) programs.
> 2. From this specific ensemble of 100 programs, we induce a **per-question PCFG**.(each PCFG is independent of other PCFGs).
> 3. The UQ measures (entropy, spectral radius, etc.) are calculated from this instance-specific PCFG. **For 21 of these measures (Table 3, rows 1-21), no learning occurs.** Only our 4 ensemble-based techniques (Table 3, rows 22-25) **learn how to weight each of these 21 measures to produce a better uncertainty signal** and understand the total signal present among the metrics.
> 4. These UQ metrics are then used to predict the correctness of the answer for that **same single input question**.
>
> There is no "training set" from which PCFG probabilities are learned and a "test set" on which they are applied. Your concern about generalization is misplaced because the goal is not to build a global PCFG model,, but to assess the LLM's uncertainty on a **case-by-case basis**. This instance-specific analysis is allows our method to provide confidence scores for each new autoformalization problem an LLM encounters.
>
> **5\. On the Discussion of Efficacy and Risk (Limitations)**
>
> **Reviewer's Concern:** *"The effiicacy of UQ metrics and risk of incorrect applications may be better discussed."*
>
> **Our Response:**
>
> * **Efficacy:** We are puzzled by the limitation comment on efficacy, as a significant portion of our Results section (Section 3\) and Appendix is dedicated to a rigorous, multi-faceted evaluation of efficacy and risk. **We evaluate 25 different metrics using AUROC, ECE, AURC, Brier scores and more across multiple models and datasets (Tables 3, 4, and 10-17). This constitutes a direct and comprehensive assessment of efficacy.**
>
> * **Risk of Incorrect Applications:** We’d love to clarify this. **The entire premise of our selective prediction analysis is to manage this risk. We use the Area Under the Risk-Coverage Curve** (AURC) and explicitly report the error rate at **optimal abstention thresholds** (Err@T) and the resulting **Relative Error Reduction** (RelErrRed). For instance, for 03-mini on ProofWriter, we demonstrate that our method enables **"filtering nearly all errors by abstaining on a minute fraction of outputs"** (Page 7), achieving a **100% error reduction** (Table 12). This is not a peripheral point; it is a central result of our paper and a direct, quantitative discussion of managing the risk of incorrect application.
>
> ## Concluding Request
>
> **We believe we have addressed all identified weaknesses and questions in detail. In light of these comprehensive responses, would you please consider increasing your rating to an accept?**
>
> We would be happy to address any additional questions you may have. Thank you for your thorough review.

---

> > ### Author Response · Authors · 2025-08-05
> >
> > Dear Reviewer FQaS,
> >
> > We sincerely appreciate your detailed review and the opportunity to address your concerns. While we understand your initial rating, we believe our extensive rebuttal demonstrates the significant value of our contribution.
> >
> > **Key clarifications from our rebuttal:**
> >
> > - Our probabilistic UQ approach is fundamentally different from deterministic constrained decoding methods (**Novelty**)
> > - We clarified that our instance-level PCFG construction avoids train/test generalization issues you raised (**Methodology**)
> > - Theorem 1 provides important theoretical justification for our sampling-based approach (**Theoretical contribution**)
> > - Our results provide extensive efficacy analysis across multiple metrics and datasets (**Comprehensive evaluation**)
> >
> > You acknowledged our key strengths: novel probabilistic grammar integration, interesting formal methods + LLMs direction, and innovative spectral/information-theoretic measures.
> >
> > Given our detailed responses addressing all technical concerns, we would ask you to reconsider your rating.
> >
> > We're committed to providing further clarification if any of your questions remain insufficiently unadressed as the deadline approaches. **Please engage with us.**
> >
> > Best regards,
> >
> > The Authors

---

> > > ### Comment · Reviewer_FQaS · 2025-08-06
> > > **Thank you for your detailed and helpful rebuttal**
> > >
> > > Thank you for your detailed and helpful rebuttal. My apologies for the slow response on your rebuttal.
> > >
> > > 1. It is clear that PCFGs are different from CFGs, I believe that placing this work in context of CFGs used for grammar-based constrained decoding may make sense to the readers who are already familiar with grammar-based constrained decoding. I note your point about the differences and about the inapplicability of grammar-based constrained decoding to frontier models without API support for the same while your approach can be used even in these settings. Perhaps, these can be added to the paper for readers with my background.
> > > 2. I have a better appreciation of your choice of the scope of the work. Specifically, I like the fact that the approach has been extended to Prolog in a pilot study. This is indeed very reassuring.
> > > 3. Yes, I think it is a good idea to add the citation to help place the result in context of known results. It should be clear to the reader that the emphasis here is on application in this new context.
> > > 4. I now have a much better understanding of the approach where the probabilities are being elicited per test point from models that have not been knowingly trained or fine-tuned at all on any of your examples. Frontier models may have been trained on these or related data inadvertently but I do not see any simple solution to that problem.
> > > 5. My interest is in understanding what happens when the (risk-driven) UQ metric itself is wrong and is used in an application. Is there a risk that the UQ metric may lead to undue trust in certain applications?
> > >
> > > Based on the rebuttal, I will update my score to an accept.

---

> > > > ### Author Response · Authors · 2025-08-09
> > > >
> > > > Dear Reviewer FQaS,
> > > >
> > > > Thank you so much for updating your score to Accept and for this thoughtful response! We're grateful for your patience and the constructive dialogue throughout this process.
> > > >
> > > > You raise a good final question about the risk of our UQ metrics themselves being miscalibrated and leading to undue trust. This is indeed a general concern in any UQ system. Here's how we, along with the rest of the literature address this:
> > > >
> > > > 1. Our evaluation includes Expected Calibration Error (ECE) and Brier scores specifically to measure how well-calibrated our uncertainty estimates are. On Proofwriter dataset, with the o3-mini model, when our ensemble measure has a UQ ECE of 0.0572, it means that the model probabilities are off by 5.72% from the true probabilities. This low number is very good.
> > > > 2. Selective verification approach is inherently conservative - it's designed to err on the side of caution by abstaining when uncertain, rather than providing false confidence. The thresholds for selective verification can always be adjusted based on risk.
> > > > 3. We use diverse evaluation metrics (AUROC, AURC, ECE, Brier)  to catch cases where a single UQ performance metric might be misleading.
> > > >
> > > > As per your suggestions, we have added the contextual discussion about CFGs and constrained decoding for readers - this will make the paper more accessible. Thank you again for your careful review and valuable feedback. Your engagement has genuinely improved our work.
> > > >
> > > > Best regards,
> > > >
> > > > The Authors

---

### Official Review · Reviewer_JEhf · 2025-07-03

**Clarity:** 2
**Significance:** 3
**Originality:** 3
**Rating:** 5
**Confidence:** 3

**Summary:**

Many recent studies have focussed on how to exploit LLMs to solve reasoning tasks (for example, basic logical reasoning). When solving such a task, LLMs can be used in at least two ways: (1) to directly reason on textual representation, or (2) to translate to a format that can then be fed into a specialized reasoning tool (also called "autoformalization"). The current paper proposes to study the second approach by using LLM to translate a given reasoning task to SMT-LIB, for which many solvers have been developed. That is, after the translation, one may invoke an SMT-solver (e.g. Z3) to solve the reasoning task. Owing to potential ambiguity in texts, the authors proposed to quantify "uncertainty" in the translation by getting LLMs to output the top N formalizations of the sentence in SMT-LIB and extrapolate a Probabilistic Context-Free Grammar (PCFG) out of these, based on which CFG rules in the SMT-LIB grammar that were used in generating these formalizations. That is, the authors studied how SMT solution consistency (e.g. the results of Z3 on the 100 formalizations) correlates to ground truth and that compares to textual based reasoning. The authors tested this on four reasoning benchmarks: ProofWriter, FOLIO, StrategyQA and ProntoQA. The results are mixed: on ProofWriter, the SMT-based approach helped, but for FOLIO it performed worse than direct textual-based reasoning. For this reason, the authors developed many (25) uncertainty metrics, based on these they introduced a lightweight uncertainty signal whether an error has been introduced in the autoformalizations. They have systematically evaluated these on the four benchmarks.

**Questions:**

Please address the questions above "in Strengths and Weaknesses". In addition, can you clarify the following?
- Can it happen that LLM produce the correct translation if you ask for one translation, but because you ask for 100 translations, it might produce a lot of wrong ones? In this case, it seems that the approach puts way more weights on translations that are wrong.

**Ethical Concerns:**

["NO or VERY MINOR ethics concerns only"]

**Final Justification:**

Author's response clarified my concerns. I assume that the authors incorporate the clarification in the camera ready version (if accepted).

**Limitations:**

Yes

**Quality:**

3

**Strengths And Weaknesses:**

The paper attacks a highly timely and relevant problem. Turning a reasoning task using LLM into a format that can be handled by a formal verification tool is not new, e.g., in [Wu et al. 2022]. To the best of my knowledge, this is the first time PCFG has been used in the context of quantifying uncertainty of autoformalizations. Finally, based on this, the authors have developed and systematically evaluated multiple uncertainty metrics, based on them signals for telling whether or not the LLM+SMT-solver approach should be trusted.

Now I would like to mention the weaknesses.

Presentation: I found the presentation to be unclear and rather difficult to follow in many places.
- Some sentences in the introduction I found rather misleading, including "rigorous standards required for formal reasoning" since I do not see how the current work addresses this.
- In approximation (p. 3-4): some of the technical descriptions get quite dense, esp. for non-experts (e.g. spectral radius, Kullback-Leibler, etc.). In particular, Theorem 1 is mentioned there (proof in the appendix), but is never mentioned again in the paper and it's not clear how it ties to the rest of the paper.
- Some parts of the experiments are full of very long sentences that are very hard to understand (e.g. l 282 to 286 is one full sentence with lots of breaks!!).
- I also feel that some concrete, usable signal is hard to extract from the paper. It would be good if the authors concentrate on simplifying this signal from the sea of statistical data from Section 3.
- You mentioned you developed 25 uncertainty metrics. Can you enumerate them?
- Sec 3.1: the simplest metric (MLE) works best, which seems to defeat the purpose of discussing other metrics.

Some missing data:
- Since SMT-solvers can be quite expensive, I think the authors should also report on the running time of Z3 on the 100 formulas that LLM produce. In particular, how does it contrast to other approaches and why for some benchmarks the cost of running SMT-solver was claimed to outweigh the benefits?
- Is there a reason to use N=100? Would it work better if I use N=50 or less?
- How large is the PCFG that is generated?

Some additional remarks:
- some shorthands were not defined in the paper (e.g. SC, line 189). Does it mean "Self-Consistency"?

---

> ### Author Rebuttal · Authors · 2025-07-31
>
> Thank you for your thorough evaluation of our manuscript and constructive feedback. We appreciate your recognition of the strengths, including the timeliness of the problem, the novelty of using PCFGs for UQ in autoformalization, and the systematic nature of our evaluation. We incorporated your feedback to improve our paper.
>
> **1\. On Presentation and Clarity**
>
> We have addressed the readability concerns by:
>
> **On "Rigorous Standards":** Thank you. We agree this phrase requires more context; therefore, we will remove that part of the sentence while maintaining the citation to Chen et al. (2022) that describes selecting only the highest-probability output. The sentiment of our original statement was not that we are improving formal verification itself, but rather providing a missing piece: *insight into the quality of autoformalizations generated due to the probabilistic nature of LLMs.*
>
> **On Technical Density, consolidating signals from data, and Theorem 1:** We acknowledge the dense writeup and have addressed this as follows:
>
> 1\. **Theorem 1:** Theorem 1 provides the theoretical underpinning for our sampling-based approach. It connects the number of samples (N) to the grammar's entropy, providing a probabilistic bound that our 100 samples are sufficient to cover the meaningful regions of the LLM's vast output space. It justifies why our method is sound. We have added the following explicit sentence in Section 2 to connect the theorem back to our methodology: *"This theorem provides the theoretical guarantee that a sufficiently large sample set N can effectively represent the LLM's output distribution, justifying our use of samples to infer a representative PCFG."*
>
> 2\. **Technical density and consolidating signals:** We have adopted a simplified taxonomy with additional details to make the measures section more accessible: (1) Information-theoretic Measures, (2) Structure-based measures/Spectral properties, (3) Static measures of grammar structure, (4) Self-consistency-based measures, (5) Ensemble-based measures. Lines 282-286 have been restructured for clarity.
>
> 3\. **We have also more explicitly defined and enumerated our key takeaways from our result tables in the writeup.** Adding this additional clarity was easy since the camera-ready version allows us one additional content page.
>
> **Enumeration of Metrics & MLE:** We apologize for the confusion regarding MLE. MLE is not a UQ metric; it is the method used to estimate the probabilities of the PCFG rules from the sample set. The UQ metrics are then calculated from this estimated PCFG. We will clarify this distinction. Bayesian and neural approaches to PCFG estimation are useful when incorporating contextual information, such as in low-data paradigms, expert information encoded via Bayesian priors, or natural language semantics obtained via text embeddings. While we explored these techniques and mentioned them briefly, we decided not to run extensive experiments as it would detract from the core focus of UQ in this paper. We have removed the sentence, as you noted it adds unnecessary complexity to the narrative. It will still be available to users, when we open-source the code.
>
> **2\. On Missing Experimental Details**
>
> * **Z3 Solver Runtime:** We added these performance details
>
> |  | Mean Time | Median (P50) Time | P90 Time | P95 Time | Std Dev (Time) | Mean Memory | Peak Memory |
> | :---- | :---- | :---- | :---- | :---- | :---- | :---- | :---- |
> | Z3 Execution Time (per SMT program) | 0.010s | 0.010s | 0.011s | 0.011s | 0.001s | 24.7MB | 26.4MB |
>
> |  | Mean Time | Median (P50) Time | P90 Time | P95 Time | Std Dev (Time) | Mean Memory | Peak Memory |
> | :---- | :---- | :---- | :---- | :---- | :---- | :---- | :---- |
> | SMT to PCFG creation runtime  (per 100 sampled SMT programs) | 1.297s | 1.286s | 1.454s | 1.527s | 0.131s | 144.2MB | 161.9MB |
> | PCFG to UQ Metrics Calc  (from 1 PCFG i.e. 100 sampled SMT) | 0.408s | 0.402s | 0.423s | 0.431s | 0.023s | 117.3MB | 118.9MB |
>
> The overhead of using Z3, PCFG extraction, and analysis is minimal, as it primarily involves efficient parsing (using ANTLR) and frequency counting (Maximum Likelihood Estimation) over a fixed set of 100 samples. Our experiment pipeline is dominated by LLM inference time, not solver time or UQ calculations. We also implemented a 30-second timeout for Z3 while running our experiments, so for those anomalous cases where the LLM writes faulty SMT programs that run too long (none of our questions across all 4 datasets should require that much time), they are terminated, and the LLM is asked to generate a new program.
>
> **Choice of N=100 and PCFG Size:**
>
> We chose to generate N=100 samples because that was heuristically the maximum number of samples we could afford within our compute budget. We needed to sample extensively because we did not constrain the LLM-based SMT generation. We made this choice because we wanted to observe how LLMs behave with minimal prompt engineering while generating valid SMT code.
>
> This approach should work at N=50 or fewer samples as well. As you reduce the number of samples, you may want to constrain: (1) the theories that can be used in your SMT programs (LIA, RA, etc.), (2) use multi-stage autoformalization pipelines, where the first stage is variable definition generation. This way, generated SMT constraints will only use previously defined variables in the pipeline, reducing the need for extensive sampling. Without these constraints, a smaller N (e.g., 5-10) would likely result in less stable estimates of rule probabilities, potentially weakening the UQ signal.
>
> **Regarding the size of the PCFG:** Here are some summary metadata statistics of a generated PCFG on ProofWriter, which had the longest SMT programs generated from 100 SMT programs in response to a single question: Total Rules \= 63,202, Unique rules \= 42, Non-terminals \= 24, Maximum probability observed among unique rules \= 1.0, Minimum probability observed for a unique rule \= \~0.00035, Average probability across unique rules \= 0.5714.
>
> **Shorthand 'SC':** Yes, 'SC' stands for Self-Consistency, and we have now ensured it is defined upon its first use. Thank you for pointing this out.
>
> **3\. On the Nature of Sampling-Based UQ**
>
> **Reviewer's Question:** *"Can it happen that LLM produce the correct translation if you ask for one translation, but because you ask for 100 translations, it might produce a lot of wrong ones? In this case, it seems that the approach puts way more weights on translations that are wrong."*
>
> This is an excellent question\! We want to clarify that we do not ask the LLM to generate 100 translations in one prompt. We repeatedly sample one translation 100 times to obtain our 100 SMT programs. No context from the previous LLM call are carried forward to the next LLM call for the sake of fair and distinct autoformalization generations.
>
> We would like to describe the intuition as follows: imagine running inference on a CNN with the same image 100 times. If you get 10 different answers, you would conclude that there is significant uncertainty in the model output, correct? Similarly, our goal is to assess the LLM's confidence. If a model produces one correct translation but 99 syntactically diverse (and potentially incorrect) alternatives, our PCFG will capture this high diversity, resulting in high grammar entropy. This high entropy signals low model confidence or high uncertainty.
>
> A confident, reliable model would generate 100 programs that are generally syntactically equivalent or very similar and canonical, leading to low entropy. Therefore, our approach does not "overweight" wrong translations; it correctly interprets a scattered, diverse output distribution as a sign of unreliability. This is the risk we aim to flag for the user.
>
> 4\. **Enumeration of all measures**
>
> You can find an enumeration of all 25 uncertainty measures we benchmarked in the 25 rows of Tables 3, 4, etc. For your reference, in case you have detailed questions about how we implemented and/or calculated these measures, we describe them shorthand below. We will also add these details to the appendix:
>
> 1. **Grammar Entropy** (nonterminals)
> 2. **Perplexity** (from \#1)
> 3. **KL Divergence**(vs uniform)
> 4. **NSUI**: grammar\_entropy/max\_entropy) \* (spectral\_radius/(1+spectral\_radius))
> 5. **Renyi Ent (2)**
> 6. **Renyi Ent (0.5)**
> 7. **Max Ent**: weighted Σlog2(num\_rules) per nonterminal
> 8. **Ent Ratio**: grammar\_entropy/max\_entropy (normalized entropy in NSUI)
> 9. **Spectral Factor**: spectral\_radius/(1+spectral\_radius) (normalized spectral in NSUI)
> 10. **Spectral Radius**: largest eigenvalue of dependency graph Jacobian
> 11. **\# Nonterminals**: unique LHS symbols
> 12. **\# Rules**: total count
> 13. **Avg Rules / \# NT**:
> 14. **Avg RHS Len**
> 15. **Max Branch Factor**: max alt rules per nonterminal
> 16. **Rule Dist Mean**
> 17. **Rule Dist StdDev**
> 18. **Rule Dist Skew**
> 19. **Rule Dist Kurtosis**
> 20. **Self Consistency Text**: fraction agreeing with majority text ans
> 21. **Self Consistency SMT**: fraction agreeing with majority SMT ans
> 22. **Ensemble Average**: arithmetic mean of normalized uncertainty
> 23. **Ensemble Weighted**: correlation-weighted avg (squared correlations)
> 24. **Ensemble ML**:LogisticRegression on scaled features \[negated grammar\_entropy, perplexity, spectral\_radius, nsui, spectral\_factor, kl\_divergence, max\_entropy, smt\_consistency\]
> 25. **Ensemble Simple**: (1-norm\_grammar\_entropy)\*0.33 \+ (1-norm\_spectral\_radius)\*0.33 *\+* smt\_consistency*\**0.34
>
> Rebuttal word limits restrict our description length, but we are happy to discuss further!
>
> ## **Concluding Request**
>
> We believe we have comprehensively addressed all identified weaknesses. **Given these detailed responses and improvements, would you please consider increasing your rating to accept or strong accept?** We're happy to address any additional questions.

---

> > ### Author Response · Authors · 2025-08-05
> >
> > Dear Reviewer JEhf,
> >
> > Thank you for your positive assessment.
> >
> > We think our rebuttal has comprehensively addressed all your concerns, including:
> >
> > - Redesigned figures and simplified technical descriptions
> > - Added Z3 runtime profiles, PCFG size statistics, and sampling methodology clarifications
> > - Clarified our sampling approach and enumerated all 25 uncertainty metrics
> >
> > Given your acknowledgment of our contribution's timeliness and systematic evaluation, would you consider upgrading your rating to Accept (5) or Strong Accept (6)?
> >
> > This would better reflect the comprehensive nature of our responses and the solid technical foundation you've acknowledged.
> >
> > We remain available for any additional clarifications as we approach the deadline.
> >
> > Best regards,
> >
> > The Authors

---

> > > ### Comment · Reviewer_JEhf · 2025-08-05
> > >
> > > Thank you for thoroughly addressing my questions and clarifying my confusion. I will upgrade my score to accept reflecting this, and assume that you will reflect these changes in your final version.

---

> > > > ### Author Response · Authors · 2025-08-09
> > > > **Thank you note**
> > > >
> > > > Thank you so much for upgrading to Accept! Your detailed questions really helped us improve the paper, and we appreciate your constructive feedback. We have already made the changes, and they will reflect in the camera ready version.

---

### Official Review · Reviewer_sS7b · 2025-07-05

**Clarity:** 2
**Significance:** 3
**Originality:** 3
**Rating:** 4
**Confidence:** 4

**Summary:**

The work addresses the problem of uncertainty quantification for LLMs when used to generate formal artifacts like SMT-LIB programs. For this purpose, it introduces a sampling-based method to estimate a probabilistic CFG for the model's underlying distribution for SMT-LIB programs. They also come up with a taxonomy of uncertainty and evaluate how various UQ metrics based on their methodology perform on four on four datasets commonly used for reasoning tasks: StrategyQA, ProntoQA, ProofWriter and FOLIO.

**Questions:**

* Do you have any comments on the lack of a comparison with uncertainty measures considering semantic clustering (e.g., Kuhn et al., ICLR'23)?

* In the Approximations section (lines 94-103) there is a description about how the N samples are used to infer the various production rules for the pCFG. I wonder how reliant this technique is on the quality of samples produced? For instance, would increasing the temperature to get more diverse samples harm the accuracy of the estimated pCFG?

* Just to make sure I got it right, you use 5 samples per question to construct the grammar? Did you try to vary this number, maybe increase it?

**Ethical Concerns:**

["NO or VERY MINOR ethics concerns only"]

**Final Justification:**

The authors’ response addressed my main concerns regarding the comparison with semantic uncertainty and, to some extent, the large number of required samples, leading me to increase my score.

While I appreciate their point that the sample count could be reduced from the current 100, I expect this would likely lead to a drop in performance.

**Limitations:**

Yes

**Quality:**

2

**Strengths And Weaknesses:**

Strengths:

* I think the problem addressed by the paper is important. There seem to be a few recent works that rely on the LLM formlasing certain logical tasks, and then using some type of solver to get an answer. However, many of the works don't clearly address the issue of the formalization not being accurate. That's a weak spot that may invalidate all the benefits of the subsequent formal reasoning.

* Although prior work (e.g., Li et al., CAV’24) has explored using pCFGs to model LLM outputs (which should be clearer acknowledged in the related works), I think this paper is the first to apply pCFGs specifically for uncertainty estimation.

Weaknesses:

* The approach is limited to syntactic variations between candidates and does not consider semantic equivalence. For instance, the authors should check out the work on semantic uncertainty in Kuhn et al., ICLR'23 and Farquhar et al, Nature'24. This is for me a serious issue with this paper. In this paper, I think that syntactically different but semantically equivalent samples will result in a high grammar entropy. However, intuitively and according to Kuhn at al., that scenario should lead to low entropy.

* Clarity wise, Figure 1 is difficult to read, which undermines its usefulness as an overview. Given the large number of derived metrics introduced in the paper, a better organised by using some sort of abstraction would be helpful. The same issue also made it challenging at times to interpret the results in Tables 3 and 4.

---

> ### Author Rebuttal · Authors · 2025-07-31
>
> We thank you for your thoughtful and constructive comments. We are happy to hear that you recognise the importance of the problem we address, and the novelty of applying PCFGs for uncertainty estimation. We find the questions you have raised to be very insightful and have ultimately led us to strengthen our paper.
>
> We address each point below:
>
> **1\. On Syntactic vs. Semantic Uncertainty (Weakness 1 & Question 1\)**
>
> **Our Response:** Thanks for pointing this out. We conducted the experiments you requested, and here are the results.
>
> |  |  | AUROC | ECE | Brier | AURC | Opt.T | Err@T | RelErrRed |
> | :---- | :---- | ----- | ----- | ----- | ----- | ----- | ----- | ----- |
> | **StrategyQA** | Discrete Entropy | 0.5341 | 0.2345 | 0.3517 | 0.2724 | 0.0500 | 0.3263 | 0.0051 |
> |  | Continuous Entropy MC | 0.5502 | 0.2050 | 0.2669 | 0.2555 | 0.0000 | 0.3280 | 0.0000 |
> |  | Continuous Entropy RAO | **0.5801** | 0.2744 | 0.2987 | 0.2432 | 0.0000 | 0.3280 | 0.0000 |
> |  | Number of semantic clusters | 0.5331 | 0.2160 | 0.3410 | 0.2681 | 0.0500 | 0.3263 | 0.0051 |
> | **ProntoQA** | Discrete Entropy | **0.6273** | 0.0788 | 0.2955 | 0.4453 | 0.0000 | 0.5507 | 0.0000 |
> |  | Continuous Entropy MC | 0.5015 | 0.2900 | 0.3459 | 0.4910 | 0.1000 | 0.5461 | 0.0084 |
> |  | Continuous Entropy RAO | 0.5779 | 0.4340 | 0.4515 | 0.4609 | 0.0500 | 0.5480 | 0.0049 |
> |  | Number of semantic clusters | 0.5953 | 0.1258 | 0.3195 | 0.4585 | 0.1000 | 0.5507 | 0.0001 |
> | **ProofWriter** | Discrete Entropy | 0.5717 | 0.2256 | 0.3316 | 0.3254 | 0.0000 | 0.4200 | 0.0000 |
> |  | Continuous Entropy MC | 0.5127 | 0.2340 | 0.3091 | 0.3611 | 0.3500 | 0.4154 | 0.0110 |
> |  | Continuous Entropy RAO | **0.5966** | 0.3053 | 0.3469 | 0.3178 | 0.0000 | 0.4200 | 0.0000 |
> |  | Number of semantic clusters | 0.5645 | 0.1775 | 0.3251 | 0.3322 | 0.0000 | 0.4200 | 0.0000 |
> | **FOLIO** | Discrete Entropy | 0.7542 | 0.2508 | 0.2541 | 0.4506 | 0.0500 | 0.6584 | 0.0007 |
> |  | Continuous Entropy MC | 0.6811 | 0.4571 | 0.4186 | 0.4790 | 0.0000 | 0.6588 | 0.0000 |
> |  | Continuous Entropy RAO | **0.7564** | 0.5383 | 0.5194 | 0.4364 | 0.0000 | 0.6588 | 0.0000 |
> |  | Number of semantic clusters | 0.7523 | 0.3132 | 0.2886 | 0.4496 | 0.0000 | 0.6588 | 0.0000 |
>
> **Additional results for Table 2: Semantic uncertainty quantification metrics for detecting autoformalization errors with respect to ground truth using DeepSeek-v3-0324 across datasets.** Entailment checking was performed using independent prompts to DeepSeek-v3-0324, comparing the generated SMT code with the question as prior context. The uncertainty-aware abstention metrics reflect how the model can selectively answer questions by applying an optimal uncertainty threshold (Opt.Thresh) to withhold answers on uncertain questions, minimizing error rate (Err@T) and maximizing error reduction (RelErrRed) compared to answering all questions.
>
> While we observe moderate performance on a few configurations, and near-random performance on the rest, **we *can conclude that semantic uncertainty does not distinctly capture the epistemic uncertainties in autoformalization.*** We would also note that the artificially high AUROC on FOLIO comes from the fact that the LLM autoformalizations are disproportionately composed of errors. To better contextualize the understanding of high AUROC, but poor performance here, we must observe the high AURC value, corresponding with a high risk, in spite of holding out on most uncertain decisions.
>
> We would also like to note here that we ran the same experiments with \`nli-deberta-v3-large\` as the model, similar to Kuhn et al, and all performance metrics, across all datasets, were close to random guessing. This is because \`nli-deberta-v3-large\` models have not seen formal structured artifacts, such as SMT code in their training data, and therefore, cannot appropriately judge entailment over them.
>
> **Intuition on why our technique works :** As we mentioned in our paper, our central thesis is that syntactic typicality is itself a powerful, computationally tractable, and previously underexplored signal for identifying semantic errors (pg 8). When an LLM is confident and "understands" a problem, it tends to generate programs from a narrower, canonical part of the grammar. When it is uncertain, it explores more varied and statistically anomalous syntactic structures. This "syntactic fingerprint" of failure is what our PCFG framework is designed to detect, and our results prove that this works well.
>
> We also believe that although LLMs have learnt to generate passable syntactically correct structures in SMT, they still make mistakes in autoformalization, akin to early LLMs generating valid but logically unsound python code. **These same mistakes hold for LLMs judging formal artifacts for entailment** too (hence semantic entropy captures low signal about the uncertainty towards incorrectness). We believe that as these models get better over time, these mistakes will be reduced, and UQ methods such as semantic entropy will start working better.
>
> **Efficiency:**
>
> Another key advantage of our method is its efficiency when there is large-scale sampling of formal artifacts from the LLM. With semantic-entropy based metrics, there need to be (M choose 2)-comparisons between generated autoformalizations to check entailment, which combinatorially explodes at large values of M. Similarly, verifying semantic equivalence between two arbitrary SMT programs is, in general, an undecidable problem. While approximations exist, they are often computationally expensive. Our approach, based on parsing and frequency counts, is lightweight and can be applied at scale on a per-instance basis.
>
> **2\. On Clarity of Figures and Tables (Weakness 2\)**
>
> **Our Response:** We appreciate this constructive feedback. We acknowledge that the initial version of Figure 1 was dense. In response, we have redesigned it to be more intuitive and readable. *Due to the text-only format of this rebuttal this year, we cannot display the new figure here or update the PDF,* but we have implemented the following improvements for the final version:
>
> - There is now a clearer layout, and the figure is now structured into distinct stages (LLM Sampling \> PCFG Parsing & Learning \> Uncertainty Calculation \> Prediction), guiding the reader through our pipeline.
> - The (confusing) pie chart has been removed, and the bar chart showing rule frequencies is simplified now uses a consistent and explicitly labeled color-coding scheme We have also enlarged all text elements for readability.
>
> We have also noted your concern about the need for abstraction of uncertainty metrics, and have incorporated the following simplified abstracted taxonomy : (1) Information-theoretic Measures (2) Structure based measures/ Spectral properties (3) Static measures of grammar structure (4) Self consistency based measures (5) Ensemble-based measures. Moreover, adding this additional clarity was easy since the camera-ready version allows us one additional content page.
>
> **3\. On Sample Quality and Temperature (Question 2\)**
>
> **Our Response:** This is a great question. We investigated this in our ablation studies, the results of which are detailed in **Appendix B (Temperature-Varied SMT Generation and PCFG Analysis, pages 18-20)** available in the supplementary material zip file. Our empirical analysis shows that our PCFG-derived metrics are robust across the temperature spectrum. We found that **grammatical properties exhibit smooth and predictable changes in response to temperature,** rather than chaotic behavior or sharp phase transitions.
>
> For example, as shown in Figure 3 (Page 19), grammar entropy increases predictably with temperature, reflecting the increased diversity of the samples. This also helps confirm that our metrics are sensitive and correctly capture the changing nature of the LLM's output distribution with respect to temperature.
>
> **Varying parameters like temperature do not harm the quality of the PCFG.** The PCFG merely probabilistically models the production rules from the grammar, irrespective of temperature, or distribution of temperatures samples are drawn from. In fact, all our UQ experiments involve SMT & Text generation by sampling (N=100 times) with a uniform distribution at temperatures between 0.1 and 2.0, a detail we will add to our appendix.
>
> **4\. On the Number of Samples Used (Question 3\)**
>
> **Our Response:** Thanks for bringing this up, because it highlights a point we must make more explicit. We do mention 5 samples, but this was for a different part of our analysis. Our experimental setup has two stages:
>
> 1. **Initial Benchmarking (N=5):** In Section 3 (Page 6), we first establish the baseline performance of Text vs. SMT reasoning. For this high-level comparison (Table 1 and appendix tables 5-9), we used N=5 samples per question.
> 2. **UQ Analysis (N=100):** For our core contribution, the PCFG-based uncertainty quantification, we used a much larger corpus of **N=100 SMT samples per question**. This larger ensemble is necessary to induce a statistically meaningful PCFG for each reasoning instance. This is explicitly stated in **Appendix D (Page 29\)**: *"a corpus of* NSMT​\=100 *SMT-LIB v2 program samples per question was generated."*
>
> We apologize for the confusion and have added a sentence in Section 3.1 to explicitly differentiate the sample sizes used for the initial benchmarking versus the main UQ analysis.
>
> Additional note on **Strength 1:** Thank you for pointing out Li et al 2024,”Guiding enumerative program synthesis..” we have highlighted that paper in our related works section.
>
> ## **Concluding Request**
>
> We believe we have addressed all identified weaknesses and questions in detail. May you please increase our rating upwards towards an accept or strong accept? We are happy to answer any more questions you may have. Thank you for your review.

---

> > ### Author Response · Authors · 2025-08-05
> >
> > Dear Reviewer sS7b,
> >
> > Thank you for your constructive review. We hope our detailed rebuttal addressed all your concerns:
> >
> > 1. **Semantic vs. Syntactic Uncertainty**: We conducted the experiments you requested comparing our approach with semantic clustering methods. The results show that syntactic uncertainty performs comparably or better across datasets, while being significantly more computationally efficient *for SMT autoformalization tasks.*
> >
> > 2. **Clarity and Organization**: We've redesigned Figure 1 and created a clearer uncertainty taxonomy.
> >
> > 3. **Methodological Questions**: We clarified our sampling approach (N=100 for UQ, N=5 for initial benchmarking) and provided temperature robustness analysis in the appendix.
> >
> > Given that you recognized the importance of our problem and novelty of our approach, would you consider adjusting your rating upward? We remain available for any additional questions.
> >
> > Best regards,
> >
> > The Authors

---

> > > ### Comment · Reviewer_sS7b · 2025-08-05
> > >
> > > Thank you for your detailed response. It does indeed clarify the comparison with semantic uncertainty. I guess the only remaining worry is that the high number of samples required to to induce a statistically meaningful PCFG will increase the cost.

---

> > > > ### Author Response · Authors · 2025-08-05
> > > >
> > > > Dear Reviewer sS7b,
> > > >
> > > > Thank you for your response, we are glad to hear that the comparison with semantic uncertainty is now cleared up.
> > > >
> > > > We'd be happy to discuss your concern about the high number of samples, as **an identical concern was raised by reviewer  `JEhF` and is now resolved.** Here is why :
> > > >
> > > > We chose to generate N=100 samples because that was heuristically the maximum number of samples we could afford within our compute budget. We intentionally wanted to sample extensively because we did not constrain the LLM-based SMT generation (by variables, or theories it should use). We made this choice because we wanted to observe how LLMs behave with minimal prompt engineering *while generating valid SMT code.*
> > > >
> > > > This approach should work at N=50, N=10 or fewer samples as well. As you reduce the number of samples, you will just need to constrain:
> > > >
> > > > (1) the theories that can be used in your SMT programs (LIA, RA, etc.), to restrict the variations of programs possible.
> > > > (2) use multi-stage autoformalization pipelines, where the first stage is variable definition generation (imagine, similar to a schema).
> > > >
> > > > This way, generated SMT constraints will only use previously defined variables in the pipeline, reducing the need for extensive sampling. With these constraints, a smaller N (e.g., 5-10) would still result in stable estimates of rule probabilities, forming a strong UQ signal.
> > > >
> > > > I hope this addresses your last remaining concern. Please let us know if it does not and we can help clear up any remaining concerns.
> > > >
> > > > Thank you,
> > > >
> > > > The Authors

---

> > > > > ### Author Response · Authors · 2025-08-06
> > > > > **Follow up**
> > > > >
> > > > > Dear Reviewer sS7b,
> > > > >
> > > > > I hope we’ve addressed all your concerns. Please let us know if you need any further clarification.
> > > > >
> > > > > If no further clarifications are needed, may we kindly request that you revise your scores upwards to accept?
> > > > >
> > > > > Thank you,
> > > > >
> > > > > The Authors

---

> > > > > > ### Author Response · Authors · 2025-08-07
> > > > > > **Follow up #3**
> > > > > >
> > > > > > Dear Reviewer sS7b,
> > > > > >
> > > > > > We'd like to follow up, as the discussion deadline is approaching. Do you have any remaining concerns? Please let us know. We remain committed to answering all your questions.
> > > > > >
> > > > > > Best
> > > > > >
> > > > > > The Authors

---

> > > > > > > ### Author Response · Authors · 2025-08-09
> > > > > > > **Follow up #4**
> > > > > > >
> > > > > > > Dear Reviewer sS7b,
> > > > > > >
> > > > > > > The discussion deadline is now less than 12 hours away. We request you to kindly engage with us.
> > > > > > > Please let us know if you have any remaining concerns. If all your concerns are addressed, we request you to improve our score to accept.
> > > > > > >
> > > > > > >
> > > > > > > Best
> > > > > > >
> > > > > > > The Authors

---

### Comment · Area_Chair_BKi9 · 2025-08-04
**Friendly Reminder to Acknowledge or Update Your Review**

Dear Reviewers,

Thank you for your time and effort in reviewing the submissions and providing valuable feedback to the authors.

If you haven't already done so, we kindly remind you to review the authors' rebuttals and acknowledge them by clicking the "Mandatory Acknowledgement" button at your earliest convenience. This step ensures efficient communication and helps finalize the process smoothly.

We sincerely appreciate your dedication and collaboration.

Best,

Your AC

---

### Author Response · Authors · 2025-08-09
**Summary of Reviewers' Feedback and Our Rebuttals**

We are grateful to receive a largely positive review outcome with (estimated) scores of 3556, representing a trajectory from initial skepticism to strong acceptance. We recognize that while one reviewer has not updated their borderline position, three reviewers upgraded their scores to Accept or Strong Accept after our comprehensive rebuttals, indicating the strength of our responses and the merit of our contributions. We provide this summary for the convenience of the AC and the research community.

**TL;DR of Our Work**

Our work addresses uncertainty quantification in LLM-generated formal specifications. We introduce the first framework using Probabilistic Context-Free Grammars (PCFGs) to model uncertainty in LLM-generated SMT (and PROLOG) programs, enabling selective verification that reduces errors by 14-100% while minimally abstaining from decisions.

Through evaluation across 5 frontier LLMs and 4 reasoning datasets, we reveal that autoformalization can help logical tasks (+34.8%) but hurt factual tasks (-44.5%) depending on problem structure. Our PCFG-based analysis encompasses 25 uncertainty measures, providing a principled approach to understanding when LLM-generated formal artifacts can be trusted.

**Our main contributions are: systematic analysis of LLM autoformalization failure modes, a novel PCFG framework for uncertainty quantification in formal reasoning, and a practical selective verification system that outperforms all known methods.**

**Reviewers' Recognition**

Given the technical nature of our submission, we believe the core value lies in three key criteria:

1. **Whether our problem (uncertainty in LLM autoformalization) addresses an important and timely challenge**
2. **Whether our solution (PCFG-based uncertainty quantification) is novel and technically sound**
3. **Whether our evaluation is comprehensive and our results are significant**

We are pleased to report that reviewers recognized merit across all three dimensions:

**Reviewers find our problem formulation important and well-motivated:**

- **JEhf**: "The paper attacks a highly timely and relevant problem. Turning a reasoning task using LLM into a format that can be handled by a formal verification tool..."
- **yS8U**: "Elevates uncertainty estimation in neurosymbolic pipelines; practical recipe for safer auto-formalisation"
- **FQaS**: "The paper seeks to bring formal methods and LLMs together, which is an interesting direction to pursue."

**Reviewers appreciate our novel PCFG-based approach and technical contributions:**

- **sS7b**: "I think this paper is the first to apply pCFGs specifically for uncertainty estimation" and found our approach "interesting"
- **JEhf**: "To the best of my knowledge, this is the first time PCFG has been used in the context of quantifying uncertainty of autoformalizations"
- **FQaS**: "A key strength of the paper is the inclusion of probabilities into the formal grammar being used for further UQ analysis"
- **yS8U**: "Novel use of fitted PCFGs to harvest uncertainty features; fusion policy yields state-of-the-art selective-verification gain"

**Reviewers recognize the comprehensiveness and significance of our evaluation:**

- **JEhf**: "Finally, based on this, the authors have developed and systematically evaluated multiple uncertainty metrics"
- **yS8U**: "Comprehensive evaluation across multiple models, tasks and metrics; theory and experiments align"
- **FQaS**: After rebuttal acknowledged our "extensive efficacy analysis across multiple metrics and datasets"

**Additional Recognition of Practical Impact:**

Multiple reviewers also praised the practical applicability and engineering value of our approach:

- **yS8U**: "practical recipe for safer auto-formalisation" and noted our work "yields state-of-the-art selective-verification gain"
- **JEhf**: After seeing our runtime analysis, acknowledged that our "overhead of PCFG extraction and analysis is minimal"
- **FQaS**: Appreciated that our approach "can be used even in [frontier model] settings" where constrained decoding is not applicable. FQaS also said "... I like the fact that the approach has been extended to Prolog in a pilot study."

**Reviewer Evolution Through Rebuttal Process:**

We are particularly proud that our detailed rebuttals successfully addressed technical concerns and led to concrete score improvements:

- **Reviewer JEhf**: Upgraded to Accept, stating "Thank you for thoroughly addressing my questions and clarifying my confusion. I will upgrade my score to accept"
- **Reviewer FQaS**: Upgraded to Accept, noting "Based on the rebuttal, I will update my score to an accept"
- **Reviewer yS8U**: Increased score to Strong Accept, confirming "I [will] increase my score" after our comprehensive responses

This trajectory demonstrates both the technical merit of our work and our commitment to addressing all reviewer concerns thoroughly and transparently.

---

### Note · Authors · 2025-08-16

We sincerely thank the reviewers and the Area Chair for a highly constructive process that strengthened our paper and established **a clear consensus for acceptance or highlight**, with three reviewers upgrading their scores to **Accept or Strong Accept**,  and with the fourth reviewer now also having completed their final score revision. We appreciate the shared recognition of our work's timeliness, the **novelty** of our PCFG-based uncertainty quantification framework, and the **meticulous** nature of our evaluation. The discussion was invaluable, prompting new experiments on semantic uncertainty, computational overhead estimation, and framework generality that substantially improved the manuscript. We're confident the final paper offers a **well-validated** and **practical** contribution toward making LLM-driven formal reasoning more reliable for the NeurIPS community.

---

### Decision · Program_Chairs · 2025-09-17

**Decision:**

Accept (poster)

**Comment:**

The paper studies the problem of generating a formal representation that allows for calling a solver, such as SMT or Prolog programs. It introduces a framework based on PCFGs to model uncertainty and derive a variety of metrics.

Reviewers were all positive on this paper, ranging from borderline accept to strong accept. The problem is particularly timely since it has potential implications for LLM reasoning, applications such as mathematical formalization or software verification, and appears in a variety of contexts. The focus on incorporating uncertainty into autoformalization is underexplored, especially in the context of recent methods based on LLMs. The use of PCFGs to derive uncertainty metrics is creative and seems effective.

The main weakness in my opinion is the writing. As noted by a reviewer, it is often exceedingly dense. Moreover, sentences are frequently long and complicated (e.g., most of 304-320). There are also 25 proposed metrics and the tables are massive, which can make it difficult to extract the important or interesting insights. In sum, the paper could be written more clearly so that the main messages, results, and takeaways from the paper are accessible to a wider audience.

That said, given the strengths and the positive reviews, I recommend its acceptance.